# Beyond Stationarity: Convergence Analysis of Stochastic Softmax Policy Gradient Methods

## Abstract

Markov Decision Processes (MDPs) deliver a formal framework for modeling and solving sequential decision-making problems. In this paper, we make several contributions towards the theoretical understanding of (stochastic) policy gradient methods for MDPs. The focus lies on proving convergence (rates) of softmax policy gradient towards global optima in undiscounted finite-time horizon problems, i.e. $\gamma = 1$, without regularization. Such problems are relevant for instance for optimal stopping or specific supply chain problems. Our estimates must differ significantly from several recent articles that involve powers of $(1 - \gamma)^{-1}$.

The main contributions are the following. For undiscounted finite-time MDPs we prove asymptotic convergence of policy gradient to a global optimum and derive a convergence rate using a weak Polyak-Łojasiewicz (PL) inequality. In each decision epoch, the derived error bound depends linearly on the remaining duration of the MDP. In the second part of the analysis, we quantify the convergence behavior for the stochastic version of policy gradient. The analysis yields complexity bounds for an approximation arbitrarily close to the global optimum with high probability.

As a by-product, our stochastic gradient arguments prove that the plain vanilla REINFORCE algorithm for softmax policies indeed approximates global optima for sufficiently large batch sizes.

## 1   Introduction

Policy gradient methods continue to enjoy great popularity in practice due to their model-free nature and high flexibility. Despite their far-reaching history (Williams, 1992; Sutton et al., 1999; Konda and Tsitsiklis, 1999; Kakade, 2001), there were no proofs for the global convergence of these algorithms for a long time. Nevertheless, they have been very successful in many applications, which is why numerous variants have been developed in the last few decades, whose convergence analysis, if available, is mostly limited to convergence to stationary points (Pirotta et al., 2013; Schulman et al., 2015; Papini et al., 2018; Clavera et al., 2018; Shen et al., 2019; Xu et al., 2020b; Huang et al., 2020; Xu et al., 2020a; Huang et al., 2022).

In recent years, notable advancements have been achieved in the convergence analysis towards global optima (Fazel et al., 2018; Agarwal et al., 2021; Mei et al., 2020; Bhandari and Russo, 2021, 2022; Cen et al., 2022; Xiao, 2022; Alfano and Rebeschini, 2023). These achievements are partially attributed to the utilization of (weak) gradient domination or Polyak-Łojasiewicz (PL) inequalities (Polyak, 1963). As examined in Karimi et al. (2016) a PL-inequality and smoothness implies a linear convergence rate for gradient descent methods. In certain cases, only a weaker form of the PL inequality can be derived, which states that it is only possible to limit the norm of the gradient instead of the squared norm of the gradient by the distance to the optimum. Despite this limitation, $\mathcal{O}(1/n)$-convergence can still be achieved in some instances.

The research community has predominantly focused on discounted Markov decision processes (MDPs) with infinite time horizon: $(\mathcal{S}, \mathcal{A}, p, r, \gamma)$ is an MDP, where $\mathcal{S}$ is a finite state space, $\mathcal{A}$ is a finite action space, $p$ is a transition function such that $p(s'|s,a)$ denotes the probability of transitioning from state $s$ to state $s'$ under action $a$. The reward function is given by $r : \mathcal{S} \times \mathcal{A} \to R$, where $R \subseteq \mathbb{R}$ is usually assumed to be bounded and positive, and $\gamma \in (0,1)$ is a discount factor. The value function under consideration takes the form

$$V^\pi(s) = \mathbb{E}_{S_0=s, A_t \sim \pi(\cdot|S_t), S_{t+1} \sim p(\cdot|S_t, A_t)} \Big[ \sum_{t=0}^{\infty} \gamma^t r(S_t, A_t) \Big], \tag{1}$$

for all $s \in \mathcal{S}$. Investigating a stationary policy applied in every time point suffices for discounted MDPs (Puterman, 2005, Theorem 6.1.1). Yet, in this paper, we focus on MDPs with finite-time horizons and without a discount factor, i.e., $\gamma = 1$. There is a prevailing argument that finite-time MDPs do not require additional scrutiny as they can be transformed into infinite horizon MDPs. However, specific challenges arise in certain scenarios, such as optimal stopping (Li et al., 2009) or finite-time inventory control problems (Bhandari and Russo, 2022), where a non-stationary policy becomes necessary. Unlike in infinite time horizon MDPs, reducing the problem to stationary policies is inadequate for finite-time MDPs, and a new policy must be trained recursively at each time step (Puterman, 2005). Our convergence analysis comprises two steps: firstly, we investigate convergence at each time step and secondly, we examine the error accumulation through backward induction. A detailed discussion of finite-time MDPs is presented in Section 2. There are some recent articles also studying policy gradient of finite-time horizon MDPs considering fictitious discount algorithms (Guo et al., 2022) or finite-time linear quadratic control problems (Hambly et al., 2021, 2022; Zhang et al., 2021).

We begin with a discussion of relevant results for discounted MDPs that encourage our contributions. In Agarwal et al. (2021), the global asymptotic convergence of policy gradient is demonstrated under tabular softmax parametrization, and convergence rates are derived using log-barrier regularization and natural policy gradient. Building upon this work, Mei et al. (2020) showed the first convergence rates for policy gradient using non-uniform PL-inequalities (Mei et al., 2021), specifically for tabular softmax parametrization. However, this convergence rate is fundamentally dependent on the discount factor, $(1 - \gamma)^{-6}$, and cannot be readily extrapolated to undiscounted MDPs with finite-time horizons.

To bridge this gap, we consider policy gradient under tabular softmax parametrization, but in undiscounted MDPs with finite-time horizons and non-stationary policies. In Section 3, we show asymptotic convergence to a global optimum and subsequently derive a global convergence rate using a weaker form of the PL-inequality. The convergence rate at a fixed time point is linearly depending on the remaining duration of the MDP, which is a better property compared to $(1 - \gamma)^{-6}$. The issue of dependency on $\gamma$ when it approaches 1 is a significant subject in the context of discounting, and various efforts have been made to mitigate this dependency. For instance, employing entropy regularization as demonstrated in Mei et al. (2020) or applying mirror descent as described in Xiao (2022) can enhance the rate of convergence.

In the second part of the paper, we abandon the assumption that the exact gradient is known and focus on the model free stochastic policy gradient method. For this type of algorithm, very little is known even in the discounted case. Agarwal et al. (2021) discussed the approximate natural policy gradient for log-linear policies, and Ding et al. (2022) derived complexity bounds for entropy-regularized stochastic policy gradient. They use a well-chosen stopping time which measures the distance to the set of optimal parameters, and simultaneously guarantees convergence to the regularized optimum prior to the occurrence of the stopping time by using a small enough step size and large enough batch size. Similar to this idea, we construct a different stopping time in this work, which allows us to analyze convergence of the stochastic policy gradient method in the finite, non-stationary case and also in the infinite discounted case without regularization. The stopping time we propose measures the distance between the policy gradient and stochastic policy gradient trajectories and stops when the stochastic gradient differs too far from the exact gradient updates. This allows us to derive complexity bounds for an approximation arbitrarily close to the global optimum that does not require a set of optimal parameters, which is relevant when considering softmax parametrization.

To the best of our knowledge, the results presented in this paper provide the first convergence analysis of softmax policy gradient in the undiscounted finite-time MDP setting without regularization. We note that discussions in Bhandari and Russo (2022) do not apply to softmax parametrization, as they assume the existence of optimal parameters in the parameter space.

The remainder of this manuscript is structured as follows: In Section 2, we discuss finite-time MDPs and explain how to solve them using backward induction. In Section 3, we show asymptotic convergence to a global optimum and derive the corresponding convergence rate. Moreover, in Section 4, we present the results pertaining to finite-time stochastic policy gradient and in Section 5 we analyze the error accumulation using backward induction for exact and stochastic gradients. In Section 6, we provide our findings regarding infinite discounted MDPs, where we derive complexity bounds for the REINFORCE algorithm.

## 2   Finite-time horizon MDPs

A finite-time MDP is defined by a tuple $(\mathcal{H}, \mathcal{S}, \mathcal{A}, r, p)$ with $\mathcal{H} = \{0, \ldots, H-1\}$ decision epochs, finite state space $\mathcal{S} = \mathcal{S}_0 \cup \cdots \cup \mathcal{S}_{H-1}$, finite action space $\mathcal{A} = \bigcup_{s \in \mathcal{S}} \mathcal{A}_s$, a reward function $r : \mathcal{S} \times \mathcal{A} \to \mathbb{R}$ and transition function $p : \mathcal{S} \times \mathcal{A} \to \Delta(\mathcal{S})$ with $p(\mathcal{S}_{h+1}|s, a) = 1$ for every $h < H-1$, $s \in \mathcal{S}_h$ and $a \in \mathcal{A}_s$. Let $\Delta(D)$ denote the set of all probability measures over a finite set $D$. Due to finite decision epochs, the choice of the action is time dependent, i.e. non-stationary policies $\pi = (\pi_h)_{h=0}^{H-1}$ must be considered, where $\pi_h : \mathcal{S}_h \to \Delta(\mathcal{A})$ for every $h \in \mathcal{H}$ is such that $\pi_h(\mathcal{A}_s|s) = 1$ for every $s \in \mathcal{S}_h$. Denote by $\pi_{(h)} = (\pi_k)_{k=h}^{H-1}$ the sub-policy of $\pi$ form $h$ to $H-1$, and define the $h$-state value function under policy $\pi$ for every $s \in \mathcal{S}_h$ by

$$V_h^{\pi_{(h)}}(s) := \mathbb{E}_s^{\pi_{(h)}} \Big[ \sum_{k=h}^{H-1} r(S_k, A_k) \Big], \quad h \in \mathcal{H}, \tag{2}$$

where $\mathbb{E}_s^{\pi_{(h)}}$ is the expectation under the measure such that $S_h = s$, $A_k \sim \pi_k(\cdot|S_k)$ and $S_{k+1} \sim p(\cdot|S_k, A_k)$ for $h \le k < H-1$. The $h$-state-action value function for every tuple $(s, a) \in \mathcal{S}_h \times \mathcal{A}_s$ is defined by

$$Q_h^{\pi_{(h+1)}}(s, a) := r(s, a) + \sum_{s' \in \mathcal{S}_{h+1}} p(s'|s, a) V_{h+1}^{\pi_{(h+1)}}(s'), \quad h \le H-2. \tag{3}$$

Note that $Q_h$ is independent of policy $\pi_h$ and for $H-1$, $Q_{H-1}(s, a) := r(s, a)$ independently of any policy. Furthermore, define the $h$-state-action advantage function

$$A_h^{\pi_{(h)}}(s, a) := Q_t^{\pi_{(h+1)}}(s, a) - V_h^{\pi_{(h)}}(s), \quad s \in \mathcal{S}_h, a \in \mathcal{A}_s. \tag{4}$$

In the following, we will suppress the dependence of $\pi_{(h)}$ and write $\pi$ in the superscripts of $V_h$, $Q_h$ and $A_h$, when the policy is clear out of context. We denote by

$$V_h^{\pi}(\mu_h) := \mathbb{E}_{s \sim \mu_h}[V_h^{\pi}(s)]$$

the value function for an initial state distribution $\mu_h$ on $\mathcal{S}_h$ in epoch $h \in \mathcal{H}$. The performance difference lemma (Kakade and Langford, 2002) is a useful identity to compare policies. It turns out to be very useful to prove convergence of policy gradient methods (Agarwal et al., 2021). For finite-time MDPs the following version is proved in the supplementary material:

**Lemma 2.1** (Performance difference lemma). *For any $h \in \mathcal{H}$ and for any pair of policies $\pi$ and $\pi'$ the following holds true for every $s \in \mathcal{S}_h$:*

$$V_h^{\pi}(s) - V_h^{\pi'}(s) = \sum_{k=h}^{H-1} \mathbb{E}_{S_h=s}^{\pi_{(h)}} \Big[ A_k^{\pi'}(S_k, A_k) \Big].$$

In order to address finite-time MDPs it becomes necessary to consider non-stationary policies because the optimal decision at each time point depends on the time horizon until the end of the problem. Thus, to solve finite-time MDPs with policy gradient a time-dependent parametrization of the policy is required. Consider a parameter space denoted by $\Theta = \Theta_0 \times \cdots \times \Theta_{H-1}$, where a policy parameter $\theta = (\theta_0, \ldots, \theta_{H-1}) \in \Theta$ includes $H$ different parameters. A parametric policy $\pi^{\theta} = (\pi^{\theta_h})_{h=0}^{H-1}$ is defined such that the policy in epoch $h$ depends only on the parameter $\theta_h$. It is worth noting that finite-time MDPs are typically solved using backward induction as known from dynamic programming theory (Puterman, 2005). In order to obtain the optimal solution for a finite-time MDP through backward induction the parametrization must have the capability to approximate any deterministic policy. This is because deterministic optimal policies exist for finite-time MDPs similar to discounted

MDPs. These conditions have made the tabular softmax policy a subject of extensive research in the context of discounted MDPs, owing to its ability to meet these requirements (Mei et al., 2020; Agarwal et al., 2021; Ding et al., 2022). Let $\Theta_h = \mathbb{R}^{d_h}$ for all $h \in \mathcal{H}$, where $d_h = \sum_{s \in \mathcal{S}_h} |\mathcal{A}_s|$ the number of state-action pairs in epoch $h$. Then the tabular softmax parametrization is defined to be

$$\pi^\theta(a|s) = \frac{\exp(\theta(s,a))}{\sum_{a' \in \mathcal{A}} \exp(\theta(s,a'))}, \quad \theta = (\theta(s,a))_{s \in \mathcal{S}_h, a \in \mathcal{A}_s} \in \mathbb{R}^{d_h}. \tag{5}$$

In the forthcoming chapters, we will center our convergence analysis on this parametrization. Nevertheless, we emphasize that the results presented in this section are also valid for any other parametrization.

To solve a finite-time MDP the problem is partitioned into $h$ sub-problems, with each epoch being considered separately. Given any fixed policy $\tilde{\pi}$, the objective function in epoch $h$ is defined to be the $h$-state value function in state $s \in \mathcal{S}_h$ under the policy $(\pi^{\theta_h}, \tilde{\pi}_{(h+1)}) := (\pi^{\theta_h}, \tilde{\pi}_{h+1}, \dots, \tilde{\pi}_{H-1})$,

$$J_{h,s}(\theta_h) := E_{S_h=s}^{(\pi^{\theta_h}, \tilde{\pi}_{(h+1)})} \Big[ \sum_{k=h}^{H-1} r(S_k, A_k) \Big]. \tag{6}$$

An optimal parameter $\theta_h^*$ is then sought such that $J_{h,s}(\theta_h^*) = \sup_{\theta \in \Theta_h} J_{h,s}(\theta)$, for all $s \in \mathcal{S}_h$. In order to attain an optimal policy at each time point, this problem is approached via backward induction, and the parametrization $\tilde{\pi}$ in equation (6) is selected to be the pre-optimized one. Assuming that the parametrization is able to approximate an optimal policy (e.g. the softmax parametrization), then the backward induction yields optimal parameters $\theta_h^*, \dots, \theta_{H-1}^*$ in the sense that, see Puterman (2005, Sec. 4.5),

$$J_{h,s}(\theta_h^*) = \sup_{\theta_h \in \Theta_h, \dots, \theta_{H-1} \in \Theta_{H-1}} V_h^{\pi^\theta}(s),$$

for all $s \in \mathcal{S}_h$. To employ the policy gradient method, it is essential to compute the gradient of $J_{h,s}(\theta)$ with respect to $\theta$ for a given policy $\tilde{\pi}$. Notably, the forthcoming policy $\tilde{\pi}$ can be *any* policy, independent of the current parameter $\theta$, which is trained during epoch $h$. This approach significantly deviates from the one used in discounted MDPs, such as in Sutton et al. (1999), where a stationary policy is parametrized and utilized at every time step. Despite the differences, a policy gradient theorem can still be attained, allowing the gradient of the objective function to be written as an expectation.

**Theorem 2.2.** *For a fixed policy $\tilde{\pi}$ and $h \in \mathcal{H}$ the gradient of $J_{h,s}(\theta)$ defined in (6) is given by*

$$\nabla J_{h,s}(\theta) = \mathbb{E}_{S_h=s, A_h \sim \pi^\theta(\cdot|s)}[\nabla \log(\pi^\theta(A_h|S_h)) Q_h^{\tilde{\pi}}(S_h, A_h)].$$

As for the value function, we denote by $J_h(\theta) := \mathbb{E}_{s \sim \mu_h}[J_{h,s}(\theta)]$ the objective function under some initial state distribution $\mu_h$ on $\mathcal{S}_h$. Algorithm 1 summarizes policy gradient in finite-time MDPs.

---

**Algorithm 1:** Policy Gradient for finite-time MDPs and non-stationary policies

**Result:** Approximate policy $\hat{\pi}^* \approx \pi^*$
Initialize $\theta^{(0)} = (\theta_0^{(0)}, \dots, \theta_{H-1}^{(0)}) \in \Theta$
**for** $h = H-1, \dots, 0$ **do**
    Choose fixed step size $\eta_h$ and number of training steps $N_h$
    **for** $n = 0, \dots, N_h - 1$ **do**
        Calculate $\nabla J_h(\theta_h^{(n)})$ with fixed policy $\hat{\pi}^*$ after $h$
        $\theta_h^{(n+1)} = \theta_h^{(n)} + \eta_h \nabla J_h(\theta_h^{(n)})$
    **end**
    Set $\hat{\pi}_h^* = \pi^{\theta_h^{(N_h)}}$
**end**

---

Training each time point separately and having a fixed policy $\tilde{\pi}$ after $h$, we state a version of the performance difference lemma given this specific setting.

**Corollary 2.3.** *For any $h \in \mathcal{H}$ and two policies $\pi$ and $\pi'$: If $\pi_{(h+1)} = \pi'_{(h+1)}$, it holds that*

$$V_h^\pi(s) - V_h^{\pi'}(s) = \mathbb{E}_{S_h=s}^{\pi_{(h)}} \Big[ A_h^{\pi'}(S_h, A_h) \Big].$$

## 3 Convergence Analysis of Softmax Policy Gradient

Before we combine all decision epochs as stated in Algorithm 1, we provide convergence results for each $h \in \mathcal{H}$ given that the policy after $h$ is fixed and denoted by $\tilde{\pi}$. The error analysis over time is then employed in Section 5.

*Assumption* 3.1. Throughout the remaining manuscript we assume that the rewards are bounded in $[0, R^*]$, for some $R^* > 0$.

### 3.1 Asymptotic convergence

The choice of tabular softmax parametrization is particularly convenient as derivatives are simple.

**Lemma 3.2.** *Let $h \in \mathcal{H}$, then the partial derivatives of $J_h$ with respect to $\theta$ take the following form*

$$\frac{\partial J_h(\theta)}{\partial \theta(s, a)} = \mu(s) \pi^\theta(a|s) A_h^{(\pi^\theta, \tilde{\pi}^{(h+1)})}(s, a).$$

Furthermore, $J_h$ is a smooth function with respect to $\theta$. The proof is based on a more general result which proves smoothness for all parametrizations with bounded gradient and Hessian of the log-policy.

**Proposition 3.3.** *Let $h \in \mathcal{H}$ and consider the objective function $J_h(\theta)$. If there exists $G, M > 0$ such that*

$$||\nabla \log \pi^\theta(a|s)||_2 \leq G \quad and \quad ||\nabla^2 \log \pi^\theta(a|s)||_2 \leq M,$$

*for all $s \in \mathcal{S}_h$, $a \in \mathcal{A}_s$, then for any initial state distribution $\mu_h$ of $S_h$ the function $J_h(\theta)$ is $\beta_h$-smooth in $\theta$ with $\beta_h = (H - h)R^*(G^2 + M)$.*

Smoothness under these assumptions in the discounted finite-time setting with stationary policy was shown for example in Xu et al. (2020b) and Xu et al. (2020a). We obtain the following smoothness parameter:

**Lemma 3.4.** *Let $h \in \mathcal{H}$, then the $h$-state value function under softmax parametrization, $\theta \mapsto J_h(\theta)$, is $\beta_h$-smooth with $\beta_h = 2(H - h)R^*|\mathcal{A}|$.*

We point out that the smoothness parameter is independent of the choice of $\tilde{\pi}$. A consequence of the smoothness is the asymptotic convergence of the objective function towards a global maximum. As each epoch is considered separately we just write $\theta_n$ instead of $\theta_h^{(n)}$ until Section 5.

**Theorem 3.5.** *Let $h \in \mathcal{H}$ and consider the gradient ascent updates*

$$\theta_{n+1} = \theta_n + \eta_h \nabla J_h(\theta_n) \tag{7}$$

*for arbitrary $\theta_0 \in \mathbb{R}^{d_h}$. We assume that $\mu_h(s) > 0$ for all $s \in \mathcal{S}_h$ and $0 < \eta_h \leq \frac{1}{\beta_h}$. Then, for all $s \in \mathcal{S}_h$, $J_{h,s}(\theta_n)$ converges to $J_{h,s}^*$ for $n \to \infty$, where $J_{h,s}^* = \sup_\theta J_{h,s}(\theta) < \infty$.*

The difficulties that arise from softmax parametrization are the same as discussed in Agarwal et al. (2021) for the infinite time setting: The softmax policy approximates an optimal deterministic policy. Therefore, parameters converge to $-\infty$ for suboptimal actions and to $\infty$ for optimal actions. The idea of the proof follows the outline of the discounted MDP setting except for one main distinction: the action-value function $Q_h$ is independent of the policy gradient updates such that no limiting process has to be constructed. A detailed proof is provided in B.1.

Note that the assumption $\mu_h(s) > 0$ for all $s \in \mathcal{S}_h$ is necessary for sufficient exploration. The same assumption is needed for the initial distribution of a discounted MDP in Agarwal et al. (2021, Thm. 10). Furthermore, Mei et al. (2020, Prop. 3) have demonstrated the necessity of this assumption.

### 3.2 Convergence rate

In order to derive a convergence rate for tabular softmax parametrized finite-time MDPs we will establish a weaker form of the PL-inequality. Therefore, consider for $h \in \mathcal{H}$ a deterministic optimal policy $\pi_h^*$, given that the policy after $h$ is fixed by $\tilde{\pi}$, i.e. for all $s \in \mathcal{S}_h$,

$$\pi_h^*(\cdot|s) = \underset{\pi(\cdot|s): \text{Policy}}{\operatorname{argmax}} V_h^{(\pi, \tilde{\pi}^{(h+1)})}(s).$$

Please note here that the optimal policy and also $J_{h,s}^*$ depend on the choice of $\tilde{\pi}$.

**Lemma 3.6** (weak PL-inequality). *For the objective $J_h$ it holds that*

$$\|\nabla J_h(\theta)\|_2 \geq \min_{s \in \mathcal{S}_h} \pi^\theta(a_h^*(s)|s)(J_h^* - J_h(\theta)),$$

*where $a_h^*(s) = argmax_{a \in \mathcal{A}_s} \pi_h^*(a|s)$ and $J_h^* = \sup_\theta J_h(\theta)$.*

The term $\min_{s \in \mathcal{S}} \pi^\theta(a_h^*(s)|s)$ also appears in similar form in the discounted setting in Mei et al. (2020). The main challenge is to bound this term from below uniformly in $\theta$ appearing in the gradient ascent updates. Due to asymptotic convergence this can be achieved, where it is necessary to assume $\mu_h(s) > 0$ for all $s \in \mathcal{S}_h$.

**Lemma 3.7.** *Let $h \in \mathcal{H}$, $\mu_h(s) > 0$ for all $s \in \mathcal{S}_h$ and consider the sequence $(\theta_n)_{n \in \mathbb{N}_0}$ generated by (7) for arbitrarily initialized $\theta_0 \in \mathbb{R}^{d_h}$. Then it holds that $c_h := \inf_{n \geq 0} \min_{s \in \mathcal{S}_h} \pi^{\theta_n}(a_h^*(s)|s) > 0$.*

We emphasize that the constant $c_h$ is influenced by the initial parameter $\theta_0$ thereby making it a parameter dependent on the model, as it is also for discounted MDPs in Mei et al. (2020).

**Theorem 3.8.** *Let $h \in \mathcal{H}$, $\mu_h(s) > 0$ for all $s \in \mathcal{S}_h$ and consider the sequence $(\theta_n)_{n \in \mathbb{N}_0}$ generated by (7) for arbitrarily initialized $\theta_0 \in \mathbb{R}^{d_h}$. Define $c_h := \inf_{n \geq 0} \min_{s \in \mathcal{S}_h} \pi^{\theta_n}(a_h^*(s)|s) > 0$ by Lemma 3.7 and choose step size $\eta_h = \frac{1}{\beta_h}$ with $\beta_h = 2(H - h)R^*|\mathcal{A}|$. Then it holds that*

$$J_h^* - J_h(\theta_n) \leq \frac{4(H - h)R^*|\mathcal{A}|}{c_h^2 n},$$

*where $J_h^* = \sup_\theta J_h(\theta)$.*

The error bound depends on the time horizon up to the last time point, meaning intuitively that an optimal policy for earlier time points in the MDP (smaller $h$) is harder to achieve and requires a longer learning period then later time points ($h$ near to $H$). Comparing this result to the convergence rate for discounted MPDs we note that the linear dependency on the time horizon is less aggressive than the factor $(1 - \gamma)^{-1}$. In addition, the magnitude of the state space $\mathcal{S}_h$ does not have a direct impact on the rate. However, the constant $c_h$ indirectly introduces a dependency.

## 4 Convergence Analysis of Stochastic Softmax Policy Gradient

For the rest of this paper we drop the assumption of knowing $\nabla J_h(\theta)$. In this model-free setting it is only assumed that trajectories of the finite-time MDP can be simulated. Stochastic policy gradient is used to train the parameters, where in each iteration the gradient of the objective is approximated using Monte Carlo estimates. Consider $K_h$ trajectories $(s_k^i, a_k^i)_{k=h}^{H-1}$, for $i = 1, \ldots, K_h$, generated by $s_h^i \sim \mu_h$, $a_h^i \sim \pi_h^\theta$ and $a_k^i \sim \tilde{\pi}_k$ for $h < k < H$. The estimator is defined by

$$\widehat{\nabla} J_h^K(\theta) = \frac{1}{K_h} \sum_{i=1}^{K_h} \nabla \log(\pi^\theta(a_h^i|s_h^i)) \hat{Q}_h(s_h^i, a_h^i), \tag{8}$$

where $\hat{Q}_h(s_h^i, a_h^i) = \sum_{k=h}^{H-1} r(s_k^i, a_k^i)$ is an unbiased estimator of the $h$-state-action value function in $(s_h^i, a_h^i)$ under policy $\tilde{\pi}$. Then the stochastic policy gradient updates for training the parameter $\theta$ are given by

$$\theta_{n+1} = \theta_n + \eta_h \widehat{\nabla} J_h^{K_h}(\theta). \tag{9}$$

To train an optimal policy with backward induction, $\tilde{\pi}$ is chosen to be the already trained policies. As in Section 3 we first restrict our convergence analysis to one time point $h$ given a fixed policy $\tilde{\pi}$ after $h$. The entire stochastic policy gradient algorithm, often called REINFORCE, is summarized in Algorithm 2.

Under the softmax parametrization it holds true that $\widehat{\nabla} J_h^{K_h}(\theta)$ is an unbiased estimator with uniformly bounded variance due to the bounded reward assumption (see Lemma C.1).

### 4.1 Asymptotic convergence to stationary point

Using stochastic policy gradient, we obtain almost sure convergence of the value function to a stationary point for decreasing step sizes. Note that, except for this theorem we assume a constant step size.

**Algorithm 2:** REINFORCE with Backward Iteration

---

**Result:** Approximate policy $\hat{\pi}^* \approx \pi^*$

Initialize $\theta^{(0)} = (\theta_0^{(0)}, \ldots, \theta_{H-1}^{(0)}) \in \Theta$

**for** $h = H - 1, \ldots, 0$ **do**

     Choose step size $\eta_h$, number of training steps $N_h$ and batch size $K_h$

     **for** $n = 0, \ldots, N_h - 1$ **do**

         **for** $i = 1, \ldots K_h$ **do**

             Sample trajectory $(s_k^i, a_k^i)_{k=h}^{H-1}$, s.t. $s_h^i \sim \mu_h$, $a_h^i \sim \pi_h^{\theta_h^{(n)}}$ and $a_k^i \sim \hat{\pi}_k^*$ for $k > h$

         **end**

         $\theta_h^{(n+1)} = \theta_h^{(n)} + \eta_h \widehat{\nabla} J_h^{K_h}(\theta)$, where $\widehat{\nabla} J_h^{K_h}(\theta)$ is defined in (8)

     **end**

     Set $\hat{\pi}_h^* := \pi_h^{\theta_h^{(N_h)}}$

**end**

---

**Theorem 4.1.** *For any $h \in \mathcal{H}$ consider the stochastic process $(\theta_n)_{n \geq 0}$ generated by*

$$\theta_{n+1} = \theta_n + \eta_h^{(n)} \widehat{\nabla} J_h^{K_h}(\theta),$$

*for arbitrary batch size $K_h \geq 1$ and initial $\theta_0$ such that $\mathbb{E}[J_h(\theta_0)] < \infty$. Furthermore, suppose that $\eta_h^{(n)}$ is decreasing, such that $\sum_{n \geq 0} \eta_h^{(n)} = \infty$ and $\sum_{n \geq 0} \left(\eta_h^{(n)}\right)^2 < \infty$. Then $\nabla J_h(\theta_n) \to 0$ almost surely for $n \to \infty$.*

With Lemma C.1 and the boundedness of the $h$-state value functions, this follows directly from the stochastic approximation theorem stated in Bertsekas and Tsitsiklis (2000) (see Proposition C.2 in the supplementary material).

## 4.2 Complexity bounds to approximate to global optimum with high probability

In the following denote by $(\bar{\theta}_n)_{n \geq 1}$ the deterministic sequence generated by policy gradient with exact gradients,

$$\bar{\theta}_{n+1} = \bar{\theta}_n + \eta_h \nabla J_h(\bar{\theta}_n). \tag{10}$$

Let $(\theta_n)_{n \geq 0}$ be the stochastic process from (9) such that the initial parameter agree, $\theta_0 = \bar{\theta}_0$, and the step size $\eta_h$ is the same for both processes. The natural filtration of $(\theta_n)_{n \geq 0}$ is denoted by $(\mathcal{F}_n)_{n \geq 0}$. Recall that $c_h = \min_{n \geq 0} \min_{s \in \mathcal{S}} \pi^{\bar{\theta}_n}(a^*(s)|s)$ is bounded away from 0 by Lemma 3.7. The idea of the convergence analysis for stochastic softmax policy gradient is to define the following stopping time

$$\tau := \min\{n \geq 0 : \|\theta_n - \bar{\theta}_n\|_2 \geq \frac{c_h}{4}\}.$$

This means, $\tau$ is the first time when the stochastic process $(\theta_n)_{n \geq 0}$ is *too far away* from the policy gradient trajectory $(\bar{\theta}_n)_{n \geq 0}$. Hence, all challenges encountered in the deterministic case transfer to the stochastic context, indicating that the model dependent constant $c_h$ naturally appears in the error bounds of the stochastic case. We emphasize that $\tau$ is a stopping time with respect to the filtration $(\mathcal{F}_n)_{n \geq 0}$ by construction.

First, consider the event $\{n \leq \tau\}$, i.e. $\|\theta_n - \bar{\theta}_n\|_2 \leq \frac{c_h}{4}$. It follows by the $\sqrt{2}$-Lipschitz continuity of $\theta \mapsto \pi^\theta(a^*(s)|s)$ (Lemma C.3) that $\min_{0 \leq k \leq \tau} \min_{s \in \mathcal{S}} \pi^{\theta_k}(a^*(s)|s) \geq \frac{c_h}{2} > 0$ (Lemma C.4). This allows us to use the weak PL-inequality of Lemma 3.6 to derive a convergence rate on the event $\{n \leq \tau\}$ in the following sense:

**Lemma 4.2.** *Suppose $\mu_h(s) > 0$ for all $s \in \mathcal{S}_h$, the batch size $K_h^{(n)} \geq \frac{9 c_h^2 C_h}{32 \beta_h^2 N_h^{\frac{3}{2}}} \left(1 - \frac{1}{2\sqrt{N_h}}\right) n^2$ is increasing for some $N_h \geq 1$ and the step size $\eta_h = \frac{1}{\beta_h \sqrt{N_h}}$, for fixed $h \in \mathcal{H}$. Then,*

$$\mathbb{E}\left[(J_h^* - J_h(\theta_n))\mathbf{1}_{\{n \leq \tau\}}\right] \leq \frac{16\sqrt{N_h}\beta_h}{3(1 - \frac{1}{2\sqrt{N_h}})c_h^2 n}.$$

Secondly, consider the complementary event $\{\tau \leq n\}$. We can bound the probability of this event by $\delta$ for a large enough batch size $K_h$. The proof is based on a similar result obtained by Ding et al. (2022, Lem. 6.3) for discounted MDPs.

**Lemma 4.3.** *Suppose $\mu_h(s) > 0$ for all $s \in \mathcal{S}_h$. Then, for any $\delta > 0$, we have $\mathbb{P}(\tau \leq n) < \delta$ if $K_h \geq \frac{16n^3 C_h}{\beta^2 c_h^2 \delta^2}$ and $\eta_h = \frac{1}{\sqrt{n}\beta_h}$.*

We are now ready to formulate the main result of this section.

**Theorem 4.4.** *Suppose the stochastic policy gradient updates are generated by* (9) *for arbitrary initialization $\theta_0 \in \mathbb{R}^{d_h}$. Suppose that $\mu_h(s) > 0$ for all $s \in \mathcal{S}_h$ and choose for any $\delta, \epsilon > 0$,*

    *(i) the number of training steps $N_h \geq \left( \frac{64\beta_h}{3\delta c_h^2 \epsilon} \right)^2$,*

    *(ii) the step size $\eta_h = \frac{1}{\beta_h \sqrt{N_h}}$ and the batch size $K_h = \frac{64N_h^3 C_h}{\beta^2 c_h^2 \delta^2}$.*

*Then, $\mathbb{P}\big( (J_h^* - J_h(\theta_{N_h})) \geq \epsilon \big) \leq \delta$.*

It should be noted that the choice of step size $\eta_h$ and batch size $K_h$ are closely connected and both strongly depend on the number of training steps $N_h$. Specifically, as $N_h$ increases, the batch size increases, while the step size tends to decrease to prevent exceeding the stopping time with high probability. However, it is possible to increase the batch size even further and simultaneously benefit from choosing a larger step size, or vice versa.

# 5  Error Analysis over Time

In this section, we will first examine the accumulation of error over time for the policy gradient Algorithm 1, and secondly, for the stochastic policy gradient Algorithm 2. In both cases the error accumulates linearly such that an $\frac{\epsilon}{H}$-error in each time point $h$ results in an overall error of $\epsilon$. This is due to the additive structure of the rewards and comes naturally from the backward induction of dynamic programming for finite-time MDPs.

**Theorem 5.1.** *Assume that $\mu_h(s) > 0$ for all $h \in \mathcal{H}$, $s \in \mathcal{S}_h$. Let $\epsilon > 0$, the step size $\eta_h = \frac{1}{\beta_h}$ and the batch size $N_h = \frac{4(H-h)HR^*|\mathcal{A}|}{c_h^2 \epsilon} \left\| \frac{1}{\mu_h} \right\|_\infty$. Denote by $\hat{\pi}^* = (\pi^{\theta_0^{N_0}}, \dots, \pi^{\theta_{H-1}^{N_{H-1}}})$ the final policy from Algorithm 1, then for all $s \in \mathcal{S}_0$,*

$$V_0^*(s) - V_0^{\hat{\pi}^*}(s) \leq \epsilon.$$

For the stochastic policy gradient algorithm, we obtain the following main result:

**Theorem 5.2.** *Assume that $\mu_h(s) > 0$ for all $h \in \mathcal{H}$, $s \in \mathcal{S}_h$. Let $\delta, \epsilon > 0$, the step size $\eta_h = \frac{1}{\beta_h N_h}$, number of training steps $N_h = \left( \frac{64\beta_h H^2 \left\| \frac{1}{\mu_h} \right\|_\infty}{3\delta c_h^2 \epsilon} \right)^2$ and the batch size $K_h = \frac{64N_h^2 H^2 C_h}{\beta_h c_h^2 \delta^2}$. Denote by $\hat{\pi}^* = (\pi^{\theta_0^{N_0}}, \dots, \pi^{\theta_{H-1}^{N_{H-1}}})$ the final policy from Algorithm 2, then*

$$\mathbb{P}\Big( \exists s \in \mathcal{S}_0 : V_0^*(s) - V_0^{\hat{\pi}^*}(s) \geq \epsilon \Big) \leq \delta.$$

In both results we observe that the number of training steps in each epoch depends on the constant $\left\| \frac{1}{\mu_h} \right\|_\infty = \max_{s \in \mathcal{S}} \frac{1}{\mu_h(s)}$. The proofs of Section D reveal that this constant occurs to ensure that the objective $J_{h,s}(\theta_h^{(N_h)})$ is close to $J_{h,s}^*$ for every $s \in \mathcal{S}_h$.

# 6  Convergence Analysis of Stochastic Policy Gradient in Infinite Horizons

In this final section, we show how to combine the results of Mei et al. (2020) with our stochastic gradient arguments to show that the plain vanilla REINFORCE algorithm without regularization can approximate global maxima if the batch sizes are chosen properly. Our theoretically derived batch sizes are clearly not of practical use but give a first insight why REINFORCE requires large

batch sizes to give reasonable approximations. In the following, we consider the discounted MDP setting from Equation (1) with rewards taking values in $[0, 1]$, i.e. $R^* = 1$, and tabular softmax parametrization $\pi^\theta$ from (5) with $\theta \in \Theta = \mathbb{R}^{|\mathcal{S}||\mathcal{A}|}$. The objective function $J(\theta) := \mathbb{E}_{S_0 \sim \mu}[V^{\pi^\theta}(S_0)]$ is defined for an initial state distribution $\mu$. It is important to highlight that $\pi^\theta$ is now a stationary policy used in every epoch. Our arguments rely on the weak PL-inequality for the exact value function. Mei et al. (2020) proved that

$$\left\|\frac{\partial V^{\pi^\theta}(\mu)}{\partial \theta}\right\|_2 \geq \left\|\frac{d_\rho^{\pi^*}}{d_\mu^{\pi^\theta}}\right\|_\infty \frac{\min_{s \in \mathcal{S}} \pi^\theta(a^*(s)|s)}{\sqrt{|\mathcal{S}|}}(V^*(\rho) - V^{\pi^\theta}(\rho)),$$

where $a^*(s) = \operatorname{argmax} \pi^*(\cdot|s)$ the optimal action in state $s$ and $\left\|\frac{d_\rho^{\pi^*}}{d_\mu^{\pi^\theta}}\right\|_\infty$ is the distribution mismatch coefficient introduced in Agarwal et al. (2021). We present an alternative version in Lemma E.2 without the constant $|\mathcal{S}|^{-1/2}$. The typical approach to prove convergence of stochastic gradient schemes is to iteratively compare the stochastic gradient update to the deterministic one and then control the error. This is not always possible, but for stochastic softmax policy gradient we show that the error can be controlled for large enough batch sizes. We proceed in a manner similar to Section 4.2. Thus, to state the theorem let us denote by

$$\bar{\theta}_{n+1} = \bar{\theta}_n + \eta \nabla J(\bar{\theta}_n), \quad \theta_{n+1} = \theta_n + \eta \widehat{\nabla} J^K(\theta) \tag{11}$$

the policy gradient and stochastic policy gradient schemes. Also denote by $c := \min_{n \geq 0} \min_{s \in \mathcal{S}} \pi^{\bar{\theta}_n}(a^*(s)|s)$ the model dependent constant from the weak PL-inequality of (Mei et al., 2020, Lem. 8). For the algorithm we use the unbiased gradient estimator proposed by Zhang et al. (2020) which the authors used to prove convergence to a stationary point. Our main contribution is the following convergence result towards the global optimum:

**Theorem 6.1.** *Let* $(\bar{\theta}_n)_{n \geq 0}$ *and* $(\theta_n)_{n \geq 0}$ *be the (stochastic) policy gradient updates from* (11) *for arbitrary initial* $\bar{\theta}_0 = \theta_0 \in \Theta$. *Suppose* $\mu(s) > 0$ *for all* $s \in \mathcal{S}$ *and choose for any* $\delta, \epsilon > 0$,

  *(i) the number of training steps* $N \geq \left(\frac{258}{3\epsilon\delta c^2(1-\gamma)^3}\right)^2$,

  *(ii) step size* $\eta = \frac{(1-\gamma)^3}{8\sqrt{N}}$

  *(iii) batch size* $K = \max\left\{\frac{9(1-\gamma)^4 c^2 C}{2048}(\sqrt{N} - \frac{1}{2})\left\|\frac{d_\mu^{\pi^*}}{\mu}\right\|_\infty^{-2}, \frac{4(1-\gamma)^6 N^3 C}{c^2\delta^2}\right\}$.

*Then,* $\mathbb{P}\big((J^* - J(\theta_N)) \geq \epsilon\big) \leq \delta$, *where* $J^* = \sup_\theta J(\theta)$.

We present more details on the algorithm and the proof in Section E of the supplementary material. We emphasize that the dependency on the distribution mismatch coefficient and the model dependent constant $c$ are unavoidable since the stochastic gradient ascent is derived from the deterministic gradient ascent. To the best of our knowledge, this is the first convergence analysis for stochastic policy gradient with softmax parametrization without regularization. So far, Ding et al. (2022) derived complexity bounds for convergence of softmax policy gradient to the entropy-regularized optimum.

# 7 Conclusion and Future Work

In this paper, we have presented a convergence analysis of policy gradient methods for undiscounted MDPs with finite-time horizon in the tabular setting. Assuming exact gradients we have obtained an $\mathcal{O}(1/n)$-convergence rate which is linearly dependent on the time horizon. In the model-free setting we have derived complexity bounds to approximate the error to global optima with high probability. Moreover, we were able to extend this result to discounted MDPs without regularization.

In the finite-time case, it would be intriguing to explore policy parametrizations with a smaller parameter space as for example log-linear policies. Additionally, investigating modern policy gradient algorithms such as TRPO and natural policy gradient within the context of finite-time MDPs could further enhance the convergence rate. In the stochastic setting, it is desirable to eliminate the model-dependent parameter from the complexity bounds to construct a practicable algorithm. This would require an improved convergence analysis of policy gradient with exact gradients.

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
