# Supplementary Material

 Subsequently, we provide a complete collection of proofs for the stated results in the main body. We
 restate these results to enhance readability and ensure a clear understanding of the proof details.

## A    Proofs of Section 2

 **Lemma 2.1** (Performance difference lemma)**.** *For any $h \in \mathcal{H}$ and for any pair of policies $\pi$ and $\pi'$*
 *the following holds true for every $s \in \mathcal{S}_h$:*

$$V_h^\pi(s) - V_h^{\pi'}(s) = \sum_{k=h}^{H-1} \mathbb{E}_{S_h=s}^{\pi^{(h)}}\Big[ A_k^{\pi'}(S_k, A_k) \Big].$$

*Proof.*

$$
\begin{aligned}
V_h^\pi(s) - V_h^{\pi'}(s) &= \mathbb{E}_{S_h=s}^{\pi^{(h)}}\Big[ \sum_{k=h}^{H-1} r(S_k, A_k) \Big] - V_h^{\pi'}(s) \\
&= \mathbb{E}_{S_h=s}^{\pi^{(h)}}\Big[ \sum_{k=h}^{H-1} r(S_k, A_k) + \sum_{k=h}^{H-1} V_k^{\pi'}(S_k) - \sum_{k=h}^{H-1} V_k^{\pi'}(S_k) \Big] - V_h^{\pi'}(s) \\
&= \mathbb{E}_{S_h=s}^{\pi^{(h)}}\Big[ \sum_{k=h}^{H-1} r(S_k, A_k) + \sum_{k=h+1}^{H-1} V_k^{\pi'}(S_k) - \sum_{k=h}^{H-1} V_k^{\pi'}(S_k) \Big] \\
&= \mathbb{E}_{S_h=s}^{\pi^{(h)}}\Big[ \sum_{k=h}^{H-1} r(S_k, A_k) + \sum_{k=h}^{H-2} V_{k+1}^{\pi'}(S_{k+1}) - \sum_{k=h}^{H-1} V_k^{\pi'}(S_k) \Big] \\
&= \mathbb{E}_{S_h=s}^{\pi^{(h)}}\Big[ \sum_{k=h}^{H-1} \big( r(S_k, A_k) + V_{k+1}^{\pi'}(S_{k+1}) - V_k^{\pi'}(S_k) \big) \Big] \\
&= \mathbb{E}_{S_h=s}^{\pi^{(h)}}\Big[ \sum_{k=h}^{H-1} A_k^{\pi'}(S_k, A_k) \Big] \\
&= \sum_{k=h}^{H-1} \mathbb{E}_{S_h=s}^{\pi^{(h)}}\Big[ A_k^{\pi'}(S_k, A_k) \Big],
\end{aligned}
$$

 where we have used that $r(S_k, A_k) + V_{k+1}^{\pi'}(S_{k+1}) = Q_k^{\pi'}(S_k, A_k)$. In the fifth equation we used the
 notation $V_H \equiv 0$ and note that $Q_{H-1} \equiv r$ independent of any policy. $\qquad\square$

 Unless explicitly specified, all differentiations are performed with respect to the variable $\theta$.
 **Theorem 2.2.** *For a fixed policy $\tilde{\pi}$ and $h \in \mathcal{H}$ the gradient of $J_{h,s}(\theta)$ defined in (6) is given by*

$$\nabla J_{h,s}(\theta) = \mathbb{E}_{S_h=s, A_h \sim \pi^\theta(\cdot|s)}[\nabla \log(\pi^\theta(A_h|S_h)) Q_h^{\tilde{\pi}}(S_h, A_h)].$$

 *Proof.* The probability of a trajectory $w = (s_h, a_h, \ldots, s_{H-1}, a_{H-1})$ under the policy
 $(\pi^\theta, \tilde{\pi}_{(h+1)}) = (\pi^\theta, \tilde{\pi}_{h+1}, \ldots, \tilde{\pi}_{H-1})$ and initial state distribution $\delta_s$ is given by

$$\mathbb{P}_s^{(\pi^\theta, \tilde{\pi}_{(h+1)})}(w) = \delta_s(s_h)\pi^\theta(a_h|s_h) \prod_{k=h+1}^{H-1} p(s_k|s_{k-1}, a_{k-1})\tilde{\pi}_k(a_k|s_k).$$

 Then,

$$
\begin{aligned}
\nabla \log(\mathbb{P}_s^{(\pi^\theta, \tilde{\pi}_{(h+1)})}(w)) = \nabla \Big( &\log(\delta_s(s_h)) + \log(\pi^\theta(a_h|s_h)) \\
&+ \sum_{k=h+1}^{H-1} \log(p(s_k|s_{k-1}, a_{k-1})) + \log(\tilde{\pi}_k(a_k|s_k)) \Big) \\
= &\nabla \log(\pi^\theta(a_h|s_h)),
\end{aligned}
$$

which is known as the log-trick. Let $\mathcal{W}$ be the set of all trajectories from $h$ to $H-1$. Note that $\mathcal{W}$ is finite due to the assumption that state and action space is finite. Then for $s \in \mathcal{S}_h$

$$
\begin{aligned}
\nabla J_{h,s}(\theta) &= \nabla \sum_{w \in \mathcal{W}} \mathbb{P}_s^{(\pi^\theta, \tilde{\pi}_{(h+1)})}(w) \sum_{k=h}^{H-1} r(s_k, a_k) \\
&= \sum_{w \in \mathcal{W}} \mathbb{P}_s^{(\pi^\theta, \tilde{\pi}_{(h+1)})}(w) \nabla \log(\mathbb{P}_s^{(\pi^\theta, \tilde{\pi}_{(h+1)})}) \sum_{k=h}^{H-1} r(s_k, a_k) \\
&= \sum_{w \in \mathcal{W}} \mathbb{P}_s^{(\pi^\theta, \tilde{\pi}_{(h+1)})}(w) \nabla \log(\pi^\theta(a_h|s_h)) \sum_{k=h}^{H-1} r(s_k, a_k) \\
&= \mathbb{E}_{S_h=s}^{(\pi^\theta, \tilde{\pi}_{(h+1)})} \Big[ \nabla \log(\pi^\theta(A_h|S_h)) \sum_{k=h}^{H-1} r(S_k, A_k) \Big] \\
&= \mathbb{E}_{S_h=s}^{(\pi^\theta, \tilde{\pi}_{(h+1)})} \Big[ \nabla \log(\pi^\theta(A_h|S_h)) \mathbb{E}_{S_h}^{\tilde{\pi}} \Big[ \sum_{k=h}^{H-1} r(S_k, A_k) | S_h, A_h \Big] \Big] \\
&= \mathbb{E}_{S_h=s, A_h \sim \pi^\theta(\cdot|s)} \Big[ \nabla \log(\pi^\theta(A_h|S_h)) \, Q_h^{\tilde{\pi}}(S_h, A_h) \Big].
\end{aligned}
$$

$\square$

**Corollary 2.3.** *For any $h \in \mathcal{H}$ and two policies $\pi$ and $\pi'$: If $\pi_{(h+1)} = \pi'_{(h+1)}$, it holds that*

$$
V_h^\pi(s) - V_h^{\pi'}(s) = \mathbb{E}_{S_h=s}^{\pi_{(h)}} \Big[ A_h^{\pi'}(S_h, A_h) \Big].
$$

*Proof.* Let $k > h$, then

$$
\begin{aligned}
&\mathbb{E}_{S_h=s}^{\pi_{(h)}} \Big[ A_k^{\pi'}(S_k, A_k) \Big] \\
&= \sum_{a \in \mathcal{A}} \pi_h(a|s) \sum_{s \in \mathcal{S}} p(s|s,a) \mathbb{E}_{S_{h+1}=s}^{\pi_{(h+1)}} \Big[ Q_k^{\pi'}(S_k, A_k) - V_k^{\pi'}(S_k) \Big] \\
&= \sum_{a \in \mathcal{A}} \pi_h(a|s) \sum_{s \in \mathcal{S}} p(s|s,a) \mathbb{E}_{S_{h+1}=s}^{\pi'_{(h+1)}} \Big[ Q_k^{\pi'}(S_k, A_k) - V_k^{\pi'}(S_k) \Big] \\
&= \sum_{a \in \mathcal{A}} \pi_h(a|s) \sum_{s \in \mathcal{S}} p(s|s,a) \Big( \mathbb{E}_{S_{h+1}=s}^{\pi'_{(h+1)}} \Big[ \mathbb{E}_{S_k}^{\pi'} [Q_k^{\pi'}(S_k, A_k)] \Big] - \mathbb{E}_{S_{h+1}=s}^{\pi'_{(h+1)}} \Big[ V_k^{\pi'}(S_k) \Big] \Big) \\
&= \sum_{a \in \mathcal{A}} \pi_h(a|s) \sum_{s \in \mathcal{S}} p(s|s,a) \Big( \mathbb{E}_{S_{h+1}=s}^{\pi'_{(h+1)}} \Big[ V_k^{\pi'}(S_k) \Big] - \mathbb{E}_{S_{h+1}=s}^{\pi'_{(h+1)}} \Big[ V_k^{\pi'}(S_k) \Big] \Big) \\
&= 0.
\end{aligned}
$$

The claim follows with Lemma 2.1. $\square$

# B Proofs of Section 3

## B.1 Proofs of Section 3.1

First, we compute the derivative of the softmax policy for every $s \in \mathcal{S}_h$ and $a \in \mathcal{A}_s$,

$$
\pi^\theta(a|s) = \frac{e^{\theta(s,a)}}{\sum_{a' \in \mathcal{A}} e^{\theta(s,a')}},
$$

with parameter $\theta \in \mathbb{R}^{d_h}$:

$$
\frac{\partial \log(\pi^\theta(a|s))}{\partial \theta(a', s')} = \mathbf{1}_{\{s=s'\}}(\mathbf{1}_{\{a=a'\}} - \pi^\theta(a'|s')).
$$

Hence,

$$
\nabla \log(\pi^\theta(a|s)) = \Big( \mathbf{1}_{\{s=s'\}}(\mathbf{1}_{\{a=a'\}} - \pi^\theta(a'|s')) \Big)_{s' \in \mathcal{S}_h, a' \in \mathcal{A}_{s'}} \quad \in \mathbb{R}^{d_h}.
$$

**Lemma 3.2.** *Let $h \in \mathcal{H}$, then the partial derivatives of $J_h$ with respect to $\theta$ take the following form*

$$\frac{\partial J_h(\theta)}{\partial \theta(s,a)} = \mu(s)\pi^\theta(a|s)A_h^{(\pi^\theta,\tilde{\pi}_{(h+1)})}(s,a).$$

*Proof.* By the policy gradient Theorem 2.2,

$$\begin{aligned}
\nabla J_h(\theta) &= \nabla\mathbb{E}_{s\sim\mu}[J_{h,s}(\theta)] \\
&= \sum_{s\in\mathcal{S}}\mu(s)\nabla J_{h,s}(\theta) \\
&= \sum_{s\in\mathcal{S}}\mu(s)\mathbb{E}_{S_h=s,A_h\sim\pi^\theta(\cdot|s)}[\nabla\log(\pi^\theta(A_h|S_h))Q_h^{\tilde{\pi}}(S_h,A_h)].
\end{aligned}$$

Next we plug in the derivative of the softmax parametrization and obtain

$$\begin{aligned}
&\nabla J_h(\theta) \\
&= \sum_{s\in\mathcal{S}}\mu(s)\mathbb{E}_{S_h=s,A_h\sim\pi^\theta(\cdot|s)}\left[\left(\mathbf{1}_{\{S_h=s'\}}(\mathbf{1}_{\{A_h=a'\}}-\pi^\theta(a'|s'))\right)_{s'\in\mathcal{S}_h,a'\in\mathcal{A}_{s'}}Q_h^{\tilde{\pi}}(S_h,A_h)\right] \\
&= \left(\sum_{s\in\mathcal{S}}\mu(s)\sum_{a\in\mathcal{A}_s}\pi^\theta(a|s)\mathbf{1}_{\{s=s'\}}(\mathbf{1}_{\{a=a'\}}-\pi^\theta(a'|s'))Q_h^{\tilde{\pi}}(s,a)\right)_{s'\in\mathcal{S}_h,a'\in\mathcal{A}_{s'}} \\
&= \left(\mu(s')\pi^\theta(a'|s')Q_h^{\tilde{\pi}}(s',a')-\mu(s')\pi^\theta(a'|s')\sum_{a\in\mathcal{A}_s}\pi^\theta(a|s')Q_h^{\tilde{\pi}}(s',a)\right)_{s'\in\mathcal{S}_h,a'\in\mathcal{A}_{s'}} \\
&= \left(\mu(s')\pi^\theta(a'|s')(Q_h^{\tilde{\pi}}(s',a')-V_h^{(\pi^\theta,\tilde{\pi}_{(h+1)})}(s'))\right)_{s'\in\mathcal{S}_h,a'\in\mathcal{A}_{s'}} \\
&= \left(\mu(s')\pi^\theta(a'|s')A_h^{(\pi^\theta,\tilde{\pi}_{(h+1)})}(s',a')\right)_{s'\in\mathcal{S}_h,a'\in\mathcal{A}_{s'}},
\end{aligned}$$

where we used that $\sum_{a\in\mathcal{A}_s}\pi^\theta(a|s')Q_h^{\tilde{\pi}}(s',a) = J_{h,s'}(\theta) = V_h^{(\pi^\theta,\tilde{\pi}_{(h+1)})}(s')$. $\qquad\square$

**Proposition 3.3.** *Let $h \in \mathcal{H}$ and consider the objective function $J_h(\theta)$. If there exists $G, M > 0$ such that*

$$||\nabla\log\pi^\theta(a|s)||_2 \leq G \quad and \quad ||\nabla^2\log\pi^\theta(a|s)||_2 \leq M,$$

*for all $s \in \mathcal{S}_h$, $a \in \mathcal{A}_s$, then for any initial state distribution $\mu_h$ of $S_h$ the function $J_h(\theta)$ is $\beta_h$-smooth in $\theta$ with $\beta_h = (H-h)R^*(G^2+M)$.*

*Proof.* Define $\mathcal{W}$ as the set of all possible trajectories from $h$ to $H$ and consider $\hat{\pi}^\theta := (\pi^\theta, \tilde{\pi}_{(h+1)})$ as in the proof of Theorem 2.2. Fix any initial state distribution $\mu_h$ on $\mathcal{S}_h$, then the probability of $w$ is

$$p_{\mu_h}(w|\hat{\pi}^\theta) = \mu_h(s_h)\pi^\theta(a_h|s_h)\prod_{k=h+1}^{H-1}p(s_k|s_{k-1},a_{k-1})\tilde{\pi}(a_k|s_k).$$

It holds that

$$\nabla^2 J_h(\theta) = \sum_{w\in\mathcal{W}}\nabla^2 p_{\mu_h}(w|\hat{\pi}^\theta)\underbrace{\sum_{k=h}^{H-1}r(s_k,a_k)}_{:=r(w)}. \tag{12}$$

Now,

$$\begin{aligned}
\nabla^2\log\left(p_{\mu_h}(w|\hat{\pi}^\theta)\right) &= \nabla\left(p_{\mu_h}(w|\hat{\pi}^\theta)^{-1}\nabla p_{\mu_h}(w|\hat{\pi}^\theta)\right) \\
&= p_{\mu_h}(w|\hat{\pi}^\theta)^{-1}\nabla^2 p_{\mu_h}(w|\hat{\pi}^\theta) \\
&\quad - p_{\mu_h}(w|\hat{\pi}^\theta)^{-2}\nabla p_{\mu_h}(w|\hat{\pi}^\theta)\nabla p_{\mu_h}(w|\hat{\pi}^\theta)^T,
\end{aligned}$$

498 rearranging leads to

$$\nabla^2 p_{\mu_h}(w|\hat{\pi}^\theta) = p_{\mu_h}(w|\hat{\pi}^\theta)\Big(\nabla^2 \log\big(p_\mu(w|\hat{\pi}^\theta)\big) + p_{\mu_h}(w|\hat{\pi}^\theta)^{-2}\nabla p_{\mu_h}(w|\hat{\pi}^\theta)\nabla p_{\mu_h}(w|\hat{\pi}^\theta)^T\Big) \tag{13}$$

$$= p_{\mu_h}(w|\hat{\pi}^\theta)\Big(\nabla^2 \log\big(p_{\mu_h}(w|\hat{\pi}^\theta)\big) + \nabla \log(p_{\mu_h}(w|\hat{\pi}^\theta))\nabla \log(J_h(\theta))^T\Big). \tag{14}$$

499 Substitute (14) into (12):

$$\nabla^2 J_h(\theta)$$
$$= \sum_{w\in\mathcal{W}} p_{\mu_h}(w|\hat{\pi}^\theta)\Big(\nabla^2 \log\big(p_{\mu_h}(w|\hat{\pi}^\theta)\big) + \nabla \log(p_{\mu_h}(w|\hat{\pi}^\theta))\nabla \log(p_{\mu_h}(w|\hat{\pi}^\theta))^T\Big)r(w).$$

500 Using the log-trick similar to Theorem 2.2 yields

$$\nabla \log(p_{\mu_h}(w|\hat{\pi}^\theta)) = \nabla \log(\pi^\theta(a_h|s_h))$$

501 and

$$\nabla^2 \log(p_{\mu_h}(w|\hat{\pi}^\theta)) = \nabla^2 \log(\pi^\theta(a_h|s_h)).$$

502 Together with the assumption we made on the derivative and hessian of the log parametrized policy
503 we obtain

$$\| \nabla^2 J_h(\theta)\|_2$$
$$= \big\| \sum_{w\in\mathcal{W}} p_{\mu_h}(w|\hat{\pi}^\theta)\Big(\nabla^2 \log\big(p_{\mu_h}(w|\hat{\pi}^\theta)\big) + \nabla \log(p_{\mu_h}(w|\hat{\pi}^\theta))\nabla \log(p_{\mu_h}(w|\hat{\pi}^\theta))^T\Big)r(w)\big\|_2$$
$$\leq \sum_{w\in\mathcal{W}} p_{\mu_h}(w|\hat{\pi}^\theta)r(w)\Big(\|\nabla^2 \log(\pi^\theta(a_h|s_h))\|_2 + \|\nabla \log(\pi^\theta(a_h|s_h))\|_2^2\Big)$$
$$\leq \max_{w\in\mathcal{W}} r(w)(M + G^2)$$
$$\leq (H - h)R^*(M + G^2),$$

504 which completes the proof. Recall that $R^*$ is the maximal reward. $\qquad\square$

505 **Lemma 3.4.** *Let $h \in \mathcal{H}$, then the $h$-state value function under softmax parametrization, $\theta \mapsto J_h(\theta)$,*
506 *is $\beta_h$-smooth with $\beta_h = 2(H - h)R^*|\mathcal{A}|$.*

507 *Proof.* We use Proposition 3.3 for the softmax parametrization and see that

$$\|\nabla \log(\pi^\theta(a|s))\|_2 = \sqrt{\sum_{a'\in\mathcal{A}} \big(\mathbf{1}_{\{a'=a\}} - \pi^\theta(a'|s)\big)^2} \leq \sqrt{|\mathcal{A}_s|} \leq \sqrt{|\mathcal{A}|}$$

508 and (Frobenius norm)

$$\|\nabla^2 \log(\pi^\theta(a|s))\|_2 = \sqrt{\sum_{a^*\in\mathcal{A}_s}\sum_{a'\in\mathcal{A}_s} \big(\mathbf{1}_{\{a^*=a'\}}\pi^\theta(a'|s) - \pi^\theta(a^*|s)\pi^\theta(a'|s)\big)^2}$$
$$\leq \sqrt{|\mathcal{A}_s||\mathcal{A}_s|}$$
$$\leq |\mathcal{A}|.$$

509 Using Proposition 3.3 with $G = \sqrt{|\mathcal{A}|}$ and $M = |\mathcal{A}|$ yields the claim. $\qquad\square$

510 **Theorem 3.5.** *Let $h \in \mathcal{H}$ and consider the gradient ascent updates*

$$\theta_{n+1} = \theta_n + \eta_h \nabla J_h(\theta_n) \tag{7}$$

511 *for arbitrary $\theta_0 \in \mathbb{R}^{d_h}$. We assume that $\mu_h(s) > 0$ for all $s \in \mathcal{S}_h$ and $0 < \eta_h \leq \frac{1}{\beta_h}$. Then, for all*
512 *$s \in \mathcal{S}_h$, $J_{h,s}(\theta_n)$ converges to $J_{h,s}^*$ for $n \to \infty$, where $J_{h,s}^* = \sup_\theta J_{h,s}(\theta) < \infty$.*

The idea of the proof follows the line of arguments in Agarwal et al. (2021) for the asymptotic convergence of softmax policy gradient in the discounted stationary MDP setting. Thus, we first have to show a row of lemmata, compare to Lemma 41 to 51 in Agarwal et al. (2021).

**Lemma B.1** (Monotonicity). *If the learning rate satisfies $0 < \eta_h \leq \frac{1}{\beta_h} = \frac{1}{2(H-h)R^*|\mathcal{A}|}$ then $J_{h,s}(\theta_{n+1}) \geq J_{h,s}(\theta_n)$ for any $s \in \mathcal{S}_h$. Furthermore, for all $s \in \mathcal{S}_h$ there exists a limit $J_{h,s}^\infty$ such that*

$$\lim_{n \to \infty} J_{h,s}(\theta_n) = J_{h,s}^\infty < \infty.$$

*Proof.* By (Beck, 2017, Theorem 10.4) we have for any $\beta$-smooth function $f : \mathbb{R}^d \to \mathbb{R}$, that $(f(x^k))_{k \geq 0}$ is non-increasing sequence, when $x^{k+1} = x^k - \eta \nabla f(x^k)$ with $\eta_h \leq \frac{1}{\beta}$.

First note that $-J_{h,s}$ is also $\beta_h$-smooth. Then we have

$$\nabla J_h(\theta) = \nabla \Big( \sum_{s \in \mathcal{S}_h} \mu_h(s) J_{h,s}(\theta) \Big) = \sum_{s \in \mathcal{S}_h} \mu_h(s) \nabla J_{h,s}(\theta),$$

and $\frac{\partial J_{h,s}(\theta)}{\partial \theta(s',a)} = 0$ whenever $s' \neq s$. Denote by $\theta(s) = \theta(s, \cdot) \in \mathbb{R}^{|\mathcal{A}_s|}$, then

$$\theta(s)_{n+1} = \theta_n(s) + \eta_h \mu_h(s) \nabla J_{h,s}(\theta).$$

With the assumption $0 < \mu_h(s) \leq 1$ for all $s \in \mathcal{S}_h$ the first claim follows by (Beck, 2017, Theorem 10.4).

As $J_{h,s}(\theta_n) \leq (H - h)R^*$ is bounded for all $n \in \mathbb{N}$ the second claim follows directly from monotonicity. □

To save notation we fix an $h \in \mathcal{H}$. All results hold true for an arbitrary epoch. We introduce the following definitions without a subscript $h$:

$$\Delta = \min_{\{s,a \mid A_h^\infty(s,a) \neq 0\}} |A_h^\infty(s,a)|$$

where $A_h^\infty(s,a) = Q_h^{\tilde{\pi}}(s,a) - J_{h,s}^\infty$. Recall that $\tilde{\pi}$ is the fixed policy which we use for $h+1, \ldots, H-1$. For the rest of this section, we write $Q_h$ instead of $Q_h^{\tilde{\pi}}$. Further we denote by $A_h^{\theta_n}(s,a) := Q_h(s,a) - J_{h,s}(\theta_n)$, the advantage function with respect to parameter $\theta_n$.

We define the sets

$$I_0^s = \{a \in \mathcal{A}_s \mid Q_h(s,a) = J_{h,s}^\infty\},$$
$$I_+^s = \{a \in \mathcal{A}_s \mid Q_h(s,a) > J_{h,s}^\infty\},$$
$$I_-^s = \{a \in \mathcal{A}_s \mid Q_h(s,a) < J_{h,s}^\infty\}.$$

Note that we observe a fundamental difference to the proof of Agarwal et al. (2021) in the infinite time setting. We do not need a limit of the state-action value function $Q_h^\infty$, because $Q_h$ is independent of $\theta$ and only depends on $\tilde{\pi}$. We aim to prove that $I_+^s$ is an empty set, then $J_{h,s}^\infty = J_{h,a}^*$.

**Lemma B.2.** *There exists a time $N_1 > 0$ such that for all $n > N_1$, and $s \in \mathcal{S}_h$, we have*

$$A_h^{\theta_n}(s,a) < -\frac{\Delta}{4} \text{ for } a \in I_-^s; \quad A_h^{\theta_n}(s,a) > \frac{\Delta}{4} \text{ for } a \in I_+^s.$$

*Proof.* Fix $s \in \mathcal{S}_h$ arbitrarily. As $J_{h,s}(\theta_n) \to J_{h,a}^\infty$ for $n \to \infty$ and $\mathcal{S}_h$ is finite, we have that there exists $N_1 > 0$ such that for all $n > N_1$ and $s \in \mathcal{S}_h$,

$$J_{h,s}(\theta_n) > J_{h,s}^\infty - \frac{\Delta}{4}.$$

It follows for all $n > N_1$, $s \in \mathcal{S}_h$ and $a \in I_-^s$ by the definition of $\Delta$:

$$A_h^{\theta_n}(s,a) = Q_h(s,a) - J_{h,s}(\theta_n) \leq Q_h(s,a) - J_{h,s}^\infty + \frac{\Delta}{4} \leq -\Delta + \frac{\Delta}{4} < -\frac{\Delta}{4}.$$

Similarly, for all $n > N_1$, $s \in \mathcal{S}_h$ and $a \in I_+^s$ we obtain from monotonicity and the definition of $\Delta$,

$$A_h^{\theta_n}(s,a) = Q_h(s,a) - J_{h,s}(\theta_n) \geq Q_h(s,a) - J_{h,s}^\infty \geq \Delta > \frac{\Delta}{4}.$$

□

542 **Lemma B.3.** *It holds that $\frac{\partial J_h(\theta_n)}{\partial \theta_n(s,a)} \to 0$ as $n \to \infty$ for all $s \in \mathcal{S}_h$, $a \in \mathcal{A}_s$. This implies that for*
543 $a \in I_+^s \cup I_-^s$, $\pi^{\theta_n}(a|s) \to 0$ *and that* $\sum_{a \in I_0^s} \pi^{\theta_n}(a|s) \to 1$ *for* $n \to \infty$.

544 *Proof.* From (Beck, 2017, Theorem 10.15) we deduce for any $\beta$-smooth function $f : \mathbb{R}^d \to \mathbb{R}$,
545 that $\|\nabla f(x^k)\| \to 0$ for $k \to \infty$, if $x^{k+1} = x^k - \frac{1}{\beta}\nabla f(x^k)$. By Lemma 3.4 $J_h(\cdot)$ is $\beta_h$-smooth.
546 It follows by our choice of $\eta_h < \frac{1}{\beta_h}$ that $\frac{\partial J_h(\theta_n)}{\partial \theta_n(s,a)} \to 0$ as $n \to \infty$ for all $s \in \mathcal{S}_h$, $a \in \mathcal{A}_s$. Now
547 remember from Lemma 3.2

$$\frac{\partial J_h(\theta_n)}{\partial \theta_n(s,a)} = \mu_h(s)\pi^{\theta_n}(a|s)A_h^{\theta_n}(s,a),$$

548 and by Lemma B.2 $|A_h^{\theta_n}(s,a)| > \frac{\Delta}{4}$ for all $n > N_1$ and $a \in I_+^S \cup I_-^s$. As $\mu_h(s) > 0$ by assumption
549 it follows that $\pi^{\theta_n}(a|s) \to 0$ for $n \to \infty$ for all $a \in I_+^S \cup I_-^s$ from $\frac{\partial J_h(\theta_n)}{\partial \theta_n(s,a)} \to 0$ as $n \to \infty$.
550 The last claim, $\sum_{a \in I_0^s} \pi^{\theta_n}(a|s) \to 1$ for $n \to \infty$, follows immediately from $\sum_{a \in \mathcal{A}_s} \pi^{\theta_n}(a|s) = 1$
551 by:

$$\lim_{n \to \infty} \sum_{a \in I_0^s} \pi^{\theta_n}(a|s) = \lim_{n \to \infty} \left( \sum_{a \in \mathcal{A}_s} \pi^{\theta_n}(a|s) - \sum_{a \in I_+^S \cup I_-^s} \pi^{\theta_n}(a|s) \right)$$

$$= 1 - \sum_{a \in I_+^S \cup I_-^s} \lim_{n \to \infty} \pi^{\theta_n}(a|s)$$

$$= 1.$$

552 $\qquad\qquad\qquad\qquad\qquad\qquad\qquad\qquad\qquad\qquad\qquad\qquad\qquad\qquad\qquad\qquad\qquad\qquad\qquad\quad \square$

553 **Lemma B.4.** *For $a \in I_+^s$, the sequence $(\theta_n(s,a))_{n \geq 0}$ is strictly increasing for $n > N_1$ and for*
554 $a \in I_-^s$, *the sequence $(\theta_n(s,a))_{n \geq 0}$ is strictly decreasing for $n > N_1$.*

555 *Proof.* With Lemma B.2 we know that for $n > N_1$

$$A_h^{\theta_n}(s,a) > 0 \text{ for } a \in I_+^s; \quad A_h^{\theta_n}(s,a) < 0 \text{ for } a \in I_-^s,$$

556 and by Lemma 3.2

$$\frac{\partial J_h(\theta_n)}{\partial \theta_n(s,a)} = \mu_h(s)\pi^{\theta_n}(a|s)A_h^{\theta_n}(s,a).$$

557 As $\mu_h(s) > 0$ and $\pi^{\theta_n}(a|s) > 0$ by the definition of softmax parametrization, we have for all $n > N_1$

$$\frac{\partial J_h(\theta_n)}{\partial \theta_n(s,a)} > 0 \text{ for } a \in I_+^s; \quad \frac{\partial J_h(\theta_n)}{\partial \theta_n(s,a)} < 0 \text{ for } a \in I_-^s.$$

558 This implies for $a \in I_+^s$,

$$\theta_{n+1}(s,a) - \theta_n(s,a) = \eta_h\frac{\partial J_h(\theta_n)}{\partial \theta_n(s,a)} > 0,$$

559 i.e. $(\theta_n(s,a))_{n \geq 0}$ is strictly increasing for $n > N_1$ and similar for $a \in I_-^s$,

$$\theta_{n+1}(s,a) - \theta_n(s,a) = \eta_h\frac{\partial J_h(\theta_n)}{\partial \theta_n(s,a)} < 0,$$

560 i.e. $(\theta_n(s,a))_{n \geq 0}$ is strictly decreasing for $n > N_1$. $\qquad\qquad\qquad\qquad\qquad\quad \square$

561 **Lemma B.5.** *For all $s \in \mathcal{S}_h$ where $I_+^s \neq \emptyset$, we have that*

$$\max_{a \in I_0^s} \theta_n(s,a) \to \infty \quad \text{and} \quad \min_{a \in \mathcal{A}_s} \theta_n(s,a) \to -\infty \quad \text{for } n \to \infty.$$

*Proof.* By assumption $I_+^s \neq \emptyset$ there exists an $a_+ \in I_+^s$ and by Lemma B.3 we have $\pi^{\theta_n}(a_+|s) \to 0$, as $n \to \infty$. Hence, by softmax parametrization this is equivalent to

$$\frac{\exp(\theta_n(s, a_+))}{\sum\limits_{a \in \mathcal{A}_s} \exp(\theta_n(s, a))} \to 0, \text{ for } n \to \infty.$$

Using Lemma B.4, i.e. $\theta_n(s, a_+)$ is strictly increasing for $n > N_1$, we imply that $\exp(\theta_n(s, a_+))$ is strictly increasing for $n > N_1$. This implies that

$$\sum_{a \in \mathcal{A}_s} \exp(\theta_n(s, a)) \to \infty, \text{ for } n \to \infty.$$

Again by Lemma B.3 we know that

$$\sum_{a \in I_0^s} \pi^{\theta_n}(a|s) \to 1, \text{ for } n \to \infty,$$

i.e. by definition

$$\sum_{a \in I_0^s} \frac{\exp(\theta_n(s, a))}{\sum\limits_{a' \in \mathcal{A}_s} \exp(\theta_n(s, a'))} \to 1, \text{ for } n \to \infty.$$

As $\sum\limits_{a' \in \mathcal{A}_s} \exp(\theta_n(s, a')) \to \infty$ it follows that

$$\sum_{a \in I_0^s} \exp(\theta_n(s, a)) \to \infty, \text{ for } n \to \infty$$

implying

$$\max_{a \in I_0^s} \theta_n(s, a) \to \infty, \text{ for } n \to \infty.$$

For the second claim it holds that

$$\begin{aligned}
\sum_{a \in \mathcal{A}_s} \frac{\partial J_h(\theta_n)}{\partial \theta_n(s, a)} &= \mu_h(s) \sum_{a \in \mathcal{A}} \pi^{\theta_n}(a|s)(Q_h(s, a) - J_{h,s}(\theta_n)) \\
&= \mu_h(s)(\mathbb{E}_{S_h=s}^{\pi^{\theta_n}}[Q_h(S_h, A_h)] - J_{h,s}(\theta_n)) \\
&= \mu_h(s)(J_{h,s}(\theta_n) - J_{h,s}(\theta_n)) \\
&= 0.
\end{aligned}$$

By induction, we obtain $\sum_{a \in \mathcal{A}_s} \theta_n(s, a) = \sum_{a \in \mathcal{A}_s} \theta_0(s, a) := c$ for every $n > 0$ and hence

$$\min_{a \in \mathcal{A}_s} \theta_n(s, a) < \sum_{a \in \mathcal{A}_s} \theta_n(s, a) - \max_{a \in \mathcal{A}_s} \theta_n(s, a) = -\max_{a \in \mathcal{A}_s} \theta_n(s, a) + c.$$

Since $\max_{a \in \mathcal{A}_s} \theta_n(s, a) \to \infty$, because $\max_{a \in I_0^s} \theta_n(s, a) \to \infty$, we conclude $\min_{a \in \mathcal{A}_s} \theta_n(s, a) \to -\infty$ for $n \to \infty$. $\square$

**Lemma B.6.** *Suppose $a_+ \in I_+^s$. If there exists $a \in I_0^s$ such that for some $n > 0$, $\pi^{\theta_n}(a|s) \leq \pi^{\theta_n}(a_+|s)$, then for all $m > n$ it holds that $\pi^{\theta_m}(a|s) \leq \pi^{\theta_m}(a_+|s)$.*

*Proof.* Suppose there exists $a \in I_0^s$ such that for an $n > 0$, $\pi^{\theta_n}(a|s) \leq \pi^{\theta_n}(a_+|s)$. We show that $\pi^{\theta_{n+1}}(a|s) \leq \pi^{\theta_{n+1}}(a_+|s)$, then the claim follows by induction. We have

$$\begin{aligned}
\frac{\partial J_h(\theta_n)}{\partial \theta_n(s, a)} &= \mu_h(s)\pi^{\theta_n}(a|s)(Q_h(s, a) - J_{h,s(\theta_n)}) \\
&\leq \mu_h(s)\pi^{\theta_n}(a_+|s)(Q_h(s, a_+) - J_{h,s}(\theta_n)) \\
&= \frac{\partial J_h(\theta_n)}{\partial \theta_n(s, a_+)},
\end{aligned}$$

where the inequality follows with

$$Q_h(s, a_+) = Q_h(s, a_+) - J_{h,s}^\infty + J_{h,s}^\infty$$
$$> J_{h,s}^\infty$$
$$= Q_h(s, a) - J_{h,s}^\infty + J_{h,s}^\infty$$
$$= Q_h(s, a),$$

as $Q_h(s, a_+) - J_{h,s}^\infty > 0$ a.s. for $a_+ \in I_+^s$ and $Q_h(s, a) - J_{h,s}^\infty = 0$ a.s. for $a \in I_0^s$. Now by assumption we have $\pi^{\theta_n}(a|s) \leq \pi^{\theta_n}(a_+|s)$ and thus $\theta_n(s, a) \leq \theta_n(s, a_+)$. It follows

$$\theta_{n+1}(s, a) = \theta_n(s, a) + \eta_h \frac{\partial J_h(\theta_n)}{\partial \theta_n(s, a)} \leq \theta_n(s, a_+) + \eta_h \frac{\partial J_h(\theta_n)}{\partial \theta_n(s, a_+)} = \theta_{n+1}(s, a_+).$$

$\square$

Now define for every $a_+ \in I_+^s$ the set

$$B_0^s(a_+) = \{a \in I_0^s | \pi^{\theta_n}(a_+|s) \leq \pi^{\theta_n}(a|s) \text{ for all } l > 0\}$$

and denote its complement in $I_0^s$ as $\bar{B}_0^s(a_+) = I_0^s \setminus B_0^s(a_+)$.

**Lemma B.7.** *Suppose $I_+^s \neq \emptyset$. For all $a_+ \in I_+^s$, we have that $B_0^s(a_+) \neq \emptyset$ and*

$$\sum_{a \in B_0^s(a_+)} \pi^{\theta_n}(a|s) \to 1, \text{ as } n \to \infty.$$

*This implies:*

$$\max_{a \in B_0^s(a_+)} \theta_n(s, a) \to \infty, \text{ for } n \to \infty.$$

*Proof.* Let $a_+ \in I_+^s$ and consider $a \in \bar{B}_0^s(a_+)$. Then by definition of $\bar{B}_0^s(a_+)$ there exists $n' > 0$ such that $\pi^{\theta_{n'}}(a_+|s) \geq \pi^{\theta_{n'}}(a|s)$. Hence, by Lemma B.6 for all $n \geq n'$ we have $\pi^{\theta_n}(a_+|s) \geq \pi^{\theta_n}(a|s)$. As $\pi^{\theta_n}(a_+|s) \to 0$ for $n \to \infty$. We obtain $\pi^{\theta_n}(a|s) \to 0$ for $n \to \infty$, for all $a \in \bar{B}_0^s(a_+)$. Since by Lemma B.3 $\sum_{a \in I_0^s} \pi^{\theta_n}(a|s) \to 1$ for $n \to \infty$, we have that $B_0^s(a_+) \neq \emptyset$ and that $\sum_{a \in B_0^s(a_+)} \pi^{\theta_n}(a|s) \to 1$, as $n \to \infty$. The second claim follows from this as in Lemma B.5. $\square$

**Lemma B.8.** *Consider $s \in \mathcal{S}_h$ such that $I_+^s \neq \emptyset$. Then, for any $a_+ \in I_+^s$, there exists an $N_{a_+}$ such that for all $n > N_{a_+}$ we have*

$$\pi^{\theta_n}(a_+|s) > \pi^{\theta_n}(a|s) \text{ for all } a \in \bar{B}_0^s(a_+).$$

*Proof.* For every $a \in \bar{B}_0^s(a_+)$ exists time $n_a$ such that

$$\pi^{\theta_n}(a_+|s) > \pi^{\theta_n}(a|s) \text{ for all } a \in \bar{B}_0^s(a_+)$$

for all $n > n_a$ by definition. Set $N_{a_+} = \max_{a \in \bar{B}_0^s(a_+)} n_a$ and the proof is completed. $\square$

**Lemma B.9.** *Assume again $I_+^s \neq \emptyset$. For all actions $a \in I_+^s$, we have that $\theta_n(s, a)$ is bounded from below as $n \to \infty$. And for all $a \in I_-^s$, we have that $\theta_n(s, a) \to -\infty$ as $n \to \infty$.*

*Proof.* The first claim follows directly with Lemma B.4 as $\theta_n(s, a)$ is strictly increasing for all $a \in I_+^s$, $n > N_1$ and thus for all $n > N_1$ we have $\theta_n(s, a) \geq \theta_{N_1}(s, a)$. Now suppose $a \in I_-^s$, then by Lemma B.4 we have that $\theta_n(s, a)$ is strictly decreasing for $n > N_1$. Assume there exists $b$ such that $\lim_{n \to \infty} \theta_n(s, a) = b$, then $\theta_n(s, a) > b$ for all $n > N_1$. By Lemma B.5 there exists an action $a' \in \mathcal{A}_s$ such that $\theta_n(s, a') \to -\infty$ for $n \to \infty$. Consider $\delta > 0$ such that $\theta_{N_1}(s, a') \geq b - \delta$. Define for all $n > N_1$

$$\tau(n) = \max\{k \in (N_1, n] : \theta_k(s, a') \geq b - \delta\}.$$

Define also

$$\mathcal{T}^{(n)} = \left\{ \tau(n) < n' < n : \frac{\partial J_h(\theta_{n'})}{\partial \theta_{n'}(s, a')} \leq 0 \right\},$$

as the set of all indices $n'$ in $(\tau(n), n)$, where $\theta_{n'}(s, a')$ is decreasing. Next we define $Z_n := \sum_{n' \in \mathcal{T}^{(n)}} \frac{\partial J_h(\theta_{n'})}{\partial \theta_{n'}(s, a')}$, then it holds that

$$
\begin{aligned}
Z_n &= \sum_{n' \in \mathcal{T}^{(n)}} \frac{\partial J_h(\theta_{n'})}{\partial \theta_{n'}(s, a')} \\
&\leq \sum_{n' = \tau(n)+1}^{n-1} \frac{\partial J_h(\theta_{n'})}{\partial \theta_{n'}(s, a')} \\
&\leq \sum_{n' = \tau(n)}^{n-1} \frac{\partial J_h(\theta_{n'})}{\partial \theta_{n'}(s, a')} + \left| \frac{\partial J_h(\theta_{\tau(n)})}{\partial \theta_{\tau(n)}(s, a')} \right|.
\end{aligned}
$$

By Lemma 3.2 and the bounded reward assumption we have

$$\left| \frac{\partial J_h(\theta_{\tau(n)})}{\partial \theta_{\tau(n)}(s, a')} \right| == \mu_h(s) \pi^{\theta_{\tau(n)}}(a'|s) |A_h^{\theta_{\tau(n)}}(s, a')| \leq (H - h)R^*.$$

Hence,

$$
\begin{aligned}
Z_n &\leq \sum_{n' = \tau(n)}^{n-1} \frac{\partial J_h(\theta_{n'})}{\partial \theta_{n'}(s, a')} + (H - h)R^* \\
&= \frac{1}{\eta}(\theta_n(s, a') - \theta_{\tau(n)}(s, a')) + (H - h)R^* \\
&\leq \frac{1}{\eta}(\theta_n(s, a') - b + \delta) + (H - h)R^*.
\end{aligned}
$$

Then $\theta_n(s, a') \to -\infty$ for $n \to \infty$ implies that $Z_n \to -\infty$ for $n \to \infty$. As we chose $a \in I_-^s$ it holds that $|A_h^{\theta_n}(s, a)| \geq \frac{\Delta}{4}$ for $n > N_1$ with Lemma B.2 and so for all $n' \in \mathcal{T}^{(n)}$:

$$
\begin{aligned}
\left| \frac{\frac{\partial J_h(\theta_{n'})}{\partial \theta_{n'}(s, a)}}{\frac{\partial J_h(\theta_{n'})}{\partial \theta_{n'}(s, a')}} \right| &= \left| \frac{\pi^{\theta_{n'}}(a|s) A_h^{\theta_{n'}}(s, a)}{\pi^{\theta_{n'}}(a'|s) A_h^{\theta_{n'}}(s, a')} \right| \\
&\geq \frac{\pi^{\theta_{n'}}(a|s)}{\pi^{\theta_{n'}}(a'|s)} \frac{\Delta}{4(H - h)R^*} \\
&= \exp(\theta_{n'}(s, a) - \theta_{n'}(s, a')) \frac{\Delta}{4(H - h)R^*} \\
&\geq \exp(b - (b - \delta)) \frac{\Delta}{4(H - h)R^*} \\
&= \exp(\delta) \frac{\Delta}{4(H - h)R^*},
\end{aligned}
$$

where we used in the last inequality that $\theta_{n'}(s, a') \leq b - \delta$ for all $n' > \tau(n)$ and $\theta_{n'}(s, a) > b$ for all $n' > N_1$. By the definition of $\mathcal{T}^{(n)}$ these inequalities holds especially for all $n' \in \mathcal{T}^{(n)}$. Using

this we can imply that for all $n > N_1$ with $\mathcal{T}^{(n)} \neq \emptyset$,

$$
\begin{aligned}
\frac{1}{\eta}\Big(\theta_{N_1}(s,a) - \theta_n(s,a)\Big) &= \sum_{n'=N_1+1}^{n-1} \frac{\partial J_h(\theta_{n'})}{\partial \theta_{n'}(s,a)} \\
&\leq \sum_{n' \in \mathcal{T}^{(n)}} \frac{\partial J_h(\theta_{n'})}{\partial \theta_{n'}(s,a)} \\
&\leq \exp(\delta)\frac{\Delta}{4(H-h)R^*} \sum_{n' \in \mathcal{T}^{(n)}} \frac{\partial J_h(\theta_{n'})}{\partial \theta_{n'}(s,a')} \\
&= \exp(\delta)\frac{\Delta}{4(H-h)R^*} Z_n,
\end{aligned}
$$

where the first inequality holds because $\theta_{n'}(s,a)$ is strictly decreasing for $n' > N_1$, i.e. $\frac{\partial J_h(\theta_{n'})}{\partial \theta_{n'}(s,a)} > 0$ for all $n' \in \{N_1 + 1, \ldots, n-1\}$. In the second inequality we used

$$
\left| \frac{\frac{\partial J_h(\theta_{n'})}{\partial \theta_{n'}(s,a)}}{\frac{\partial J_h(\theta_{n'})}{\partial \theta_{n'}(s,a')}} \right| \geq \exp(\delta)\frac{\Delta}{4(H-h)R^*}.
$$

Note that $\frac{\partial J_h(\theta_{n'})}{\partial \theta_{n'}(s,a)} < 0$ and $\frac{\partial J_h(\theta_{n'})}{\partial \theta_{n'}(s,a')} < 0$ so that the sign of the inequality reverses.

Finally, we deduce from $Z_n \to -\infty$ that $\theta_n(s,a) \to \infty$ for $n \to \infty$, which is a contradiction to $\theta_n(s,a)$ strictly decreasing for all $n > N_1$. $\qquad\square$

**Lemma B.10.** *Consider $s \in \mathcal{S}_h$ such that $I_+^s \neq \emptyset$. Then for any $a_+ \in I_+^s$ it holds that*

$$
\sum_{a \in B_0^s(a_+)} \theta_n(s,a) \to \infty, \quad \text{for } n \to \infty.
$$

*Proof.* Let $a_+ \in I_+^s$ and $a \in B_0^s(a_+)$. Then by definition of $B_0^s(a_+)$ we have

$$
\pi^{\theta_n}(a_+|s) \leq \pi^{\theta_n}(a|s)
$$

for all $n > 0$ and hence by softmax parametrization $\theta_n(s,a_+) \leq \theta_n(s,a)$ for all $n > 0$. By Lemma B.9 we have that $\theta_n(s,a_+)$ and thus also $\theta_n(s,a)$ is bounded from below for $n \to \infty$. Together with

$$
\max_{\{a \in B_0^s(a_+)\}} \theta_n(s,a) \to \infty, \quad \text{for } n \to \infty
$$

by Lemma B.7 we deduce the claim. $\qquad\square$

Finally, we are ready to prove the asymptotic convergence of policy gradient with tabular softmax parametrization.

*Proof of Theorem 3.5.* We have to show that $I_+^s = \emptyset$ for all $s \in \mathcal{S}_h$. So assume there exists $s \in \mathcal{S}_h$ such that $I_+^s \neq \emptyset$ and let $a_+ \in I_+^s$. Then by Lemma B.10 we have

$$
\sum_{a \in B_0^s(a_+)} \theta_n(s,a) \to \infty, \quad \text{for } n \to \infty. \tag{15}
$$

For any $a \in I_-^s$ we have by Lemma B.9 that

$$
\frac{\pi^{\theta_n}(a|s)}{\pi^{\theta_n}(a_+|s)} = \exp(\underbrace{\theta_n(s,a)}_{\to -\infty} - \underbrace{\theta_n(s,a_+)}_{\text{bounded from below}}) \to 0, \quad n \to \infty.
$$

Hence, there exists $N_2 > N_1$ such that for all $n > N_2$

$$
\frac{\pi^{\theta_n}(a|s)}{\pi^{\theta_n}(a_+|s)} < \frac{\Delta}{16|\mathcal{A}|(H-h)R^*},
$$

which leads for $n > N_2$ to

$$-(H - h)R^* \sum_{a \in I_-^s} \pi^{\theta_n}(a|s) > -\frac{\Delta}{16}\pi^{\theta_n}(a_+|s). \tag{16}$$

Note that if $I_-^s = \emptyset$ we can just ignore this sum later on.

Next consider $a \in \bar{B}_0^s(a_+) \subseteq I_0^s$. By the definition of $I_0^s$ we have that $A_h^{\theta_n}(s, a) \to A_h^\infty(s, a) = 0$ for $n \to \infty$. By Lemma B.8 we have for $n \geq N_{a_+}$

$$1 < \frac{\pi^{\theta_n}(a_+|s)}{\pi^{\theta_n}(a|s)}.$$

Thus, there exists $N_3 > \max\{N_2, N_{a_+}\}$ such that for all $n \geq N_3$

$$|A_h^{\theta_n}(s, a)| < \frac{\pi^{\theta_n}(a_+|s)}{\pi^{\theta_n}(a|s)}\frac{\Delta}{16|\mathcal{A}|}.$$

This implies

$$\sum_{a \in \bar{B}_0^s(a_+)} \pi^{\theta_n}(a|s)|A_h^{\theta_n}(s, a)| < \pi^{\theta_n}(a_+|s)\frac{\Delta}{16}$$

and so

$$-\pi^{\theta_n}(a_+|s)\frac{\Delta}{16} < \sum_{a \in \bar{B}_0^s(a_+)} \pi^{\theta_n}(a|s)A_h^{\theta_n}(s, a) < \pi^{\theta_n}(a_+|s)\frac{\Delta}{16}, \tag{17}$$

for all $n > N_3$. We can conclude again for $n > N_3$,

$$\begin{aligned}
0 &= \sum_{a \in \mathcal{A}} \pi^{\theta_n}(a|s)A_h^{\theta_n}(s, a) \\
&= \sum_{a \in B_0^s(a_+)} \pi^{\theta_n}(a|s)A_h^{\theta_n}(s, a) + \sum_{a \in \bar{B}_0^s(a_+)} \pi^{\theta_n}(a|s)A_h^{\theta_n}(s, a) \\
&\quad + \sum_{a \in I_+^s} \pi^{\theta_n}(a|s)A_h^{\theta_n}(s, a) + \sum_{a \in I_-^s} \pi^{\theta_n}(a|s)A_h^{\theta_n}(s, a) \\
&> \sum_{a \in B_0^s(a_+)} \pi^{\theta_n}(a|s)A_h^{\theta_n}(s, a) - \pi^{\theta_n}(a_+|s)\frac{\Delta}{16} + \pi^{\theta_n}(a_+|s)\frac{\Delta}{4} - (H - h)R^* \sum_{a \in I_-^s} \pi^{\theta_n}(a|s) \\
&\geq \sum_{a \in B_0^s(a_+)} \pi^{\theta_n}(a|s)A_h^{\theta_n}(s, a) - \pi^{\theta_n}(a_+|s)\frac{\Delta}{16} + \pi^{\theta_n}(a_+|s)\frac{\Delta}{4} - \frac{\Delta}{16}\pi^{\theta_n}(a_+|s) \\
&> \sum_{a \in B_0^s(a_+)} \pi^{\theta_n}(a|s)A_h^{\theta_n}(s, a),
\end{aligned}$$

where we used Equation (17) and Lemma B.2 in the first inequality and Equation (16) in the second inequality. Finally, by our assumption and Equation (15) for $n > N_3$,

$$\begin{aligned}
\infty \stackrel{n \to \infty}{\longleftarrow} &\sum_{a \in B_0^s(a_+)} (\theta_n(s, a) - \theta_{N_3}(s, a)) \\
&= \eta_h \sum_{n'=N_3}^{n} \sum_{a \in B_0^s(a_+)} \frac{\partial J_h(\theta_n)}{\partial \theta_{n'}(s, a)} \\
&= \eta_h \sum_{n'=N_3}^{n} \mu_h(s) \sum_{a \in B_0^s(a_+)} \pi^{\theta_n}(a|s)A_h^{\theta_n}(s, a),
\end{aligned}$$

which contradicts $\sum_{a \in B_0^s(a_+)} \pi^{\theta_n}(a|s)A_h^{\theta_n}(s, a) < 0$. $\qquad\square$

 **B.2    Proofs of Section 3.2**

642 **Lemma 3.6** (weak PL-inequality). *For the objective $J_h$ it holds that*

$$\|\nabla J_h(\theta)\|_2 \geq \min_{s \in \mathcal{S}_h} \pi^\theta(a_h^*(s)|s)(J_h^* - J_h(\theta)),$$

643 *where $a_h^*(s) = argmax_{a \in \mathcal{A}_s} \pi_h^*(a|s)$ and $J_h^* = \sup_\theta J_h(\theta)$.*

644 *Proof.* First note that by the definition of $\pi_h^*$, we have $J_h^* = V_h^{(\pi_h^*, \tilde{\pi}_{(h+1)})}(\mu)$, because the tabular
645 softmax parametrization can approximate any deterministic policy arbitrarily well. We denote by
646 $J_{h,s}^* = V_h^{(\pi_h^*, \tilde{\pi}_{(h+1)})}(s)$ the optimal $h$-state value function for all $s \in \mathcal{S}_h$, when the policy after $h$ is
647 fixed. Using the performance difference lemma with fixed policy after $h$ (Corollary 2.3), we obtain

$$\left\| \frac{\partial J_h(\theta)}{\partial \theta} \right\|_2$$

$$= \left\| \sum_{s \in \mathcal{S}_h} \mu_h(s) \frac{\partial J_{h,s}(\theta)}{\partial \theta} \right\|_2$$

$$= \left[ \sum_{s' \in \mathcal{S}_h} \sum_{a' \in \mathcal{A}_{s'}} \left( \sum_{s \in \mathcal{S}_h} \mu_h(s) \frac{\partial J_{h,s}(\theta)}{\partial \theta(s', a')} \right)^2 \right]^{\frac{1}{2}}$$

$$\geq \sum_{s \in \mathcal{S}_h} \mu_h(s) \left| \frac{\partial J_{h,s}(\theta)}{\partial \theta(s, a_h^*(s))} \right|$$

$$= \sum_{s \in \mathcal{S}_h} \mu_h(s) \pi^\theta(a_h^*(s)|s) A_h^{(\pi^\theta, \tilde{\pi}_{(h+1)})}(s, a_h^*(s))$$

$$= \sum_{s \in \mathcal{S}_h} \mu_h(s) \pi^\theta(a_h^*(s)|s) \left( J_{h,s}^* - J_{h,s}(\theta) \right)$$

$$\geq \min_{s \in \mathcal{S}_h} \pi^\theta(a_h^*(s)|s) \left( J_h^* - J_h(\theta) \right),$$

648 where the first inequality is due to the positiveness of all other terms, and we just drop them, and in
649 the last equation we used Corollary 2.3, i.e. $A_h^{(\pi^\theta, \tilde{\pi}_{(h+1)})}(s, a_h^*(s)) = \mathbb{E}_{S_h=s}^{\pi^*}[A_h^{(\pi^\theta, \tilde{\pi}_{(h+1)})}(S_t, A_t)]$.
650 This proves the claim.                                                                                            □

651 **Lemma 3.7.** *Let $h \in \mathcal{H}$, $\mu_h(s) > 0$ for all $s \in \mathcal{S}_h$ and consider the sequence $(\theta_n)_{n \in \mathbb{N}_0}$ generated by*
652 *(7) for arbitrarily initialized $\theta_0 \in \mathbb{R}^{d_h}$. Then it holds that $c_h := \inf_{n \geq 0} \min_{s \in \mathcal{S}_h} \pi^{\theta_n}(a_h^*(s)|s) > 0$.*

653 All in all the proof follows the outline of (Mei et al., 2020, Lemma 9), but has to be adjusted to the
654 finite-time setting in a few steps.

655 *Proof.* First note that

$$J_{h,s}(\theta) = \sum_{a \in \mathcal{A}_s} \pi_t^\theta(a|s) Q_h^{\tilde{\pi}}(s, a),$$

656 where $Q_h^{\tilde{\pi}}(s, a)$ is independent of $\theta$. We will drop the subscript $\tilde{\pi}$ in $Q_h$ for the rest of the proof and
657 define for all $s \in \mathcal{S}_h$,

$$\Delta^*(s) = Q_h(s, a_h^*(s)) - \max_{a \neq a_h^*(s)} Q_h(s, a) > 0, \quad \text{and} \quad \Delta^* = \min_{s \in \mathcal{S}_h} \Delta^*(s) > 0.$$

658 Now consider for any $s \in \mathcal{S}_h$ the following sets

$$\mathcal{R}_1(s) = \left\{ \theta : \frac{\partial J_{h,s}(\theta)}{\partial \theta(s, a_h^*(s))} \geq \frac{\partial J_{h,s}(\theta)}{\partial \theta(s, a)}, \text{ for all } a \neq a_h^*(s) \right\},$$

$$\mathcal{R}_2(s) = \left\{ \theta_n : J_{h,s}(\theta_{n'}) \geq Q_h(s, a_h^*(s)) - \frac{\Delta^*(s)}{2}, \text{ for all } n' \geq n \right\}.$$

Furthermore, we define $c(s) = \frac{|\mathcal{A}|(H-h)R^*}{\Delta^*(s)} - 1$ and

$$N_c(s) = \left\{ \theta : \pi^\theta(a_h^*(s)|s) \geq \frac{c(s)}{c(s)+1} \right\}.$$

We divide the proof into the following Claims:

**Claim 1.** $\mathcal{R}(s) = \mathcal{R}_1(s) \cap \mathcal{R}_2(s)$ is a *nice* region, i.e.

      (i) $\theta_n \in \mathcal{R}(s) \Rightarrow \theta_{n+1} \in \mathcal{R}(s)$.

      (ii) $\pi^{\theta_{n+1}}(a_h^*(s)|s) \geq \pi^{\theta_n}(a_h^*(s)|s)$.

**Claim 2.** $\mathcal{N}_c(s) \cap \mathcal{R}_2(s) \subseteq \mathcal{R}_1(s) \cap \mathcal{R}_2(s)$.

**Claim 3.** For every $s \in \mathcal{S}_h$, there exists a finite-time $n_0(s) \geq 1$, such that $\theta_{n_0(s)} \in \mathcal{N}_c(s) \cap \mathcal{R}_2(s) \subseteq \mathcal{R}_1 s \cap \mathcal{R}_2(s)$ and thus $\inf_{n\geq 1} \pi^{\theta_n}(a_h^*(s)|s) = \min_{1\leq n \leq n_0(s)} \pi^{\theta_{n_0(s)}}(a_h^*(s)|s)$.

If all three claims hold true, we can finally define $n_0 = \max_{s\in\mathcal{S}_h} n_0(s)$, such that

$$\inf_{n\geq 1, s\in\mathcal{S}_h} \pi^{\theta_n}(a_h^*(s)|s) = \min_{1\leq n\leq n_0, s\in\mathcal{S}_h} \pi^{\theta_{n_0}}(a_h^*(s)|s).$$

Due to the positiveness of the softmax parametrization the assertion follows.

**Claim 1.** We first prove (i). Let $\theta_n \in \mathcal{R}(s)$ and $a \neq a_h^*(s)$. Then $\theta_{n+1} \in \mathcal{R}_2(s)$ by definition of $\mathcal{R}_2(s)$. Using Lemma 3.2 we obtain

$$\frac{\partial J_{h,s}(\theta_n)}{\partial\theta(s, a_h^*(s))} \geq \frac{\partial J_{h,s}(\theta_n)}{\partial\theta(s,a)} \tag{18}$$
$$\Leftrightarrow \pi^{\theta_n}(a_h^*(s)|s)\big(Q_h(s, a_h^*(s)) - J_{h,s}(\theta_n)\big) \geq \pi^{\theta_n}(a|s)\big(Q_h(s,a) - J_{h,s}(\theta_n)\big).$$

We divide into two cases:

      a) $\pi^{\theta_n}(a_h^*(s)|s) \geq \pi^{\theta_n}(a|s)$,

      b) $\pi^{\theta_n}(a_h^*(s)|s) < \pi^{\theta_n}(a|s)$.

In $a)$ the assumption $\pi^{\theta_n}(a_h^*(s)|s) \geq \pi^{\theta_n}(a|s)$ implies $\theta_n(s, a_h^*(s)) \geq \theta_n(s,a)$. Thus,

$$\theta_{n+1}(s, a_h^*(s)) = \theta_n(s, a_h^*(s)) + \eta_h \frac{\partial J_{h,s}(\theta_n)}{\partial\theta_n(s, a_h^*(s))}$$
$$\geq \theta_n(s,a) + \eta_h \frac{\partial J_{h,s}(\theta_n)}{\partial\theta_n(s,a)}$$
$$= \theta_{n+1}(s,a),$$

which implies $\pi^{\theta_{n+1}}(a_h^*(s)|s) \geq \pi^{\theta_{n+1}}(a|s)$. By the optimality of $a_h^*(s)$ we follow

$$\pi_t^{\theta_{n+1}}(a_h^*(s)|s)\big(Q_h(s, a_h^*(s)) - J_{h,s}(\theta_{n+1})\big) \geq \pi_t^{\theta_{n+1}}(a|s)\big(Q_h(s,a) - J_{h,s}(\theta_{n+1})\big),$$

which is by equation (18) equivalent to

$$\frac{\partial J_{h,s}(\theta_{n+1})}{\partial\theta_{n+1}(s, a_h^*(s))} \geq \frac{\partial J_{h,s}(\theta_{n+1})}{\partial\theta_{n+1}(s,a)}.$$

Hence, $\theta_{n+1} \in \mathcal{R}_1(s)$.

In $b)$ assume now that $\pi^{\theta_n}(a_h^*(s)|s) < \pi^{\theta_n}(a|s)$. As $\theta_n \in \mathcal{R}_1(s)$ equation (18) is also true in this case and rearranging of terms gives

$$\frac{\partial J_{h,s}(\theta_n)}{\partial\theta_n(s, a_h^*(s))} \geq \frac{\partial J_{h,s}(\theta_n)}{\partial\theta_n(s,a)}$$
$$\Leftrightarrow Q_h(s, a_h^*(s)) - Q_h(s,a) \geq \left(1 - \frac{\pi^{\theta_n}(a_h^*(s)|s)}{\pi^{\theta_n}(a|s)}\right)\big(Q_h(s, a_h^*(s)) - J_{h,s}(\theta_n)\big)$$
$$\Leftrightarrow Q_h(s, a_h^*(s)) - Q_h(s,a) \geq \big(1 - \exp(\theta_n(s, a_h^*(s)) - \theta_n(s,a))\big)\big(Q_h(s, a_h^*(s)) - J_{h,s}(\theta_n)\big). \tag{19}$$

Note next that by $\theta^{(n)} \in \mathcal{R}_1(s)$ and definition of $\mathcal{R}_1(s)$ we have

$$\theta_{n+1}(s, a_h^*(s)) - \theta_{n+1}(s, a)$$
$$= \theta_n(s, a_h^*(s)) + \eta_h \frac{\partial J_{h,s}(\theta_n)}{\partial \theta_n(s, a_h^*(s))} - \theta_n(s, a) - \eta \frac{\partial J_{h,s}(\theta_n)}{\partial \theta_n(s, a)}$$
$$\geq \theta_n(s, a_h^*(s)) - \theta_n(s, a)$$

and is follows $\left(1 - \exp(\theta_{n+1}(s, a_h^*(s)) - \theta_{n+1}(s, a))\right) \leq \left(1 - \exp(\theta_n(s, a_h^*(s)) - \theta_n(s, a))\right) < 1$ by assumption $b)$. We already know $\theta_{n+1} \in \mathcal{R}_2(s)$ and therefore $J_{h,s}(\theta_{n+1}) \geq Q_h(s, a_h^*(s)) - \frac{\Delta^*(s)}{2}$. This leads to

$$Q_h(s, a_h^*(s)) - J_{h,s}(\theta_{n+1}) \leq \frac{\Delta^*(s)}{2} \leq Q_h(s, a_h^*(s)) - Q_h(s, a),$$

where the last inequality is due to the definition of $\Delta^*(s)$. Combining everything leads to

$$\left(1 - \exp(\theta_{n+1}(s, a_h^*(s)) - \theta_{n+1}(s, a))\right)\left[Q_h(s, a_h^*(s)) - J_{h,s}(\theta_{n+1})\right]$$
$$\leq Q_h(s, a_h^*(s)) - Q_h(s, a),$$

which is by equation (19) equivalent to $\theta_{n+1} \in \mathcal{R}_1(s)$.

Now we come to Claim (ii).

$$\pi^{\theta_{n+1}}(a_h^*(s)|s)$$
$$= \frac{\exp(\theta_{n+1}(s, a_h^*(s)))}{\sum_{a \in \mathcal{A}} \exp(\theta_{n+1}(s, a))}$$
$$= \frac{\exp(\theta_n(s, a_h^*(s)) + \eta_h \frac{\partial J_{h,s}(\theta_n)}{\partial \theta_n(s, a_h^*(s))})}{\sum_{a \in \mathcal{A}_s} \exp(\theta_n(s, a) + \eta_h \frac{\partial J_{h,s}(\theta_n)}{\partial \theta_n(s, a)})}$$
$$\geq \frac{\exp(\theta_n(s, a_h^*(s))) \exp(\eta_h \frac{\partial J_{h,s}(\theta_n)}{\partial \theta_n(s, a_h^*(s))})}{\sum_{a \in \mathcal{A}_s} \exp(\theta_n(s, a)) \exp(\eta_h \frac{\partial J_{h,s}(\theta_n)}{\partial \theta_n(s, a_h^*(s))})}$$
$$= \pi^{\theta_n}(a_h^*(s)|s),$$

where the inequality follows by $\theta_n \in \mathcal{R}_1(s)$.

**Claim 2.** Assume $\theta \in \mathcal{N}_c(s) \cap \mathcal{R}_2(s)$ and divide again in two cases. If $a)$ $\pi^\theta(a_h^*(s)|s) \geq \max_{a \in \mathcal{A}} \pi^\theta(a|s)$, then for all $a \neq a_h^*(s)$ we have

$$\frac{\partial J_h(\theta)}{\partial \theta(s, a_h^*(s))}$$
$$= \mu_h(s)\pi^\theta(a*(s)|s)A^{\pi^\theta}(s, a_h^*(s))$$
$$\geq \mu_h(s)\pi^\theta(a|s)A^{\pi^\theta}(s, a)$$
$$= \frac{\partial J_h(\theta)}{\partial \theta(s, a)}.$$

Hence, $\theta \in \mathcal{R}_1(s)$.

The case $b)$ where $\pi^\theta(a_h^*(s)|s) < \max_{a \in \mathcal{A}_s} \pi^\theta(a|s)$ is not possible for $\theta \in \mathcal{N}_c(s)$. Assume there exists $a \neq a_h^*(s)$ such that $\pi^\theta(a_h^*(s)|s) < \pi^\theta(a|s)$. Then

$$\pi^\theta(a_h^*(s)|s) + \pi^\theta(a|s) > \frac{2c(s)}{c(s) + 1} = \frac{\frac{2|\mathcal{A}|(H-h)R^*}{\Delta^*(s)} - 2}{\frac{|\mathcal{A}|(H-h)R^*}{\Delta^*(s)}} = 2 - \frac{2\Delta^*(s)}{|\mathcal{A}|(H-h)R^*} \geq 2 - \frac{2}{|\mathcal{A}|} \geq 1,$$

because $\Delta^*(s) \leq (H-h)R^*$ by definition and $|\mathcal{A}| \geq 2$. This is a contradiction as $\pi^\theta$ is a probability distribution and Claim 2 is proven.

**Claim 3.** By the asymptotic convergence for finite-time setting Theorem 3.5, we have that $\pi^{\theta_n}(a^*(s)|s) \to 1$ for $n \to \infty$. Thus, there exists an $N_0(s) > 0$, such that $\pi^{\theta_n}(a^*(s)|s) \geq \frac{c(s)}{c(s)+1}$ for all $n \geq N_0(s)$, i.e. $\theta_n \in N_c(s)$ for all $n \geq N_0(s)$.

Furthermore, $J_h(\theta_n) \to J_h^* = Q_h(s, a^*(s))$ for $n \to \infty$ which implies the existence of $N_1 > 0$ such that $\theta_n \in \mathcal{R}_2(s)$ for all $n \geq N_1(s)$. We choose $n_0(s) = \max\{N_0(s), N_1(s)\}$ which proves Claim 3.
$\square$

**Theorem 3.8.** *Let $h \in \mathcal{H}$, $\mu_h(s) > 0$ for all $s \in \mathcal{S}_h$ and consider the sequence $(\theta_n)_{n \in \mathbb{N}_0}$ generated by (7) for arbitrarily initialized $\theta_0 \in \mathbb{R}^{d_h}$. Define $c_h := \inf_{n \geq 0} \min_{s \in \mathcal{S}_h} \pi^{\theta_n}(a_h^*(s)|s) > 0$ by Lemma 3.7 and choose step size $\eta_h = \frac{1}{\beta_h}$ with $\beta_h = 2(H-h)R^*|\mathcal{A}|$. Then it holds that*

$$J_h^* - J_h(\theta_n) \leq \frac{4(H-h)R^*|\mathcal{A}|}{c_h^2 n},$$

*where $J_h^* = \sup_\theta J_h(\theta)$.*

*Proof.* For any $\beta$-smooth function $f : \mathbb{R}^d \to \mathbb{R}$ the descent lemma gives (see Beck, 2017, Lemma 5.7)

$$f(y) \leq f(x) + \nabla f(x)^T(y-x) + \frac{\beta}{2}\|y-x\|^2.$$

As $-f$ is also $\beta$-smooth we follow

$$-f(y) \leq -f(x) - \nabla f(x)^T(y-x) + \frac{\beta}{2}\|y-x\|^2,$$

which is equivalent to

$$f(y) \geq f(x) + \nabla f(x)^T(y-x) - \frac{\beta}{2}\|y-x\|^2. \tag{20}$$

Now for gradient ascent updates

$$x_{k+1} = x_k + \alpha\nabla f(x_k)$$

we have that

$$f(x_{k+1}) \geq f(x_k) + \nabla f(x_k)^T(x_{k+1} - x_k) - \frac{\beta}{2}\|x_{k+1} - x_k\|^2$$

$$= f(x_k) + \alpha\|\nabla f(x_k)\|^2 - \frac{\beta\alpha^2}{2}\|\nabla f(x_k)\|^2$$

$$= f(x_k) + \left(\alpha - \frac{\beta\alpha^2}{2}\right)\|\nabla f(x_k)\|^2.$$

It follows for the maximum $f(x^*)$ of $f$ that

$$f(x^*) - f(x_{k+1}) \leq f(x^*) - f(x_k) - \left(\alpha - \frac{\beta\alpha^2}{2}\right)\|\nabla f(x_k)\|^2.$$

Using this for our objective function $J_h$, we obtain for the gradient ascent updates

$$\theta_{n+1} = \theta_n + \eta_h\nabla J_h(\theta_n)$$

and $J_h^* = \sup_\theta J_h(\theta)$ that

$$J_h^* - J_h(\theta_{n+1}) \leq J_h^* - J_h(\theta_n) - \underbrace{\left(\eta_h - \frac{\beta_h\eta_h^2}{2}\right)}_{=\frac{1}{2\beta_h}>0,\text{ for }\eta_h=\frac{1}{\beta_h}} \underbrace{\|\nabla J_h(\theta_n)\|^2}_{\geq c_h^2(J_h^*-J_h(\theta_n))^2}$$

$$\leq (J_h^* - J_h(\theta_n))\left(1 - \frac{c_h^2}{2\beta_h}(J_h^* - J_h(\theta_n))\right).$$

714 The second inequality follows with the PL-type inequality in Lemma 3.6.

715 Define $q = \frac{c_h^2}{4(H-h)R^*|\mathcal{A}|} = \frac{c_h^2}{2\beta_h} > 0$, then

$$J_h^* - J_h(\theta_0) \leq (H-h)R^* \leq \frac{1}{q}.$$

716 We conclude using an argument similar to Nesterov (2013, Thm. 2.1.14). Therefore, define $d_n = $
717 $J_h^* - J_h(\theta_n)$, then

$$d_{n+1} \leq d_n - \frac{1}{q}d_n^2.$$

718 Thus,

$$\frac{1}{d_{n+1}} \geq \frac{1}{d_n} + \frac{d_n}{qd_{n+1}} \geq \frac{1}{d_n} + \frac{1}{q},$$

719 where the first inequality is due to dividing by $d_n d_{n+1}$ and the second inequality follows by mono-
720 tonicity (Lemma B.1). Using a telescope-sum argument we obtain

$$\frac{1}{d_n} = d_0 + \sum_{k=0}^{n-1} \frac{1}{d_k} - \frac{1}{d_{k-1}} \geq d_0 + \frac{n}{q}.$$

721 Finally,

$$J_h^* - J_h(\theta_n) = d_n \leq \frac{1}{\frac{1}{q}n + d_0} \leq \frac{1}{q(n+1)} \leq \frac{4(H-h)R^*|\mathcal{A}|}{c_h^2 n}.$$

722 $\qquad\qquad\qquad\qquad\qquad\qquad\qquad\qquad\qquad\qquad\qquad\qquad\qquad\qquad\qquad\qquad\qquad\qquad\qquad\square$

# 723 C Proofs of Section 4

724 **Lemma C.1.** *Consider the tabular softmax parametrization. For any $h \in \mathcal{H}$ and $K_0 > 0$ it holds*
725 *that*

$$\mathbb{E}_{\mu_h}^{(\pi^\theta, (\tilde{\pi})_{(h+1)})}[\widehat{\nabla} J_h^{K_h}(\theta)] = \nabla J_h(\theta)$$

726 *and*

$$\mathbb{E}_{\mu_h}^{(\pi^\theta, (\tilde{\pi})_{(h+1)})}[\|\widehat{\nabla} J_h^{K_h}(\theta) - \nabla J_h(\theta)\|^2] \leq \frac{5(H-h)^2(R^*)^2}{K_h} =: \frac{C_h}{K}.$$

727 *Proof.* By the definition of $\widehat{\nabla} J_h^K$ we have

$$\mathbb{E}_{\mu_h}^{(\pi^\theta, (\tilde{\pi})_{(h+1)})}[\widehat{\nabla} J_h^{K_h}(\theta)]$$
$$= \mathbb{E}_{\mu_h}^{(\pi^\theta, (\tilde{\pi})_{(h+1)})} \Big[ \frac{1}{K_h} \sum_{i=1}^{K_h} \nabla \log(\pi^\theta(A_t^i|S_t^i))\hat{Q}_h(S_h^i, A_h^i) \Big]$$
$$= \mathbb{E}_{\mu_h}^{(\pi^\theta, (\tilde{\pi})_{(h+1)})} \Big[ \nabla \log(\pi^\theta(A_h^1|S_h^1))\hat{Q}_h(S_h^1, A_h^1) \Big]$$
$$= \mathbb{E}_{\mu_h}^{(\pi^\theta, (\tilde{\pi})_{(h+1)})} \Big[ \nabla \log(\pi^\theta(A_1|S_h)) \sum_{k=h}^{H-1} r(S_k, A_k) \Big],$$

728 where we used that we consider independent samples for $i = 1, \ldots, K_h$. From the proof of the policy
729 gradient Theorem 2.2, we obtain that

$$\mathbb{E}_{\mu_h}^{(\pi^\theta, (\tilde{\pi})_{(h+1)})}[\widehat{\nabla} J_h^{K_h}(\theta)]$$
$$= \mathbb{E}_{\mu_h}^{(\pi^\theta, (\tilde{\pi})_{(h+1)})} \Big[ \nabla \log(\pi^\theta(A_1|S_h)) \sum_{k=h}^{H-1} r(S_k, A_k) \Big]$$
$$= \nabla J_h(\theta).$$

For the second claim we have

$$\mathbb{E}_{\mu_h}^{(\pi^\theta,(\tilde{\pi})(h+1))}\Big[\|\widehat{\nabla}J_h^{K_h}(\theta) - \nabla J_h(\theta)\|^2\Big]$$

$$\leq \frac{1}{K_h}\mathbb{E}_{\mu_h}^{(\pi^\theta,(\tilde{\pi})(h+1))}\Big[\|\nabla\log(\pi^\theta(A_h|S_h))\hat{Q}_h(S_h,A_h) - \nabla J_h(\theta)\|^2\Big]$$

$$= \frac{1}{K_h}\mathbb{E}_{\mu_h}^{(\pi^\theta,(\tilde{\pi})(h+1))}\Big[\sum_{s\in\mathcal{S}_h}\sum_{a\in\mathcal{A}_s}\Big(\mathbf{1}_{s=S_h}(\mathbf{1}_{a=A_h} - \pi^\theta(a|s))\sum_{k=h}^{H-1}r(S_k,A_k)$$

$$- \mu_h(s)\pi^\theta(a|s)A_h^{(\pi^\theta,(\tilde{\pi})(h+1))}(s,a)\Big)^2\Big],$$

by the definition of $\widehat{\nabla}J_h^{K_h}(\theta)$ and the derivative of $\nabla J_h(\theta)$ for the softmax parametrization. Further,

$$\mathbb{E}_{\mu_h}^{(\pi^\theta,(\tilde{\pi})(h+1))}\Big[\|\widehat{\nabla}J_h^{K_h}(\theta) - \nabla J_h(\theta)\|^2\Big]$$

$$\leq \frac{1}{K_h}\mathbb{E}_{\mu_h}^{(\pi^\theta,(\tilde{\pi})(h+1))}\Big[\sum_{a\in\mathcal{A}_s}(\mathbf{1}_{a=A_h} - \pi^\theta(a|S_h))^2\Big(\sum_{k=h}^{H-1}r(S_k,A_k)\Big)^2$$

$$- 2\sum_{a\in\mathcal{A}_s}(\mathbf{1}_{a=A_h} - \pi^\theta(a|S_h))\sum_{k=h}^{H-1}r(S_k,A_k)\mu_h(s)\pi^\theta(a|S_h)A_h^{(\pi^\theta,(\tilde{\pi})(h+1))}(S_h,a)$$

$$+ \sum_{s\in\mathcal{S}_h}\sum_{a\in\mathcal{A}_s}\mu(s)^2\pi^\theta(a|s)^2A_h^{(\pi^\theta,(\tilde{\pi})(h+1))}(s,a)^2\Big].$$

We consider all three terms separately. For the first term we have

$$\mathbb{E}_{\mu_h}^{(\pi^\theta,(\tilde{\pi})(h+1))}\Big[\sum_{a\in\mathcal{A}_s}(\mathbf{1}_{a=A_h} - \pi^\theta(a|S_h))^2\Big(\sum_{k=h}^{H-1}r(S_k,A_k)\Big)^2\Big]$$

$$= \mathbb{E}_{\mu_h}^{(\pi^\theta,(\tilde{\pi})(h+1))}\Big[\Big(\sum_{k=h}^{H-1}r(S_k,A_k)\Big)^2\Big] - 2\mathbb{E}_{\mu_h}^{(\pi^\theta,(\tilde{\pi})(h+1))}\Big[\pi^\theta(A_h|S_h)\Big(\sum_{k=h}^{H-1}r(S_k,A_k)\Big)^2\Big]$$

$$+ \mathbb{E}_{\mu_h}^{(\pi^\theta,(\tilde{\pi})(h+1))}\Big[\sum_{a\in\mathcal{A}_s}\pi^\theta(a|S_h)^2\Big(\sum_{k=h}^{H-1}r(S_k,A_k)\Big)^2\Big]$$

$$\leq ((H-h)R^*)^2 - 0 + ((H-h)R^*)^2$$

$$= 2((H-h)R^*)^2,$$

by bounded reward assumption and the fact that $\pi^\theta$ is a probability distribution. For the second term, we note that $A_h^{(\pi^\theta,(\tilde{\pi})(h+1))}(S_h,a)$ can be negative, therefore we consider the absolute value and obtain

$$2\mathbb{E}_{\mu_h}^{(\pi^\theta,(\tilde{\pi})(h+1))}\Big[\sum_{a\in\mathcal{A}_s}(\mathbf{1}_{a=A_h} - \pi^\theta(a|S_h))\sum_{k=h}^{H-1}r(S_k,A_k)\mu_h(s)\pi^\theta(a|S_h)\big|A_h^{(\pi^\theta,(\tilde{\pi})(h+1))}(S_h,a)\big|\Big]$$

$$\leq 2\mathbb{E}_{\mu_h}^{(\pi^\theta,(\tilde{\pi})(h+1))}\Big[\sum_{a\in\mathcal{A}_s}1\cdot(H-h)R^*\cdot 1\cdot\pi^\theta(a|S_h)\cdot(H-h)R^*\Big]$$

$$= 2((H-h)R^*)^2.$$

For the last term we have

$$\mathbb{E}_\mu^{\pi^\theta}\Big[\sum_{s\in\mathcal{S}_h}\sum_{a\in\mathcal{A}_s}\mu(s)^2\pi_t^\theta(a|s)^2A_t^{\pi_t^\theta}(s,a)^2\Big] \leq ((H-h)R^*)^2.$$

In total, it holds that

$$\mathbb{E}_{\mu_h}^{(\pi^\theta,(\tilde{\pi})(h+1))}\Big[\|\widehat{\nabla}J_h^{K_h}(\theta) - \nabla J_h(\theta)\|^2\Big] \leq \frac{5((H-h)R^*)^2}{K_h}.$$

$\square$

 **C.1 Proofs of Section 4.1**

740 We state the stochastic approximation theorem in Bertsekas and Tsitsiklis (2000) to prove convergence
741 of stochastic softmax policy gradient to a stationary point.

742 **Proposition C.2** (Bertsekas and Tsitsiklis (2000), Proposition 3)**.** *Let $F : \mathbb{R}^d \to \mathbb{R}$ be an L-smooth*
743 *function, i.e.*

$$\|\nabla F(x) - \nabla F(y)\| \leq L\|x - y\|.$$

744 *Consider $(X_n)$ a sequence generated by*

$$X_{n+1} = X_n + \gamma_n(S_n + W_n),$$

745 *where $(\gamma_n)$ is deterministic positive step size, $S_n$ a descent direction, and $W_n$ is a random noise term.*
746 *Let $(\mathcal{F}_n)$ be an increasing sequence of $\sigma$-fields. We assume the following:*

747    *(i) $\sum_{n \geq 1} \gamma_n = \infty$, and $\sum_{n \geq 1} \gamma_n^2 < \infty$.*

748    *(ii) $(X_n)_{n \geq 0}$ and $(W_n)_{n \geq 0}$ are $(\mathcal{F}_n)$-measurable.*

749    *(iii) There exists positive constants $C_1$ and $C_2$ such that for all $n \geq 1$*
$$C_1\|\nabla F(X_n)\|^2 \leq -\nabla F(X_n)^T S_n \quad and \quad \|S_n\| \leq C_2(1 + \|\nabla F(X_n)\|^2),$$

750    *(iv) There exists a positive deterministic constant $C$ such that for all $n \geq 1$,*
$$\mathbb{E}[W_n|\mathcal{F}_n] = 0 \quad and \quad \mathbb{E}[\|W_n\|^2|\mathcal{F}_n] \leq C(1 + \|\nabla F(X_n)\|^2).$$

751 *Then either $F(X_n) \to \infty$ for $t \to \infty$ or $F(X_n)$ converges to a finite function such that*
752 $\lim_{n \to \infty} \nabla F(X_n) = 0$ *almost-surely.*

753 **Theorem 4.1.** *For any $h \in \mathcal{H}$ consider the stochastic process $(\theta_n)_{n \geq 0}$ generated by*

$$\theta_{n+1} = \theta_n + \eta_h^{(n)} \widehat{\nabla} J_h^{K_h}(\theta),$$

754 *for arbitrary batch size $K_h \geq 1$ and initial $\theta_0$ such that $\mathbb{E}[J_h(\theta_0)] < \infty$. Furthermore, suppose that*
755 $\eta_h^{(n)}$ *is decreasing, such that $\sum_{n \geq 0} \eta_h^{(n)} = \infty$ and $\sum_{n \geq 0} \left(\eta_h^{(n)}\right)^2 < \infty$. Then $\nabla J_h(\theta_n) \to 0$ almost*
756 *surely for $n \to \infty$.*

757 *Proof.* We apply Proposition C.2 as follows:
758 The function $F$ is the negative objective function with respect to parameter $\theta$, i.e.

$$F : \mathbb{R}^{d_h} \to \mathbb{R}, \quad \theta \mapsto -J_h(\theta).$$

759 Further, let

760    • $X_n \equiv \theta_n$,

761    • $S_n \equiv -\nabla F(\theta_n) = \nabla J_h(\theta_n)$,

762    • $W_n \equiv \widehat{\nabla} J_h^{K_h}(\theta_n) - \nabla J_h(\theta_n)$ and

763    • $\gamma_n \equiv \eta_h^{(n)}$.

764 Then,

$$\theta_{n+1} = \theta_n + \eta_h^{(n)} \widehat{\nabla} J_h^{K_h}(\theta_n) = X_n + \gamma_n(S_n + W_n).$$

765 Denote by $(\mathcal{F}_n)_{n \geq 0}$ the natural filtration of the stochastic process $(\theta_n)_{n \geq 0}$. Then, $X_n$ and $W_n$ are
766 $\mathcal{F}_n$-measurable and Condition (iii) is obviously satisfied using $C_1 = C_2 = 1$. By Lemma C.1 we
767 have that

$$\mathbb{E}[\widehat{\nabla} J_h^{K_h}(\theta_n)|\mathcal{F}_n] = \nabla J_h(\theta_n)$$

768 and

$$\mathbb{E}[\|\widehat{\nabla} J_h^{K_h}(\theta_n) - \nabla J_h(\theta_n)\|^2|\mathcal{F}_n] \leq \frac{C_h}{K_h}.$$

769 Thus, Condition (iv) is satisfied. Given the fact that the value function is bounded by the bounded
770 reward assumption we conclude

$$\nabla J_h(\theta_n) \to 0 \text{ for } n \to \infty.$$

771                                                                                                              □

 **C.2  Proofs of Section 4.2**

 **Lemma C.3.** *The softmax policy $\pi^\theta(a|s)$ is $\sqrt{2}$-Lipschitz with respect to $\theta \in \mathbb{R}^d$ for every $s, a$.*

 *Proof.* The derivative of the softmax function is

$$
\frac{\partial \pi^\theta(a|s)}{\partial \theta(s', a')} = \mathbf{1}_{s'=s} \left[ \frac{\mathbf{1}_{a'=a} \exp(\theta(s,a))\big(\sum_{\tilde{a} \in \mathcal{A}_s} \exp(\theta(s, \tilde{a}))\big) - \exp(\theta(s,a)) \exp(\theta(s,a'))}{\big(\sum_{\tilde{a} \in \mathcal{A}_s} \exp(\theta(s,\tilde{a}))\big)^2} \right]
$$

$$
= \mathbf{1}_{s'=s} \left[ \mathbf{1}_{a'=a} \pi^\theta(a|s) - \pi^\theta(a|s)\pi^\theta(a'|s) \right].
$$

 Therefore,

$$
\|\nabla \pi^\theta(a|s)\|_2 = \sqrt{\sum_{\tilde{a} \in \mathcal{A}_s} \Big( \mathbf{1}_{a'=a}\pi^\theta(a|s) - \pi^\theta(a|s)\pi^\theta(a'|s) \Big)^2}
$$

$$
\leq \sqrt{\pi^\theta(a|s)^2 - 2\pi^\theta(a|s)^3 + \sum_{\tilde{a} \in \mathcal{A}_s} \pi^\theta(a'|s)^2 \pi^\theta(a|s)^2}
$$

$$
\leq \sqrt{2}.
$$

 $\qquad\qquad\qquad\qquad\qquad\qquad\qquad\qquad\qquad\qquad\qquad\qquad\qquad\qquad\qquad\qquad\qquad\qquad\square$

 **Lemma C.4.** *It holds almost surely that $\min_{0 \leq n \leq \tau} \min_{s \in \mathcal{S}_h} \pi^{\theta_n}(a^*(s)|s) \geq \frac{c_h}{2}$ is strictly positive.*

 *Proof.* For every $n \leq \tau$ we obtain by the $\sqrt{2}$-Lipschitz continuity in Lemma C.3 that

$$
\pi^{\theta_n}(a^*(s)|s) \geq \pi^{\bar{\theta}_n}(a^*(s)|s) - |\pi^{\bar{\theta}_n}(a^*(s)|s) - \pi^{\theta_n}(a^*(s)|s)|
$$

$$
\geq \pi^{\bar{\theta}_n}(a^*(s)|s) - \sqrt{2}\|\theta_t - \bar{\theta}_n\|_2
$$

$$
> \frac{c_h}{2} > 0,
$$

 holds almost surely. The claim follows directly. $\qquad\qquad\qquad\qquad\qquad\qquad\qquad\qquad\square$

 **Lemma 4.2.** *Suppose $\mu_h(s) > 0$ for all $s \in \mathcal{S}_h$, the batch size $K_h^{(n)} \geq \frac{9c_h^2 C_h}{32\beta_h^2 N_h^{\frac{3}{2}}}(1 - \frac{1}{2\sqrt{N_h}})n^2$ is*

 *increasing for some $N_h \geq 1$ and the step size $\eta_h = \frac{1}{\beta_h \sqrt{N_h}}$, for fixed $h \in \mathcal{H}$. Then,*

$$
\mathbb{E}\big[(J_h^* - J_h(\theta_n))\mathbf{1}_{\{n \leq \tau\}}\big] \leq \frac{16\sqrt{N_h}\beta_h}{3(1 - \frac{1}{2\sqrt{N_h}})c_h^2 n}.
$$

 *Proof.* Fix $h \in \mathcal{H}$. Let $(\mathcal{F}_n)_{n \geq 0}$ be the natural filtration of $(\theta)_{n \geq 0}$. Exactly as in the proof of
 Theorem 3.8 we deduce from the $\beta_h$-smoothness of $J_h$ that

$$
J_h(\theta_{n+1}) \geq J_h(\theta_n) + \big(\nabla J_h(\theta_n)\big)^T(\theta_{n+1} - \theta_n) - \frac{\beta_h}{2}\|\theta_{n+1} - \theta_n\|^2, \quad \text{a.s.}
$$

 We continue with

$$
J_h(\theta_{n+1}) \geq J_h(\theta_n) + \eta_h\big(\nabla J_h(\theta_n)\big)^T \widehat{\nabla} J_h^{K_h}(\theta_n) - \frac{\beta_h \eta_h^2}{2}\|\widehat{\nabla} J_h^{K_h}(\theta_n)\|^2
$$

$$
= J_h(\theta_n) + \eta_h\big(\nabla J_h(\theta_n)\big)^T \nabla J_h(\theta_n) + \eta_h\big(\nabla J_h(\theta_n)\big)^T\big(\widehat{\nabla} J_h^{K_h}(\theta_n) - \nabla J_h(\theta_n)\big)
$$

$$
- \frac{\beta_h \eta_h^2}{2}\|\big(\widehat{\nabla} J_h^{K_h}(\theta_n) - \nabla J_h(\theta_n)\big) + \nabla J_h(\theta_n)\|^2.
$$

 We denote $\xi_n := \widehat{\nabla} J_h^{K_h}(\theta_n) - \nabla J_h(\theta_n)$ and rewrite the above inequality

$$
J_h(\theta_{n+1}) \geq J_h(\theta_n) + \eta_h\|\nabla J_h(\theta_n)\|^2 + \eta_h\langle \nabla J_h(\theta_n), \xi_n\rangle
$$

$$
- \frac{\beta_h \eta_h^2}{2}\big(\|\xi_n\|^2 + 2\langle \xi_n, \nabla J_h(\theta_n)\rangle + \|\nabla J_h(\theta_n)\|^2\big)
$$

$$
= J_h(\theta_n) + \big(\eta_h - \frac{\beta_h \eta_h^2}{2}\big)\|\nabla J_h(\theta_n)\|^2 + \big(\eta_h - \beta_h \eta_h^2\big)\langle \nabla J_h(\theta_n), \xi_n\rangle - \frac{\beta_h \eta_h^2}{2}\|\xi_n\|^2.
$$

Next, we take the conditional expectation on $\mathcal{F}_n$. Then with Lemma C.1, we obtain

$$\mathbb{E}\Big[J(\theta_{n+1})|\mathcal{F}_n\Big] \geq J(\theta_n) + \Big(\eta_h - \frac{\beta_h\eta_h^2}{2}\Big)\|\nabla J_h(\theta_n)\|^2 + \Big(\eta_h - \beta_h\eta_h^2\Big)\langle\nabla J(\theta_n), \mathbb{E}[\xi_n|\mathcal{F}_n]\rangle$$
$$- \frac{\beta_h\eta_h^2}{2}\mathbb{E}\big[\|\xi_n\|^2|\mathcal{F}_n\big]$$
$$\geq J(\theta_n) + \Big(\eta_h - \frac{\beta_h\eta_h^2}{2}\Big)\|\nabla J(\theta_n)\|^2 - \frac{\beta_h\eta_h^2 C_h}{2K_h^{(n)}}.$$

We take the expectation of this inequality on both sides under the event $\{n+1 \leq \tau\}$. Note that $\{n+1 \leq \tau\} = \{\tau \leq n\}^C$ is $\mathcal{F}_n$-measurable and that $\mathbf{1}_{\{n+1\leq\tau\}} \leq \mathbf{1}_{\{n\leq\tau\}}$ a.s., thus

$$\mathbb{E}\Big[(J_h^* - J_h(\theta_{n+1}))\mathbf{1}_{\{n+1\leq\tau\}}\Big]$$
$$= \mathbb{E}\Big[\mathbb{E}\Big[(J_h^* - J_h(\theta_{n+1}))|\mathcal{F}_n\Big]\mathbf{1}_{\{n+1\leq\tau\}}\Big]$$
$$\leq \mathbb{E}\Big[\Big(J_h^* - \mathbb{E}\Big[J_h(\theta_{n+1})|\mathcal{F}_n\Big]\Big)\mathbf{1}_{\{n\leq\tau\}}\Big]$$
$$\leq \mathbb{E}\Big[(J_h^* - J_h(\theta_n))\mathbf{1}_{\{n\leq\tau\}}\Big] - \Big(\eta_h - \frac{\beta_h\eta_h^2}{2}\Big)\mathbb{E}\Big[\|\nabla J_h(\theta_n)\|^2\mathbf{1}_{\{n\leq\tau\}}\Big] + \frac{\beta_h\eta_h^2 C_h}{2K_h^{(n)}}.$$

By Lemma 3.6 we have that $\|\nabla J_h(\theta_n)\|^2 \geq \min_{s\in\mathcal{S}}\pi^{\theta_n}(a^*(s|s))^2(J_h^* - J_h(\theta_n))^2$ almost surely, and by Lemma C.4 we have that $\min_{0\leq n\leq\tau}\min_{s\in\mathcal{S}}\pi^{\theta_n}(a^*(s|s))^2 \geq \frac{c_h}{2} > 0$ almost surely. Therefore,

$$\mathbb{E}\Big[(J_h^* - J_h(\theta_{n+1}))\mathbf{1}_{\{n+1\leq\tau\}}\Big]$$
$$\leq \mathbb{E}\Big[(J_h^* - J_h(\theta_n))\mathbf{1}_{\{n\leq\tau\}}\Big] - \Big(\eta_h - \frac{\beta_h\eta_h^2}{2}\Big)\mathbb{E}\Big[\min_{s\in\mathcal{S}}\pi^{\theta_n}(a^*(s|s))^2(J_h^* - J_h(\theta_n))^2\mathbf{1}_{\{n\leq\tau\}}\Big]$$
$$+ \frac{\beta_h\eta_h^2 C_h}{2K_h^{(n)}},$$
$$\leq \mathbb{E}\Big[(J_h^* - J_h(\theta_n))\mathbf{1}_{\{n\leq\tau\}}\Big] - \Big(\eta_h - \frac{\beta_h\eta_h^2}{2}\Big)\frac{c_h^2}{4}\mathbb{E}\Big[(J_h^* - J_h(\theta_n))\mathbf{1}_{\{n\leq\tau\}}\Big]^2 + \frac{\beta_h\eta_h^2 C_h}{2K_h^{(n)}},$$

where we used Jensen's inequality in the last step.

For $d_n := \mathbb{E}\Big[(J_h^* - J_h(\theta_n))\mathbf{1}_{\{n\leq\tau\}}\Big]$ we imply the recursive inequality

$$d_{n+1} \leq d_n - \Big(\eta_h - \frac{\beta_h\eta_h^2}{2}\Big)\frac{c_h^2}{4}d_n^2 + \frac{\beta_h\eta_h^2 C_h}{2K_h^{(n)}}.$$

Define $w := \Big(\eta_h - \frac{\beta_h\eta_h^2}{2}\Big)\frac{c_h^2}{4} > 0$ and $B = \frac{\beta_h\eta_h^2 C_h}{2} > 0$, then

$$d_{n+1} \leq d_n(1 - wd_n) + \frac{B}{K_h^{(n)}}$$

and by our choice of $\eta_h$,

$$K_h^{(n)} \geq \frac{9c_h^2 C_h}{32\beta_h^2 N_h^{\frac{3}{2}}}\Big(1 - \frac{1}{2\sqrt{N_h}}\Big)n^2 = \frac{9}{4}wBn^2,$$

Moreover, we have

$$\frac{4}{3w} = \frac{16\sqrt{N_h}\beta_h}{3(1 - \frac{1}{2\sqrt{N_h}})c_h^2}.$$

For $\beta_h = 2(H-h)R^*|\mathcal{A}|$, it holds that

$$d_1 \leq (H-h)R^* \leq \beta_h \leq \frac{4}{3w} \leq \frac{4}{3w \cdot 1},$$

because $c_h \leq 1$ and $\frac{1}{\sqrt{N_h}}(1 - \frac{1}{2\sqrt{N_h}}) < 1$ for all $N_h \geq 1$. Suppose the induction assumption $d_n \leq \frac{4}{3wn}$ holds true, then for $d_{n+1}$,

$$d_{n+1} \leq d_n - wd_n^2 + \frac{B}{K_h^{(n)}}.$$

The function $f(x) = x - wx^2$ is monotonically increasing in $[0, \frac{1}{2w}]$ and by induction assumption $d_n \leq \frac{1}{4wn} \leq \frac{1}{2w}$. So $d_n - wd_n^2 \leq \frac{4}{3wn}$ which implies

$$
\begin{aligned}
d_{n+1} &\leq d_n - wd_n^2 + \frac{B}{K_h^{(n)}} \\
&\leq \frac{4}{3wn} - \frac{16}{9wn^2} + \frac{B}{K_n} \\
&\leq \frac{4}{3wn} - \frac{16}{9wn^2} + \frac{4B}{9wBn^2} \\
&= \frac{4}{3wn} - \frac{12}{9wn^2} \\
&= \frac{4}{3w}\left(\frac{1}{n} - \frac{1}{n^2}\right) \\
&\leq \frac{4}{3w(n+1)},
\end{aligned}
$$

where we used that $K_h^{(n)} \geq \frac{9}{4}wBn^2$. We follow the claim

$$d_n \leq \frac{4}{3wn} = \frac{16\sqrt{N_h}\beta}{3(1 - \frac{1}{2\sqrt{N_h}})c_h^2 n}.$$

$\qquad\qquad\qquad\qquad\qquad\qquad\qquad\qquad\qquad\qquad\qquad\qquad\qquad\qquad\qquad\qquad\qquad\square$

**Lemma 4.3.** *Suppose $\mu_h(s) > 0$ for all $s \in \mathcal{S}_h$. Then, for any $\delta > 0$, we have $\mathbb{P}(\tau \leq n) < \delta$ if $K_h \geq \frac{16n^3 C_h}{\beta^2 c_h^2 \delta^2}$ and $\eta_h = \frac{1}{\sqrt{n}\beta_h}$.*

*Proof.* By the definition of $\tau$ we have

$$\mathbb{P}(\tau \leq n) = \mathbb{P}(\max_{0 \leq t \leq n} \|\theta_t - \bar{\theta}_t\| \geq \frac{c_h}{4}),$$

so we first study $\|\theta_t - \bar{\theta}_t\|$. We emphasize that Ding et al. (2022, Lemma 6.3) established a similar recursive inequality.

$$
\begin{aligned}
\|\theta_t - \bar{\theta}_t\| &= \|\theta_0 + \sum_{k=1}^{t-1} \eta_h \widehat{\nabla} J_h^{K_h}(\theta_k) - (\theta_0 + \sum_{k=1}^{l-1} \eta_h \nabla J_h(\bar{\theta}_k))\| \\
&\leq \sum_{k=1}^{t-1} \eta_h \|\widehat{\nabla} J_h^{K_h}(\theta_k) \nabla J_h(\bar{\theta}_k)\| \\
&\leq \eta_h \sum_{k=1}^{t-1} (\|\widehat{\nabla} J_h^{K_h}(\theta_k) - \nabla J_h(\theta_k)\| + \|\nabla J_h(\theta_k) - \nabla J_h(\bar{\theta}_k)\|).
\end{aligned}
$$

We define again $\xi_k = \widehat{\nabla} J_h^{K_h}(\theta_k) - \nabla J_h(\theta_k)$ and continue

$$
\begin{aligned}
\|\theta_t - \bar{\theta}_t\| &\leq \eta_h \sum_{k=1}^{t-1} (\|\xi_k\| + \beta_h \|\theta_k - \bar{\theta}_k\|) \\
&= \eta_h \sum_{k=1}^{t-1} \|\xi_k\| + \eta_h \beta_h \sum_{k=1}^{t-1} \|\theta_k - \bar{\theta}_k\|.
\end{aligned}
$$

809 Using this inequality sequentially leads to

$$
\|\theta_t - \bar{\theta}_t\| \leq \eta_h \sum_{k=1}^{t-1} \|\xi_k\| + \eta_h \beta_h \sum_{k=1}^{t-1} \|\theta_k - \bar{\theta}_k\|
$$

$$
\leq \eta_h \sum_{k=1}^{t-1} \|\xi_k\| + \eta_h \beta_h \sum_{k=1}^{t-2} \|\theta_k - \bar{\theta}_k\| + \eta_h \beta_h \Big( \eta_h \sum_{k=1}^{t-2} \|\xi_k\| + \eta_h \beta_h \sum_{k=1}^{t-2} \|\theta_k - \bar{\theta}_k\| \Big)
$$

$$
= \eta_h \sum_{k=1}^{t-1} \|\xi_k\| + \eta_h^2 \beta_h \sum_{k=1}^{t-2} \|\xi_k\| + (1 + \eta_h \beta_h) \eta_h \beta_h \sum_{k=1}^{t-2} \|\theta_k - \bar{\theta}_k\|
$$

$$
= \eta_h \|\xi_{t-1}\| + \eta_h (1 + \eta_h \beta_h) \sum_{k=1}^{t-2} \|\xi_k\| + (1 + \eta_h \beta_h) \eta_h \beta_h \sum_{k=1}^{t-2} \|\theta_k - \bar{\theta}_k\|
$$

$$
\leq \sum_{k=1}^{t-1} \eta_h (1 + \eta_h \beta_h)^{t-k-1} \|\xi_k\|.
$$

810 Applying Markov's inequality results in

$$
\mathbb{P}(\tau \leq n) = \mathbb{P}(\max_{0 \leq t \leq n} \|\theta_t - \bar{\theta}_t\| \geq \frac{c_h}{4})
$$

$$
\leq \mathbb{P}(\sum_{k=1}^{n-1} \eta_h (1 + \eta_h \beta_h)^{n-k-1} \|\xi_k\| \geq \frac{c_h}{4})
$$

$$
\leq \frac{4 \sum_{k=1}^{n-1} \eta_h (1 + \eta_h \beta_h)^{n-k-1} \mathbb{E}[\|\xi_k\|]}{c_h}
$$

$$
\leq \frac{4 n \eta_h (1 + \eta_h \beta_h)^{n-1} \sqrt{\frac{C_h}{K_h}}}{c_h},
$$

811 where in the last inequality $\mathbb{E}[\|\xi_k\|] \leq \sqrt{\mathbb{E}[\|\xi_k\|^2]} \leq \sqrt{\frac{C_h}{K_h}}$ by Jensen's inequality and Lemma C.1.

812 Now we plug in the choice of $\eta_h = \frac{1}{\sqrt{n}\beta_h}$,

$$
\mathbb{P}(\tau \leq n) \leq \frac{4n \frac{1}{\sqrt{n}\beta_h} (1 + \frac{1}{\sqrt{n}\beta_h}\beta_h)^{n-1} \sqrt{\frac{C_h}{K_h}}}{c_h}
$$

$$
= \frac{4\sqrt{n}(1 + \frac{1}{\sqrt{n}})^{n-1} \sqrt{C_h}}{\beta_h c_h \sqrt{K_h}}
$$

$$
\leq \frac{4\sqrt{n} n \sqrt{C_h}}{\beta_h c_h \sqrt{K_h}},
$$

813 where the last step is due to $f(x) = (1 + \frac{1}{\sqrt{x}})^{x-1} \leq x$ for all $x \geq 1$. We follow that $\mathbb{P}(\tau < n) < \delta$ if

$$
K_h \geq \frac{16 n^3 C_h}{\beta_h^2 c_h^2 \delta^2}.
$$

814 $\qquad\qquad\qquad\qquad\qquad\qquad\qquad\qquad\qquad\qquad\qquad\qquad\qquad\qquad\qquad\qquad$ □

815 **Theorem 4.4.** *Suppose the stochastic policy gradient updates are generated by* (9) *for arbitrary*
816 *initialization $\theta_0 \in \mathbb{R}^{d_h}$. Suppose that $\mu_h(s) > 0$ for all $s \in \mathcal{S}_h$ and choose for any $\delta, \epsilon > 0$,*

817     *(i) the number of training steps $N_h \geq \left( \frac{64 \beta_h}{3 \delta c_h^2 \epsilon} \right)^2$,*

818     *(ii) the step size $\eta_h = \frac{1}{\beta_h \sqrt{N_h}}$ and the batch size $K_h = \frac{64 N_h^3 C_h}{\beta^2 c_h^2 \delta^2}$.*

819  *Then,* $\mathbb{P}\big((J_h^* - J_h(\theta_{N_h})) \geq \epsilon\big) \leq \delta.$

820  *Proof.* We separate the probability using the stopping time $\tau$ and obtain

$$
\begin{aligned}
\mathbb{P}\Big((J_h^* - J_h(\theta_{N_h})) \geq \epsilon\Big) &\leq \mathbb{P}\Big(\{\tau \geq N_h\} \cap \{(J_h^* - J_h(\theta_{N_h})) \geq \epsilon\}\Big) \\
&\quad + \mathbb{P}\Big(\{\tau \leq N_h\} \cap \{(J_h^* - J_h(\theta_{N_h})) \geq \epsilon\}\Big) \\
&\leq \frac{\mathbb{E}\Big[(J_h^* - J_h(\theta_{N_h}))\mathbf{1}_{\{\tau \geq N_h\}}\Big]}{\epsilon} + \mathbb{P}(\tau \leq N_h) \\
&\leq \frac{1}{\epsilon}\frac{16\beta_h\sqrt{N_h}}{3(1 - \frac{1}{2\sqrt{N_h}})c_h^2 N_h} + \frac{\delta}{2} \\
&\leq \frac{\delta}{2} + \frac{\delta}{2} \\
&= \delta,
\end{aligned}
$$

821  where the second inequality it due to Lemma 4.2 and Lemma 4.3. The last inequality follows by our
822  choice of $N_h$:

$$
\frac{16\beta_h}{3\epsilon(1 - \frac{1}{2\sqrt{N_h}})c_h^2\sqrt{N_h}} \leq \frac{\delta}{2}
$$

823  for $N_h \geq \big(\frac{32\beta_h}{3\epsilon\delta c_h^2} + \frac{1}{2}\big)^2$, which is satisfied for $N_h \geq \big(\frac{64\beta_h}{3\epsilon\delta c_h^2}\big)^2$. Note further that we could use
824  Lemma 4.2 in the equation above with a constant batch size $K_h$, because

$$
\max\left\{\frac{9c_h^2 C_h}{32\beta_h^2 N_h^{\frac{3}{2}}}(1 - \frac{1}{2\sqrt{N_h}})n^2, \frac{16N_h^3 C_h}{\beta^2 c_h^2 \frac{\delta}{2}^2}\right\} = \frac{16N_h^3 C_h}{\beta^2 c_h^2 \frac{\delta}{2}^2},
$$

825  for all $n \leq N_h$, as $(1 - \frac{1}{2\sqrt{N_h}}) < 1$, $c_h < 1$ and $\frac{C_h}{\beta^2} < 1$. $\qquad\square$

## D  Proofs of Section 5

827  **Theorem 5.1.** *Assume that $\mu_h(s) > 0$ for all $h \in \mathcal{H}$, $s \in \mathcal{S}_h$. Let $\epsilon > 0$, the step size $\eta_h = \frac{1}{\beta_h}$ and*
828  *the batch size $N_h = \frac{4(H-h)HR^*|\mathcal{A}|}{c_h^2\epsilon}\big\|\frac{1}{\mu_h}\big\|_\infty$. Denote by $\hat{\pi}^* = (\pi^{\theta_0^{N_0}}, \ldots, \pi^{\theta_{H-1}^{N_{H-1}}})$ the final policy*
829  *from Algorithm 1, then for all $s \in \mathcal{S}_0$,*

$$
V_0^*(s) - V_0^{\hat{\pi}^*}(s) \leq \epsilon.
$$

830  *Proof.* First note that by our choice of the future policy $\tilde{\pi} = \hat{\pi}^*$ we have

$$
J_{h,s}(\theta_h^{(N_h)}) = V_h^{\hat{\pi}^*}(s). \tag{21}
$$

831  By Theorem 3.8 we obtain

$$
J_h^* - J_h(\theta_h^{(N_h)}) \leq \frac{4(H-h)R^*|\mathcal{A}|}{c_h^2 N_h}.
$$

832  For every $s \in \mathcal{S}_h$, denote by $\delta_s$ the dirac measure on state $s$, then

$$
\begin{aligned}
J_{h,s}^* - J_{h,s}(\theta_h^{(N_h)}) &= \sum_{s' \in \mathcal{S}_h} \mu_h(s')\frac{\delta_s(s')}{\mu_h(s')}J_{h,s}^* - J_{h,s}(\theta_h^{(N_h)}) \\
&\leq \Big\|\frac{1}{\mu_h}\Big\|_\infty (J_h^* - J_h(\theta_h^{(N_h)})) \tag{22} \\
&\leq \frac{4(H-h)R^*|\mathcal{A}|}{c_h^2 N_h}\Big\|\frac{1}{\mu_h}\Big\|_\infty,
\end{aligned}
$$

where $\left\|\frac{1}{\mu_h}\right\|_\infty = \max_{s\in\mathcal{S}_h} \frac{1}{\mu_h(s)} > 0$ by assumption. As $N_h = \frac{4(H-h)HR^*|\mathcal{A}|}{c_h^2\epsilon}\left\|\frac{1}{\mu_h}\right\|_\infty$, it holds that

$$J_{h,s}^* - J_{h,s}(\theta_h^{(N_h)}) \leq \frac{\epsilon}{H} \tag{23}$$

for every $s \in \mathcal{S}_h$. For $h = H-1$ it follows directly by (21) and the specialty of the last time point that for all $s \in \mathcal{S}_{H-1}$,

$$V_{H-1}^*(s) - V_{H-1}^{\hat{\pi}^*}(s) = J_{H-1,s}^* - J_{h,s}(\theta_h^{(N_h)}) \leq \frac{\epsilon}{H}.$$

Assume now that for all $s \in \mathcal{S}_h$,

$$V_h^*(s) - V_h^{\hat{\pi}^*}(s) \leq \frac{\epsilon(H-h)}{H}. \tag{24}$$

Then it holds for all $s \in \mathcal{S}_{h-1}$ that,

$$\begin{aligned}
J_{h-1,s}^* &= \max_{a\in\mathcal{A}_s}\left(r(s,a) + \sum_{s'\in\mathcal{S}_h} p(s'|s,a)V_h^*(s) - \sum_{s'\in\mathcal{S}_h} p(s'|s,a)(V_h^*(s) - V_h^{\hat{\pi}^*}(s))\right) \\
&\geq \max_{a\in\mathcal{A}_s}\left(r(s,a) + \sum_{s'\in\mathcal{S}_h} p(s'|s,a)V_h^*(s)\right) - \frac{\epsilon(H-h)}{H} \\
&= V_{h-1}^*(s) - \frac{\epsilon(H-h)}{H},
\end{aligned} \tag{25}$$

by the Bellman expectation equation for finite-time MDPs (Puterman (2005)). We close the backward induction using (21) such that for all $s \in \mathcal{S}_{h-1}$,

$$\begin{aligned}
V_{h-1}^*(s) - V_{h-1}^{\hat{\pi}^*}(s) &= V_{h-1}^*(s) - J_{h-1,s}^* + J_{h-1,s}^* - V_{h-1}^{\hat{\pi}^*}(s) \\
&\leq \frac{\epsilon(H-h)}{H} + \frac{\epsilon}{H} \\
&= \frac{\epsilon(H-(h-1))}{H}.
\end{aligned} \tag{26}$$

Finally, it holds for $h = 0$ and all $s \in \mathcal{S}_0$ that

$$V_0^*(s) - V_0^{\hat{\pi}^*}(s) \leq \epsilon.$$

$\square$

**Theorem 5.2.** *Assume that $\mu_h(s) > 0$ for all $h \in \mathcal{H}$, $s \in \mathcal{S}_h$. Let $\delta, \epsilon > 0$, the step size $\eta_h = \frac{1}{\beta_h N_h}$, number of training steps $N_h = \left(\frac{64\beta_h H^2\left\|\frac{1}{\mu_h}\right\|_\infty}{3\delta c_h^2\epsilon}\right)^2$ and the batch size $K_h = \frac{64N_h^2 H^2 C_h}{\beta_h c_h^2\delta^2}$. Denote by $\hat{\pi}^* = (\pi^{\theta_0^{N_0}}, \dots, \pi^{\theta_{H-1}^{N_{H-1}}})$ the final policy from Algorithm 2, then*

$$\mathbb{P}\left(\exists s \in \mathcal{S}_0 : V_0^*(s) - V_0^{\hat{\pi}^*}(s) \geq \epsilon\right) \leq \delta.$$

*Proof.* As in the exact gradient case (21) we have by our choice of the future policy $\tilde{\pi} = \hat{\pi}^*$ that

$$J_{h,s}(\theta_h^{(N_h)}) = V_h^{\hat{\pi}^*}(s). \tag{27}$$

By Theorem 4.4 we have that

$$\mathbb{P}\left(J_h^* - J_h(\theta_h^{(N_h)}) \geq \frac{\epsilon}{H\left\|\frac{1}{\mu_h}\right\|_\infty}\right) \leq \frac{\delta}{H},$$

by our choice of $N_h$, $\eta_h$ and $K_h$.

For every $s \in \mathcal{S}_h$, denote by $\delta_s$ the dirac measure on state $s$, then as in (22)

$$J_{h,s}^* - J_{h,s}(\theta_h^{(N_h)}) \leq \left\|\frac{1}{\mu_h}\right\|_\infty (J_h^* - J_h(\theta_h^{(N_h)})) \quad \text{a.s.}$$

849 Thus, for all $h \in \mathcal{H}$ it holds that

$$\mathbb{P}\Big(\exists s \in \mathcal{S}_h : J_{h,s}^* - J_{h,s}(\theta_h^{(N_h)}) \geq \frac{\epsilon}{H}\Big) \leq \mathbb{P}\Big(J_h^* - J_h(\theta_h^{(N_h)}) \geq \frac{\epsilon}{H\big\|\frac{1}{\mu_h}\big\|_\infty}\Big) \leq \frac{\delta}{H}. \qquad (28)$$

850 Define the event $A_h := \{J_{h,s}^* - J_{h,s}(\theta_h^{(N_h)}) < \frac{\epsilon}{H}, \forall s \in \mathcal{S}_h\}$. Then (29) is equivalent to $\mathbb{P}(A_h^C) \leq \frac{\delta}{H}$.

851 For $h = H - 1$ it follows directly with (27) and the special property of the last time point that

$$\mathbb{P}\Big(\exists s \in \mathcal{S}_h : V_{H-1}^*(s) - V_{H-1}^{\hat{\pi}^*}(s) \geq \frac{\epsilon}{H}\Big) = \mathbb{P}\Big(\exists s \in \mathcal{S}_h : J_{H-1,s}^* - J_{H-1,s}(\theta_h^{(N_h)}) \geq \frac{\epsilon}{H}\Big) \leq \frac{\delta}{H}.$$

852 We close the proof by induction. Assume for some $0 < h < H$ that

$$\mathbb{P}\Big(\exists s \in \mathcal{S}_h : V_h^*(s) - V_h^{\hat{\pi}^*}(s) \geq \frac{\epsilon(H-h)}{H}\Big) \leq \frac{\delta(H-h)}{H}. \qquad (29)$$

853 Define $B_h := \{V_h^*(s) - V_h^{\hat{\pi}^*}(s) < \frac{\epsilon(H-h)}{H}, \forall s \in \mathcal{S}_h\}$. Similar to (25), on the event $B_h$ it holds that

$$J_{h-1,s}^* = \max_{a \in \mathcal{A}_s}\Big(r(s,a) + \sum_{s' \in \mathcal{S}_h} p(s'|s,a)V_h^*(s) - \sum_{s' \in \mathcal{S}_h} p(s'|s,a)(V_h^*(s) - V_h^{\hat{\pi}^*}(s))\Big)$$

$$> \max_{a \in \mathcal{A}_s}\Big(r(s,a) + \sum_{s' \in \mathcal{S}_h} p(s'|s,a)V_h^*(s)\Big) - \frac{\epsilon(H-h)}{H}$$

$$= V_{h-1}^*(s) - \frac{\epsilon(H-h)}{H}.$$

854 We obtain on the event $A_{h-1} \cap B_h$ that (compare to (26))

$$V_{h-1}^*(s) - V_{h-1}^{\hat{\pi}^*}(s) = V_{h-1}^*(s) - J_{h-1,s}^* + J_{h-1,s}^* - V_{h-1}^{\hat{\pi}^*}(s)$$

$$< \frac{\epsilon(H-h)}{H} + \frac{\epsilon}{H}$$

$$= \frac{\epsilon(H-(h-1))}{H},$$

855 for every $s \in \mathcal{S}_{h-1}$. Hence, $A_{h-1} \cap B_h \subseteq B_{h-1}$. Finally, we close the induction by

$$\mathbb{P}\Big(\exists s \in \mathcal{S}_{h-1} : V_{h-1}^*(s) - V_{h-1}^{\hat{\pi}^*}(s) \geq \frac{\epsilon(H-(h-1))}{H}\Big)$$

$$= 1 - \mathbb{P}(B_{h-1}) \leq 1 - \mathbb{P}(A_{h-1} \cap B_h) = \mathbb{P}(A_{h-1}^C \cup B_h^C) \leq \mathbb{P}(A_{h-1}^C) + \mathbb{P}(B_h^C)$$

$$= \mathbb{P}\Big(\exists s \in \mathcal{S}_{h-1} : J_{h-1,s}^* - J_{h-1,s}(\theta_{h-1}^{(N_h-1)}) \geq \frac{\epsilon}{H}\Big)$$

$$\quad + \mathbb{P}\Big(\exists s \in \mathcal{S}_h : V_h^*(s) - V_h^{\hat{\pi}^*}(s) \geq \frac{\epsilon(H-h)}{H}\Big)$$

$$\leq \frac{\delta}{H} + \frac{\delta(H-h)}{H}$$

$$= \frac{\delta(H-(h-1))}{H}.$$

856 For $h = 0$ we have shown the claim

$$\mathbb{P}\Big(\exists s \in \mathcal{S}_0 : V_0^*(s) - V_0^{\hat{\pi}^*}(s) \geq \epsilon\Big) \leq \delta.$$

857 $\qquad\qquad\qquad\qquad\qquad\qquad\qquad\qquad\qquad\qquad\qquad\qquad\qquad\qquad\qquad\qquad\qquad\qquad$ □

# E   Proofs of Section 6

859 We denote by $\mathrm{GEOM}(p)$ the geometric distribution with parameter $p \in (0, 1]$.

860 Algorithm 3 states the construction of an approximate gradient $\widehat{\nabla}J^K(\theta) \approx \nabla J(\theta)$. Note that for batch
861 size $K = 1$, $\widehat{\nabla}J^1(\theta)$ is the estimator $\hat{\nabla}J(\theta)$ proposed in (Zhang et al., 2020, Eq. (3.6)). Furthermore,
862 it is important to highlight that the tabular softmax parametrization meets the assumptions made by
863 (Zhang et al., 2020, Ass. 3.1):

---
**Algorithm 3:** Estimate unbiased gradient for $\nabla J(\theta)$

---
**Data:** Let $\theta \in \Theta$.
**Result:** Approximate gradient $\widehat{\nabla} J^K(\theta)$
**for** $i = 1, \ldots, K$ **do**
$\quad$ Sample $T \sim \text{GEOM}(1-\gamma)$
$\quad$ Sample trajectory $(s_0^i, a_0^i, \ldots, s_T^i, a_T^i)$, s.t. $s_0 \sim \mu$, $a_t^i \sim \pi^\theta(\cdot|s_t^i)$, $s_{t+1}^i \sim p(\cdot|s_t^i, a_t^i)$
$\quad$ Sample $T' \sim \text{GEOM}(1-\gamma^{\frac{1}{2}})$
$\quad$ Set $\tilde{s}_0^i = s_T^i$, $\tilde{a}_0^i = a_T^i$
$\quad$ Sample trajectory $(\tilde{s}_1^i, \tilde{a}_1^i, \ldots, \tilde{s}_{T'}^i, \tilde{a}_{T'}^i)$, s.t. $\tilde{s}_t^i \sim p(\cdot|\tilde{s}_{t-1}^i, \tilde{a}_{t-1}^i)$, $\tilde{a}_t^i \sim \pi^\theta(\cdot|\tilde{s}_t^i)$
$\quad$ Set $\hat{Q}(s_T^i, a_T^i) := \sum_{t'=0}^{T'} \gamma^{\frac{t'}{2}} R(\tilde{s}_{t'}^i, \tilde{a}_{t'}^i)$.
**end**
Set $\widehat{\nabla} J^K(\theta) = \frac{1}{K} \sum_{i=1}^K \hat{Q}(s_T^i, a_T^i) \nabla \log(\pi^\theta(a_T^i|s_T^i))$.

---

- We assume that the rewards are bounded in $[0, R^*]$.

- The softmax parametrization is differentiable with respect to $\theta$, and $\nabla \log(\pi^\theta(a|s))$ exists. Moreover, by Lemma 3.4 we have that the gradient of $\log(\pi^\theta(a|s))$ is Lipschitz and that $\|\nabla \log(\pi^\theta(a|s))\|_2 \leq \sqrt{|\mathcal{A}|}$.

**Lemma E.1.** *The estimator $\widehat{\nabla} J^K(\theta)$ from algorithm 3 is an unbiased estimator of $\nabla J(\theta)$. Moreover, there exists $C > 0$ such that*

$$\mathbb{E}[\|\widehat{\nabla} J^K(\theta) - \nabla J(\theta)\|_2^2] \leq \frac{C}{K}.$$

*Proof.* By (Zhang et al., 2020, Theorem 4.3) we have that for $\theta \in \Theta$ deterministic

$$\mathbb{E}[\widehat{\nabla} J^1(\theta)] = \nabla J(\theta)$$

and

$$\|\nabla J(\theta)\|_2 \leq \frac{R^* B_\Theta}{(1-\gamma)^2}, \quad \|\widehat{\nabla} J^1(\theta)\|_2 \leq \frac{R^* B_\Theta}{(1-\gamma)(1-\gamma^{\frac{1}{2}})} \text{ a.s.,}$$

where $B_\Theta$ such that $\|\log(\pi^\theta(a|s))\|_2 \leq B_\Theta$. From the proof of Lemma 3.4 we have that $B_\Theta = \sqrt{|\mathcal{A}|}$. We deduce from Algorithm 3, that

$$E[\widehat{\nabla} J^K(\theta)] = \frac{1}{K} \sum_{i=1}^K \mathbb{E}[\widehat{\nabla} J^1(\theta)] = \nabla J(\theta).$$

For the variance we have

$$\mathbb{E}[\|\widehat{\nabla} J^K(\theta) - \nabla J(\theta)\|_2^2] \leq \frac{1}{K} \mathbb{E}[\|\widehat{\nabla} J^1(\theta) - \nabla J(\theta)\|_2^2]$$

$$\leq \frac{1}{K}\Big(\mathbb{E}[\|\widehat{\nabla} J^1(\theta)\|_2^2] + 2\mathbb{E}[\|\widehat{\nabla} J^1(\theta)\|_2]\|\nabla J(\theta)\|_2 + \|\nabla J(\theta)\|_2^2\Big)$$

$$\leq \frac{1}{K}\Big(\frac{(R^*)^2|\mathcal{A}|}{(1-\gamma)^2(1-\gamma^{\frac{1}{2}})^2} + 2\frac{R^*\sqrt{|\mathcal{A}|}}{(1-\gamma)(1-\gamma^{\frac{1}{2}})}\frac{R^*\sqrt{|\mathcal{A}|}}{(1-\gamma)^2} + \frac{(R^*)^2|\mathcal{A}|}{(1-\gamma)^4}\Big)$$

$$= \frac{(R^*)^2|\mathcal{A}|}{K}\Big(\frac{1}{(1-\gamma)^2(1-\gamma^{\frac{1}{2}})^2} + \frac{2}{(1-\gamma)^3(1-\gamma^{\frac{1}{2}})} + \frac{1}{(1-\gamma)^4}\Big).$$

Define $C = (R^*)^2|\mathcal{A}|\Big(\frac{1}{(1-\gamma)^2(1-\gamma^{\frac{1}{2}})^2} + \frac{2}{(1-\gamma)^3(1-\gamma^{\frac{1}{2}})} + \frac{1}{(1-\gamma)^4}\Big)$ proves the claim. $\quad\square$

Using this estimator we can formulate the REINFORCE algorithm as presented in Williams (1992) in Algorithm 4.

**Algorithm 4:** REINFORCE for discounted MDPs

**Result:** Approximate policy $\hat{\pi}^* \approx \pi^*$
Initialize $\theta_0 \in \mathbb{R}^{|\mathcal{S}||\mathcal{A}|}$
Choose step size $\eta$, number of training steps $N$ and batch size $K$
**for** $n = 0, \dots, N-1$ **do**
    Sample $\widehat{\nabla} J^K(\theta_n)$ as in Algorithm 3
    Set $\theta_{n+1} = \theta_n + \eta \widehat{\nabla} J^K(\theta_n)$
**end**
Set $\hat{\pi} = \pi^{\theta_N}$.

**Lemma E.2.**
$$\left\|\frac{\partial V^\pi(\mu)}{\partial \theta}\right\|_2 \geq \left\|\frac{d_\mu^{\pi^*}}{\mu}\right\|_\infty^{-1} \frac{\min_{s\in\mathcal{S}} \pi^\theta(a^*(s)|s)}{1-\gamma}(V^*(\mu) - V^{\pi^\theta}(\mu)).$$

878    *Proof.* We rewrite the norm of the gradient as follows
$$\left\|\frac{\partial V^\pi(\mu)}{\partial \theta}\right\|_2 = \left\|\sum_{s\in\mathcal{S}}\mu(s)\frac{\partial V^\pi(s)}{\partial \theta}\right\|_2$$
$$= \Big(\sum_{s'\in\mathcal{S}}\sum_{a'\in\mathcal{A}}\Big(\sum_{s\in\mathcal{S}}\mu(s)\frac{\partial V^\pi(s)}{\partial\theta(s',a')}\Big)^2\Big)^{\frac{1}{2}}$$
$$= \Big(\sum_{a'\in\mathcal{A}}\Big(\sum_{s\in\mathcal{S}}\mu(s)\frac{\partial V^\pi(s)}{\partial\theta(s,a')}\Big)^2\Big)^{\frac{1}{2}}$$

879    Note that we can interchange the derivative and the sum without further arguments because the state
880    space $\mathcal{S}$ is assumed to be finite. We continue as in the proof of (Mei et al., 2020, Lemma 8),
$$\left\|\frac{\partial V^\pi(\mu)}{\partial \theta}\right\|_2 \geq \Big|\sum_{s\in\mathcal{S}}\mu(s)\frac{\partial V^\pi(s)}{\partial\theta(s,a^*(s))}\Big|$$
$$= \Big|\frac{\partial V^\pi(\mu)}{\partial\theta(\cdot,a^*(\cdot))}\Big|$$
$$= \frac{1}{1-\gamma}\sum_{s\in\mathcal{S}}|d_\mu^{\pi^\theta}(s)\pi^\theta(a^*(s)|s)A^{\pi^\theta}(s,a^*(s))|$$
$$= \frac{1}{1-\gamma}\sum_{s\in\mathcal{S}}d_\mu^{\pi^\theta}(s)\pi^\theta(a^*(s)|s)|A^{\pi^\theta}(s,a^*(s))|$$
$$\geq \frac{1}{1-\gamma}\Big\|\frac{d_\mu^{\pi^*}}{d_\mu^{\pi^\theta}}\Big\|_\infty^{-1}\min_{s\in\mathcal{S}}\pi^\theta(a^*(s)|s)\sum_{s\in\mathcal{S}}d_\mu^{\pi^*}(s)A^{\pi^\theta}(s,a^*(s))$$
$$= \Big\|\frac{d_\mu^{\pi^*}}{d_\mu^{\pi^\theta}}\Big\|_\infty^{-1}\min_{s\in\mathcal{S}}\pi^\theta(a^*(s)|s)(V^*(\mu) - V^{\pi^\theta}(\mu)).$$

881    Furthermore, we can bound the distribution mismatch coefficient uniformly for all $\theta$,
$$d_\mu^{\pi^\theta}(s) \geq (1-\gamma)\mu(s),$$
882    by Mei et al. (2020, Thm. 4), such that $\left\|\frac{d_\mu^{\pi^*}}{d_\mu^{\pi^\theta}}\right\|_\infty^{-1} \leq (1-\gamma)^{-1}\left\|\frac{d_\mu^{\pi^*}}{\mu}\right\|_\infty^{-1}.$    □

883    Recall the definitions of $(\theta_n)_{n\geq 0}$ and $(\bar{\theta}_n)_{n\geq 0}$ from (11). We denote by $\mathcal{F}_n$ the natural filtration of
884    the process $(\theta_n)_{n\geq 0}$. With respect to this filtration we define the stopping time
$$\tau = \min\{n \geq 0 : \|\theta_n - \bar{\theta}_n\| \geq \frac{c}{4}\}, \tag{30}$$

where $c = \min_{n \geq 0} \min_{s \in \mathcal{S}} \pi^{\bar{\theta}_n}(a^*(s)|s) > 0$ by (Mei et al., 2020, Lemma 9) and $a^*(s)$ the optimal action of the deterministic optimal policy $\pi^*$.

**Lemma E.3.** *It holds almost surely that* $\min_{0 \leq n \leq \tau} \min_{s \in \mathcal{S}} \pi^{\theta_n}(a^*(s)|s) \geq \frac{c}{2}$ *is strictly positive.*

*Proof.* Due to the Lipschitz continuity of the softmax function the proof is line-by-line as in Lemma C.4. $\qquad\square$

**Lemma E.4.** *Suppose* $\mu(s) > 0$ *for all* $s \in \mathcal{S}$*, batch size* $K_n \geq \frac{9(1-\gamma)^4 c^2 C}{2048 N^{\frac{3}{2}}}(1 - \frac{1}{2\sqrt{N}})\left\|\frac{d_\mu^{\pi^*}}{\mu}\right\|_\infty^{-2} n^2$ *for some* $N \geq 1$ *and the step size* $\eta = \frac{(1-\gamma)^3}{8\sqrt{N}}$*, then*

$$\mathbb{E}\Big[(J^* - J(\theta_n))\mathbf{1}_{\{n \leq \tau\}}\Big] \leq \frac{128\sqrt{N}}{3(1 - \frac{1}{2\sqrt{N}})(1-\gamma)c^2 n}\left\|\frac{d_\mu^{\pi^*}}{\mu}\right\|_\infty^2.$$

*Proof.* We slightly modify the proof of Lemma 4.2 for finite-time MDPs. First, we deduce from the $\beta$-smoothness of $J$, with $\beta = \frac{8}{(1-\gamma)^3}$ (Mei et al. (2020), Agarwal et al. (2021)) that

$$J(\theta_{n+1}) \geq J(\theta_n) + \big(\eta - \frac{\beta\eta^2}{2}\big)\|\nabla J(\theta_n)\|^2 + \big(\eta - \beta\eta^2\big)\langle\nabla J(\theta_n), \xi_n\rangle - \frac{\beta\eta^2}{2}\|\xi_n\|^2,$$

where $\xi_n := \widehat{\nabla}J^K(\theta_n) - \nabla J(\theta_n)$. Next we take the conditional expectation on $\mathcal{F}_n$. Then by Lemma E.1 we obtain

$$\mathbb{E}\Big[J(\theta_{n+1})|\mathcal{F}_n\Big] \geq J(\theta_n) + \Big(\eta - \frac{\beta\eta^2}{2}\Big)\|\nabla J(\theta_n)\|^2 - \frac{\beta\eta^2 C}{2K_n}.$$

Subtracting this equation form $J^*$ and taking the expectation under the event $\{n+1 \leq \tau\}$ results in:

$$\mathbb{E}\Big[(J^* - J(\theta_{n+1}))\mathbf{1}_{\{n+1 \leq \tau\}}\Big]$$
$$\leq \mathbb{E}\Big[(J^* - J(\theta_n))\mathbf{1}_{\{n \leq \tau\}}\Big] - \Big(\eta - \frac{\beta\eta^2}{2}\Big)\mathbb{E}\Big[\|\nabla J(\theta_n)\|^2\mathbf{1}_{\{n \leq \tau\}}\Big] + \frac{\beta\eta^2 C}{2K_n}$$

With the PL-type inequality Lemma E.2 and $\min_{0 \leq n \leq \tau} \min_{s \in \mathcal{S}} \pi^{\theta_n}(a^*(s)|s) \geq \frac{c}{2}$ by Lemma E.3 we have

$$\mathbb{E}\Big[(J^* - J(\theta_{n+1}))\mathbf{1}_{\{n+1 \leq \tau\}}\Big]$$
$$\leq \mathbb{E}\Big[(J^* - J(\theta_n))\mathbf{1}_{\{n \leq \tau\}}\Big] - \Big(\eta - \frac{\beta\eta^2}{2}\Big)\frac{c^2}{4(1-\gamma)^2}\left\|\frac{d_\mu^{\pi^*}}{\mu}\right\|_\infty^{-2}\mathbb{E}\Big[(J^* - J(\theta_n))\mathbf{1}_{\{n \leq \tau\}}\Big]^2 + \frac{\beta\eta^2 C}{2K_n}.$$

For $d_n := \mathbb{E}\Big[(J^* - J(\theta_n))\mathbf{1}_{\{n \leq \tau\}}\Big]$ we obtain the recursive inequality

$$d_{n+1} \leq d_n - \Big(\eta - \frac{\beta\eta^2}{2}\Big)\frac{c^2}{4(1-\gamma)^2}\left\|\frac{d_\mu^{\pi^*}}{\mu}\right\|_\infty^{-2}d_n^2 + \frac{\beta\eta^2 C}{2K_n}.$$

We define $w := \Big(\eta - \frac{\beta\eta^2}{2}\Big)\frac{c^2}{4(1-\gamma)^2}\left\|\frac{d_\mu^{\pi^*}}{\mu}\right\|_\infty^{-2}$ and $B = \frac{\beta\eta^2 C}{2} > 0$ such that

$$d_{n+1} \leq d_n(1 - wd_n) + \frac{B}{K_n}.$$

Note that $w > 0$ by the assumption $\mu(s) > 0$ for all $s \in \mathcal{S}$. Then by our choice of $K_n$ and $\eta$ it holds that

$$K_n \geq \frac{9(1-\gamma)^4 c^2 C}{2048 N^{\frac{3}{2}}}(1 - \frac{1}{2\sqrt{N}})\left\|\frac{d_\mu^{\pi^*}}{\mu}\right\|_\infty^{-2} n^2$$
$$= \frac{9c^2 C}{32(1-\gamma)^2\beta^2 N^{\frac{3}{2}}}(1 - \frac{1}{2\sqrt{N}})\left\|\frac{d_\mu^{\pi^*}}{\mu}\right\|_\infty^{-2} n^2 = \frac{9}{4}wBn^2.$$

Furthermore, we have

$$\frac{4}{3w} = \frac{16\sqrt{N}\beta(1-\gamma)^2}{3(1-\frac{1}{2\sqrt{N}})c^2}\left\|\frac{d_\mu^{\pi^*}}{\mu}\right\|_\infty^2.$$

We obtain for $\beta = \frac{8}{(1-\gamma)^3}$ that

$$d_1 \leq \frac{1}{(1-\gamma)} \leq \beta(1-\gamma)^2 \leq \frac{4}{3w} \leq \frac{4}{3w\cdot 1},$$

because $c \leq 1$, $\left\|\frac{d_\mu^{\pi^*}}{\mu}\right\|_\infty^2 \geq 1$ and $\frac{1}{\sqrt{N}}(1-\frac{1}{2\sqrt{N}}) < 1$ for all $N \geq 1$.

Suppose the induction assumption $d_n \leq \frac{4}{3wn}$ holds true. The induction conclusion follows exactly as in the proof of Lemma 4.2: First, recall the recursive inequality

$$d_{n+1} \leq d_n - wd_n^2 + \frac{B}{K_n}.$$

The function $f(x) = x - wx^2$ is monotonically increasing in $[0, \frac{1}{2w}]$, and by induction assumption $d_n \leq \frac{1}{4wn} \leq \frac{1}{2w}$. Thus,

$$\begin{aligned}
d_{n+1} &\leq d_n - wd_n^2 + \frac{B}{K_n} \\
&\leq \frac{4}{3wn} - \frac{16}{9wn^2} + \frac{B}{K_n} \\
&\leq \frac{4}{3wn} - \frac{16}{9wn^2} + \frac{4B}{9wBn^2} \\
&= \frac{4}{3wn} - \frac{12}{9wn^2} \\
&= \frac{4}{3w}\left(\frac{1}{n} - \frac{1}{n^2}\right) \\
&\leq \frac{4}{3wn},
\end{aligned}$$

by the choice of $K_n \geq \frac{9}{4}wBn^2$. We deduce the claim

$$d_n \leq \frac{4}{3wn} = \frac{16\sqrt{N}\beta(1-\gamma)^2}{3(1-\frac{1}{2\sqrt{N}})c^2n}\left\|\frac{d_\mu^{\pi^*}}{\mu}\right\|_\infty^2 = \frac{128\sqrt{N}}{3(1-\frac{1}{2\sqrt{N}})(1-\gamma)c^2n}\left\|\frac{d_\mu^{\pi^*}}{\mu}\right\|_\infty^2.$$

$\square$

**Lemma E.5.** *Suppose $\mu(s) > 0$ for all $s \in S$. For any $N \geq 1$, if $\eta_h = \frac{(1-\gamma)^3}{\sqrt{N}8}$ and $K \geq \frac{N^3 C(1-\gamma)^6}{c^2\delta^2}$, then $\mathbb{P}(\tau \leq N) \leq \delta$.*

*Proof.* The proof follows line by line from the proof of Lemma 4.3 for the finite-time MDP. $\square$

**Theorem 6.1.** *Let $(\bar\theta_n)_{n\geq 0}$ and $(\theta_n)_{n\geq 0}$ be the (stochastic) policy gradient updates from (11) for arbitrary initial $\bar\theta_0 = \theta_0 \in \Theta$. Suppose $\mu(s) > 0$ for all $s \in S$ and choose for any $\delta, \epsilon > 0$,*

    *(i) the number of training steps $N \geq \left(\frac{258}{3\epsilon\delta c^2(1-\gamma)^3}\right)^2$,*

    *(ii) step size $\eta = \frac{(1-\gamma)^3}{8\sqrt{N}}$*

    *(iii) batch size $K = \max\left\{\frac{9(1-\gamma)^4c^2C}{2048}(\sqrt{N}-\frac{1}{2})\left\|\frac{d_\mu^{\pi^*}}{\mu}\right\|_\infty^{-2}, \frac{4(1-\gamma)^6N^3C}{c^2\delta^2}\right\}.$*

*Then, $\mathbb{P}\big((J^* - J(\theta_N)) \geq \epsilon\big) \leq \delta$, where $J^* = \sup_\theta J(\theta)$.*

921    *Proof.* We separate the probability using the stopping time $\tau$ and obtain

$$
\begin{aligned}
\mathbb{P}\Big((J^* - J(\theta_N)) \geq \epsilon\Big) &\leq \mathbb{P}\Big(\{\tau \geq N\} \cap \{(J^* - J(\theta_N)) \geq \epsilon\}\Big) \\
&\quad + \mathbb{P}\Big(\{\tau \leq N\} \cap \{(J^* - J(\theta_N)) \geq \epsilon\}\Big) \\
&\leq \frac{\mathbb{E}\Big[(J^* - J(\theta_N))\mathbf{1}_{\{\tau \geq N\}}\Big]}{\epsilon} + \mathbb{P}(\tau \leq N) \\
&\leq \frac{1}{\epsilon} \frac{128\sqrt{N}}{3(1-\gamma)(1 - \frac{1}{2\sqrt{N}})c^2 N} \Big\|\frac{d_\mu^{\pi^*}}{\mu}\Big\|_\infty^2 + \frac{\delta}{2} \\
&\leq \frac{\delta}{2} + \frac{\delta}{2} \\
&= \delta,
\end{aligned}
$$

922    where the second inequality holds due to Lemma E.4 and Lemma E.5. The last inequality follows by
923    our choice of $N$:

$$
\frac{128}{3\epsilon(1-\gamma)(1 - \frac{1}{2\sqrt{N}})c^2\sqrt{N}} \Big\|\frac{d_\mu^{\pi^*}}{\mu}\Big\|_\infty^2 \leq \frac{\delta}{2}
$$

924    if and only if $N \geq \big(\frac{256}{3\epsilon\delta c^2(1-\gamma)}\big\|\frac{d_\mu^{\pi^*}}{\mu}\big\|_\infty^2 + \frac{1}{2}\big)^2$, which is satisfied if $N \geq \big(\frac{258}{3\epsilon\delta c^2(1-\gamma)^3}\big)^2 \big\|\frac{d_\mu^{\pi^*}}{\mu}\big\|_\infty^4$.
925    Note that we can use Lemma E.4 in the equation above with a constant batch size, because

$$
\begin{aligned}
&\max\Big\{\frac{9(1-\gamma)^4 c^2 C}{2048 N^{\frac{3}{2}}}(1 - \frac{1}{2\sqrt{N}})\Big\|\frac{d_\mu^{\pi^*}}{\mu}\Big\|_\infty^{-2} n^2, \frac{(1-\gamma)^6 N^3 C}{c^2 \frac{\delta}{2}^2}\Big\} \\
&\leq \max\Big\{\frac{9(1-\gamma)^4 c^2 C}{2048}(\sqrt{N} - \frac{1}{2})\Big\|\frac{d_\mu^{\pi^*}}{\mu}\Big\|_\infty^{-2}, \frac{4(1-\gamma)^6 N^3 C}{c^2 \delta^2}\Big\},
\end{aligned}
$$

926    for all $n \leq N$.                                                       $\square$