# OpenReview forum: "Beyond Stationarity: Convergence Analysis of Stochastic Softmax Policy Gradient Methods"
_NeurIPS.cc/2023/Conference — Submitted to NeurIPS 2023_

### Official Review · Reviewer_h4oF · 2023-06-29

**Soundness:** 2 fair
**Presentation:** 3 good
**Contribution:** 1 poor
**Rating:** 3
**Confidence:** 3

**Summary:**

The paper presents a convergence (rate) analysis for a variant of policy-gradient learning of tabular-softmax policy models under finite-horizon MDP setting with discounting factor $\gamma=1$. In this variant, the policy parameters are updated in an epoch-by-epoch manner, from the last decision epoch at the horizon back to the first decision epoch. For each epoch, the learning is a vanilla (stochastic) gradient ascent process, with the objective function being the value obtained assuming fixed return in future epochs. It seems to me that most results in the paper are obtained following similar ideas with previous theoretical studies of policy gradient under discounted MDP settings (e.g. Agarwal et al., 2021, Mei et al., 2020).

**Strengths:**

I appreciate the general motivation of this work which aims at analyzing RL algorithms under undiscounted setting. In my opinion (and as the authors also pointed out in the paper), many performance bounds of RL algorithms derived under the discounted MDP setting crucially depend on the factor  $\frac{1}{1-\gamma}$, thus fail to accurately characterize the performance of RL algorithm under undiscounted setting. On the other hand, many real-world problems in practice are indeed undiscounted MDP problems. Therefore, directly analyzing RL algorithms under undiscounted MDP settings, or deriving results that do not degenerate when $\gamma$ approaches 1, is relevant and important research direction.

**Weaknesses:**

However, several major concerns make me hard to appreciate this particular work. Specifically,

1. The so-called policy gradient algorithm as analyzed in this paper is not really the standard policy gradient algorithm as generally perceived and applied -- the latter updates the parameters of states for all decision epochs simultaneously in each gradient step while the algorithm variant targeted in this paper will update parameters only for one epochs each time.
I think the standard co-updating strategy is much more practically relevant. For example, in RL training of language models, the problem is indeed an undiscounted MDP problem, and standard policy gradient methods (with PPO upgradation) are indeed widely used; see the InstructGPT paper (https://arxiv.org/abs/2203.02155) as an example. I thus have concern on the relevance and significance of the algorithm (not the problem setting) studied in this paper.

2. The epoch-wise updating strategy as assumed in this paper makes most of the analysis a straightforward adaptation of the standard analysis method used in the literature. Moreover, it seems to me that the epoch-wise learning problem analyzed in this paper is just a contextual bandit problem -- the paper assumes the policy $\tilde{\pi}$ for all future epochs is fixed, as well as assuming that the distribution $\mu_h$ for states encountered at the epoch is also fixed, but in this case we are essentially just maximizing an "augmented immediate reward (with future returns counted, which is fixed)" over actions conditioned on a context distribution. Since contextual bandit problem is a special case of discounted MDP (with arbitrary $\gamma$), I wonder if most results presented in this paper trivially hold given known results about discounted MDP.

3. I also have concerns on the authors' interpretation of their mathematical results. For example, the paper claims (as a main contribution) that Theorem 3.8 derives a linear performance bound to the horizon H. However, the bound contains the factor c_h, and it's not clear if c_h itself can be a function of H (or of H-h). c_h can be considered a "constant" only with respect to n, not to H. Moreover, Theorem 3.8 only gives a per-epoch performance while previous results gives the overall performance. My understanding is that when we consider the convergence of epoch 0 (which is what we eventually care), we would need to count the time spent on epoch 1 ~ H-1 too, which would bring in another factor of H to the bound?

4. Similarly, throughout the paper it's assumed that the state distribution $\mu_h$ is given for all epoch h, which is not a reasonable assumption in my opinion because except for $\mu_0$, the state distributions for all other epochs actually depend on the algorithm. This is in sharp contrast to the previous papers which only assume that the *initial* state distribution $\mu_0$ is given (which is indeed given). Again, this $\mu_h$ is considered a "constant" when interpreting the bound derived in Theorem 5.2, although $\mu_h$ (for h>0) may heavily depend on structures both in the problem and in the algorithm, in a highly non-trivial manner.

5. The majority of the paper assumes that the reward is nonnegative, which is a bit of a limitation, given that the sign of the rewards can greatly affects the performance of policy gradient.

**Questions:**

(A) You argued that "*reducing the problem to stationary policies is inadequate for finite-time MDPs, and a new policy must be trained recursively at each time step*" (Line 50). I am not sure that this assertion is true. The so-called non-stationary policy $\pi$ (defined at Line 104) is exactly a stationary policy over the entire state space (note that you have defined the state space $S$ as the union of the disjoint epoch-wise state spaces; see Line 100). Since the policy $\pi$ at Line 104 is just an ordinary stationary policy over $S$, the standard policy gradient algorithm can be applied, and we don't have to train it recursively. Since this disagreement directly leads to my major concern 1 as stated above, I am open to hear your rebuttal if I am misunderstanding something here.

(B) Line 154: It seems to me that $J_h$ depends not only on $\theta$ but also on $\mu_h$ and $\tilde{\pi}$. However, the notation here is hiding this fact. This in turn hide some subtle issues in the analysis later. For example, explicitly writing $J_h(\theta, \mu_h, \tilde{\pi})$ would make it clear that in Algorithm 1 the $\mu_h$ is left undetermined.

(C) Line 263: What is $C_h$? I originally thought this is a typo for $c_h$, until I found that in Theorem 5.2 both $C_h$ and $c_h$ show up in the definition of $K_h$.

**Limitations:**

The paper well discussed the limitations of this work in Section 7.

---

> ### Author Rebuttal · Authors · 2023-08-09
>
> Dear reviewer, thank you very much for your careful and constructive review which we appreciate a lot!
>
> **Your practical concern**
>
> There is an increasing gap between (extremely) successful practical applications of RL and solid foundations. While the approach of training all policies at once is used in practise (and works) there is limited theoretical understanding. In contrast our approach seems unnatural but allows a very clean analysis. Following your criticism we tried to fill the gap, and believe to understand what is happening.
>
> The reason we introduce our approach is dynamic programming (=backwards induction for finite MDPs). There is another way to see why the approach makes a lot of sense. The policy gradient theorem (Thm 2.2) shows that estimation errors are traced backwards by the Q-function. If the policy at later epochs is poorly estimated, the policy gradient at earlier epochs estimates poorly. Already from the gradient representation it is perfectly reasonable to first concentrate on later epochs, then concentrate on earlier epochs. We see our approach and the practically approach as two extreme algorithms. Now we argue that, in fact, their convergence properties are very similar.  In the main rebuttal we added graphs for both extremal approaches in the very simple MDP problem of optimally stopping when throwing a dice 5 times. There both extremal approaches perform almost equally well. Our approach is slightly better if a given accuracy is to be achieved.
>
> Why do we think our analysis is important also for the simultaneous algorithm? From our point of view it becomes quite clear how the simultaneous PG approach should be analyzed. Ignoring the training of earlier epochs, later epochs can be handled as we did. At steps suggested by our estimates the focus shifts towards an earlier epoch, the currently estimated parameter is considered as new $\theta_h^{(0)}$. We have a clear view how to proceed to reduce simultaneous PG to our backwards PG with those $\theta_h^{(0)}$ but will not be able to finish the analysis in time for this article. Our current analysis shows how many steps are needed in order to achieve a given accuracy, improving accuracy by one order forces to increase time-steps by factor 10. In fact, the simulation example shows that the estimates are pretty tight. Such an insight extended to the simultaneous algorithm would be very interesting in applications.
>
> From a practical point of view we believe that our study can be very beneficial. Algorithms should use replay buffers in a way that first focus stronger on later policies instead of using all epochs equally. We also believe that there will be combinations of both extremal algorithms that perform better. For instance, first passing backwards with our algorithm to obtain a reasonably good approximation with relatively few samples followed by training all policies at once.
>
> **Questions**
>
> (A) There is indeed a miss-understanding here. We defined the state space as the union of (possibly) time-dependent state spaces but want to point out that they do not need to be disjoint (and typically are equal). Given this setting a non-stationary policy is required such that given the same state in different epochs we are allowed to take different optimal actions. For example, in the simple optimal-stopping problem for the dice it is clear that the stopping-action strongly depends on time. To be more precise, having 5 tries at time 4 the optimal policy stops if and only if the dice shows 4,5,6, whereas at time 3 the optimal policy stops at 5, 6 only.
>
> (B) You are right, thank you. We will change the notation. For your concern about $\mu_h$ please see the later comment.
>
> (C) $C_h$ is defined in Lem C.1 and is the bound on the variance of the stochastic estimator of the gradient. We will change the notation of that constant.
>
> **About $\mu_h$**
>
> In cases where starting at any time of the MDP is impossible you can always choose a uniform distribution over the action space to ensure that every possible state in the MDP in reached with positive probability. Hence, choosing a uniform policy until epoch $h$ and just assuming $\mu_0$ strictly positive results in a "start distribution" $\mu_h$ which is strictly positive in every reachable state at $h$. Using this, Thm 5.1 and 5.2 can be straightforward generalized to only assuming $\mu_0$ strictly positive.
>
> **About contextual bandits**
>
> Somehow all finite-time MDP problems could be interpreted as a nested sequence of contextual bandit problems as dynamic programming suggests the backwards iteration we exploit. Since the dependencies of model parameters in policy gradient estimates (such as Mei2020) are not sufficiently explicit, we believe this point of view probably leads nowhere. Moreover, we emphasize that our main contribution, the analysis of **stochastic** policy gradient, is completely new and also not considered in previous work.
>
> **About interpretation of $c_h$**
>
> Thank you for pointing out that $c_h$ generally depends on $H$. This is true and, in general, we should not speak of linear dependence of the error bound. In fact, also in Mei2020 a similar constant depends on $\gamma$ without mentioning the dependence.
> Your concern should be investigated in more detail, also in the articles that appeared for discounted MDPs. For the second concern we refer you to Sec 5, where we analyzed the error over time. We indeed observe that an additional dependence on $H$ occurs when considering the whole algorithm. You can find the second $H$ in the rates in Thm 5.1 and 5.2 This leads to an overall dependence of $H^2$ (plus the $c_h$ of course).
>
> **About non-negative rewards**
>
> Positive rewards are considered wlog and are typical in the analysis of policy gradient algorithms. Using the standard base-line trick for policy gradient theorems the results can be extended to bounded rewards. We will add a comment in our manuscript.

---

> > ### Comment · Reviewer_h4oF · 2023-08-16
> >
> > Thanks for the rebuttal.
> >
> > > the approach of training all policies at once is used in practise (and works) there is limited theoretical understanding
> >
> > The "simultaneous" version of policy gradient is, in my opinion, really the algorithm that most people refer to when talking about "policy gradient", *not only in practice, but also in most theory works*. The algorithm is perhaps not analyzed in the undiscounted setting yet, but this is exactly why we *should* analyze it under this setting, right? I don't see why the fact that "the standard algorithm is not analyzed" motivates us to turn to analyze another algorithm.
> >
> >
> > > our approach seems unnatural but allows a very clean analysis
> >
> > As I mentioned in Concern 2, I doubt if the epoch-by-epoch setting as you assumed has made the analysis a bit too easy. Given fixed future payoffs, the problem of finding optimal action *for a single epoch* is (just) a contextual bandit problem for which performance of policy gradient can be easily derived (because contextual bandit is a special case of discounted MDP). So, while I agree that your algorithm setting allows easier analysis, I think this makes your paper weak, actually.
> >
> >
> > >  There is indeed a miss-understanding here
> >
> > You are right that I shouldn't say "S_h are *disjoint* subsets", which I actually mean they are "*separate* subsets", but this correction doesn't touch the main point. My point here is that, the simultaneous-update version of policy gradient is also perfectly applicable in your problem setting because the so-called non-stationary policy that you are analyzing is only non-stationary with respect to the epoch-wise state spaces, but *is* a stationary policy with respect to the state space $\mathcal{S}$.
> >
> >
> > > $c_h$ generally depends on $H$
> >
> > I feel $c_h$ might also crucially depend on $\mu_h$. A positive $\mu_h$ can only guarantee convergence in the limit, but it feels unlikely that the speed of the convergence is agnostic to the specific form of $\mu_h$.
> >
> >
> > > From our point of view it becomes quite clear how the simultaneous PG approach should be analyzed
> >
> > It is great to hear that you got inspired about analyzing the standard policy gradient. I agree that it would be quite interesting if a paper can show that the epoch-by-epoch variant is better than standard PG, either through systematic experiments or through mathematical analysis. It's interesting especially because I personally and intuitively don't feel it true :)
> >
> > In any case, I think the existing evidences you presented, including the rebuttal PDF, has not really established the above. I encourage you to keep working in this direction, though!

---

> > > ### Author Response · Authors · 2023-08-17
> > >
> > > Thanks for the good discussion! Though it might go a bit far we carried out the proofs you asked for to assure you we did not try to take a theory shortcut by not studying the (simultaneous) PG algorithm.
> > >
> > > > ...interesting if a paper can show that the epoch-by-epoch variant is better than standard PG...especially because I ... don't feel it true :)
> > >
> > > Challenge accepted. We did the Maths. In short: the provable bounds for backwards PG are indeed more intriguing than simultaneous PG. It scales better in $H$ and even better, the disturbing (and possibly huge) model-based constants $c_h$ can be made to disapear for backwards PG if properly initiated (not for simultaneous PG). This confirms our simple simulation example.
> > >
> > > **Two algorithms for non-stationary finite-time MDP**
> > >
> > > a) First our algorithm. You are right, we now understood your thoughts. It can be interpreted as a concatenation of contextual bandits, this is our Sec 3. Sec 3 is similar to Mei2020 but still needs some extra analysis, e.g. the crucial dependence on the time horizon $H-h$ (obtained in the smoothness) would not be visible in Mei2020 by seeing a contextual bandit as MDP with $\gamma=0$. (But please keep in mind, our major contribution is the stochastic case of later sections!)
> > >
> > > b) Second, what you call the PG Algorithm: artificially use a stationary policy by adding the time coordinate to the state-space, then considering an undiscounted stationary MDP with finite time-horizon. This leads to a stationary softmax policy with $H |\mathcal{S}| |\mathcal{A}|$ many parameters. We guess this is what you meant, the state-space is artificially made disjoint.
> > >
> > > **The Maths**
> > >
> > > We did the analysis for simultaneous PG to finally convince you, that our viewpoint is indeed worth looking at and that your personal feelings might not be completely true :)
> > >
> > > The analysis of "standard PG" can be proved with methods similar to Mei2020 (or other recent articles). Calculating the smoothness constant for the artificial undiscounted MDP then adapting the PL-inequality and putting both together gives the number of gradient iterations $$N = \frac{2 H^5 R^\ast (2-\frac{1}{|\mathcal{A}|})||\mathcal{S}|}{c^2 \epsilon} \Big\lVert \frac{d_\mu^{\pi^\ast}}{\mu}\Big\rVert_\infty^2$$to achieve an error $V^\ast(\mu) - V^{\hat{\pi}^\ast}(\mu)\leq \epsilon$. The $H^5$ is not surprising, you can compare to known results for discounted MDPs and then keep in mind that $H$ should replace $1/(1-\gamma)$. In fact, a discounted MDP can be seen as an undiscounted one terminated at an $Geo(\gamma)$ time which has mean $1/(1-\gamma)$. Thus, stopping at $H$ should give $H$ instead of $1/(1-\gamma)$.
> > >
> > > In comparison, the total number of training steps from our Thm 5.1 to achieve the same error is$$N= \sum_{h=0}^{H-1} N_h = \sum_{h=0}^{H-1} \frac{4(H-h)H R^*|\mathcal{A}|}{ c_h^2 \epsilon } \Big\lVert  \frac{1}{\mu_h}\Big\rVert_\infty.$$Note that we train $|\mathcal{S}||\mathcal{A}|$ many parameters in every $N_h$ such that $N |\mathcal{S}||\mathcal{A}|$ many partial derivatives are considered. The second viewpoint involves $H|\mathcal{S}||\mathcal{A}|$ many parameters in every training step and needs to evaluate $N H |\mathcal{S}||\mathcal{A}|$ many partial derivatives in total.
> > >
> > > **Comparison**
> > >
> > > a) the model dependent constants: Thinking about our backward inductive algorithm as a concatenation of contextual bandits, we now reaslised that the model-based constant $c_h$ simplifies to $\frac{1}{|\mathcal{A}|}$ **if we initialise softmax uniformly**. Mei2020 showed this already for bandits (in Prop 2) and this can also be transferred to contextual bandits. This is a bandit feature, i.e. using contextual bandits might not be that stupid after all. For the simultaneous training one looses this advantage and cannot get rid of the unknown $c$.
> > >
> > > b) The (non)dependence on $c$ is a clear advantage. A disadvantage is, as you already spotted, $\mu_h$. Our approach was motivated partially by optimal stopping approaches such as Becker et al. "Deep optimal stopping", JMLR 20 (2019)  where backwards induction appear. In that case (see our simulation) one would start in $\mu_h$ uniform, i.e. yields $$N= \frac{4H^3 R^*|\mathcal{A}|^3 |\mathcal{S}|}{\epsilon }$$gradient steps, which is pretty nice and a much better guarantee than we have for simultaneous PG. Of course, if one cannot start at later times, then the constant matters just as the $c$ constant matters for simultaneous PG.
> > >
> > > Perhaps you believe that that there is just an important point about the backwards version. We think it is nice (hopefully important) to see clearly how to train different epochs differently.
> > >
> > > Analysing simultaneous PG was not the purpose of this article, we focused on extending PG to SPG in the spirit of the usual extension of GD to SGD. We feel a conference contribution with the analysis of new PG scheme, standard PG, REINFORCE might be a bit heavy, but we are happy to add the proofs for simultaneous PG into the appendix.

---

> > > > ### Comment · Reviewer_h4oF · 2023-08-18
> > > >
> > > > Thanks for the follow-up rebuttal.
> > > >
> > > > > It can be interpreted as a concatenation of contextual bandits, this is our Sec 3 ... But please keep in mind, our major contribution is the stochastic case of later sections
> > > >
> > > > To my understanding, the main results in Section 4 (i.e. all lemmas and theorems in this section) are talking only about per-epoch performance too. Within an epoch your algorithm is (just) solving a contextual bandit problem, which is a degenerate case of discounted MDP, for which the performance of PG, *for both exact and stochastic*, are well known.
> > > >
> > > > > what you call the PG Algorithm: artificially use a stationary policy by adding the time coordinate to the state-space
> > > >
> > > > This "artificially constructed" policy is exactly equivalent to the policy you are analyzing in the paper. In your policy formulation (at line 104), the subscript $h$ in the sub-policies $\pi_h$ is effectively serving as the time index. This policy has $\sum_h |S_h||A|$ parameters, no matter if you call it stationary or non-stationary. The standard PG is not only perfectly applicable to this policy, applying standard PG on this policy is an *extremely* widely adopted practice in the real world (when dealing with episodic RL problems).
> > > >
> > > > Note: the above is a revised comment, because I was wrong in my original post by saying that the time index is not needed -- what I want to stress is that, to apply standard PG, there is no need to add additional parameters to the policy model.
> > > >
> > > >
> > > >
> > > > >  it might go a bit far we carried out the proofs you asked for ... Analysing simultaneous PG was not the purpose of this article
> > > >
> > > > I agree that it goes a bit far for you to prove for the standard PG, and actually, also for me and other reviewers to review such a proof,through the discussion period. In fact, I was not asking you to do this. I am giving review feedbacks to your original submission. I fully noticed that analyzing standard PG is not your purpose in the paper, and I guess I am challenging this purpose.
> > > >
> > > > In any case, I do hope you consider my (negative) feedbacks constructive, as I believe analyzing PG under undiscounted setting is an important research direction, as I've pointed out in the review comment.

---

> > > > > ### Author Response · Authors · 2023-08-18
> > > > >
> > > > > Thanks again for the very constructive discussion we really appreciate a lot!
> > > > >
> > > > > > ... discounted MDP, for which the performance of PG, *for both exact and stochastic*, are well known.
> > > > >
> > > > > We are not aware of an analysis of tabular softmax PG for discounted MDPs (without regularisation) in the *stochastic* setting. And papers dealing with regularisation only prove convergence to the regularised optima. Most papers such as Mei2020 do not consider the stochastic case. There is a number of partial results in the stochastic setting (of which several are incorrect) but none applicable in our case. The point is, that there is no guarantee for the $c_h =\inf_n \min_s \pi^{\theta_n}(a^\ast(s)|s)$ to be a constant or more precise to be bounded away from zero, when $(\theta_n)$ is a stochastic trajectory. In the stochastic setting this is tricky. We are not aware of such an analysis in the discounted setting for PG without regularisation. This is why we do carry out our analysis also for the discounted case, see Sec 6.
> > > > >
> > > > > If we are wrong (RL literature moves really fast), we kindly ask for a reference for an analysis as ours.
> > > > >
> > > > > > to apply standard PG, there is no need to add additional parameters to the policy model.
> > > > >
> > > > > Correct! The numbers of parameters of both parametrisations is the same. But there are different ways of training the parameters: Either brute force train all of them at once (this is typically used, as you say, but ignores the structure of the problem). Or using the information that dynamic programming suggests to train them recursively with the right non-constant number of training steps that comes from the L-smoothness of contextual bandits from Sec 3 (keeping in mind that the episode to episode  differ from contextual bandits as their rewards are rewards to go that get smaller for later epochs).
> > > > >
> > > > > The "artificial enlargement" is nothing but a mathematical trick to allow the analysis using stationary policies and, luckily, the analysis of discounted MDPs can be extended to that very very specific MDP without discounting (see last answer). It is not surprising that using structural information at least in theory leads to more efficient algorithms. This is what we tried to show in our previous answer, comparing both ways of training.
> > > > >
> > > > > > I fully noticed that analyzing standard PG is not your purpose in the paper, and I guess I am challenging this purpose.
> > > > >
> > > > > The analysis of standard PG in the followup rebuttal was just to convince you that our algorithm is (at least in theory) more efficient than brute force training of all parameters at once. One should be aware that training all parameters at once ignores all the advantage gained from the dynamic programming and throws away most of the structure of the problem.
> > > > >
> > > > > Although there might be good reasons to use a simple approach in practice, we believe that our duty as scientists is to develop theoretical improvements that might eventually be adopted in one way or another in practice.

---

> > > > > > ### Author Response · Authors · 2023-08-18
> > > > > > **Summary from our point of view**
> > > > > >
> > > > > > The discussion was intense but very constructive, thanks again. Here is the summary from our point of view.
> > > > > >
> > > > > > Practitioners use a brute force approach to train all parameters at once which is possible but ignores the very nature of the problem. The intrinsic nature of the problem is a contextual bandit problem (as you realised, we did not interpret it that way), with a non-trivial combination of bandits. This is nothing bad, this is actually very good! Using that intrinsic nature better algorithms can be designed (not surprising as it uses the true structure of the problem).
> > > > > >
> > > > > > In the paper we proposed and analysed the most natural one, for our discussion we compared it in the deterministic setting to the brute force simultaneous algorithm. The future will show how the theoretical understanding of the very nature of the problem problem might positively influence algorithms used in practice as well. It may or not, but that's typically the burden of theory papers.

---

### Official Review · Reviewer_jcRK · 2023-07-06

**Soundness:** 3 good
**Presentation:** 3 good
**Contribution:** 2 fair
**Rating:** 6
**Confidence:** 2

**Summary:**

This paper proves asymptotic convergence and convergence rate of (stochastic) policy gradient descent to global optimum for un-discounted finite time Markov decision process (MDP). For the deterministic version, at each decision time, they show the error bound depends linearly on the remaining time step. For the stochastic version, they derived probability bounds. They extend their analysis to REINFORCE on discounted MDP.

**Strengths:**

The presented results are sound and complete. Both convergence and convergence rate of policy gradient methods with softmax policy are established for the finite-time MDPs.


**Weaknesses:**

The main argument seems to be based on Mei et al (2020) with some modifications for un-discounted finite time MDPs case. I did not go over the details of proofs in appendix. It is hard  to evaluate the significance and the novelty of the contributions.

**Questions:**

See wekness

**Limitations:**

See weakness

---

> ### Author Rebuttal · Authors · 2023-08-09
>
> Dear reviewer, thank you very much for taking the time to review our article and the overall positive feedback!
>
>
> In order to answer the comment about your concerns about significance and novelty, in the following we will explain in more details the two main contributions of this article. One contribution, as you observed correctly, is in the spirit of Mei2020. However, let us emphasize that the main contribution is a novel analysis of the stochastic policy gradient method.
>
> **1. Contributions in the deterministic setting**
>
> The first contribution is a dynamic programming inspired version of policy gradient descent for finite-time MDPs. In fact, in practice PG is sometimes used differently, tuning all policies at once without taking into account that training errors of later times propagate backwards according to the policy gradient formula for finite-time MDPs (see Thm 2.2). It seems that our analysis is the first to give a rigorous analysis in this direction. The analysis of algorithms more common in practice will most likely be a reduction to our results, we give further discussion in the main rebuttal. You are right that our analysis in the deterministic case is based on Mei2020. Please note again that the analysis in Mei2020 crucially depends on the discount factor $\frac{1}{1-\gamma}$ and fails to transfer straight forwardly to undiscounted settings by choosing $\gamma =1$ or for cases where $\gamma$ approaches $1$.
>
>
> **2. Contributions in the stochastic setting**
>
> The second main contribution is completely novel. We show how to use techniques from stochastic approximation theory, in particular from the analysis of stochastic gradient descent methods, to extend the deterministic case to the sample based stochastic case. In contrast to the deterministic policy gradient analyzed by Mei2020 and others (under the strong assumption of exact knowledge of gradients) the stochastic policy gradient is very much different from stochastic gradient descent schemes. The reason is that samples are not iid, and the distributions of samples change in every step of the iteration. Proving that stochastic PG can be shown to work (in both undiscounted and discounted settings) is the main contribution of this article.

---

> > ### Comment · Reviewer_jcRK · 2023-08-21
> >
> > I thank the authors for their clarification. I will keep my score.

---

### Official Review · Reviewer_3eeX · 2023-07-08

**Soundness:** 4 excellent
**Presentation:** 4 excellent
**Contribution:** 3 good
**Rating:** 7
**Confidence:** 3

**Summary:**

This paper studies the convergence properties of stochastic policy gradient methods for finite-state MDPs for finite horizon problems with un-discounted optimality criteria. The convergence relies on the development of a weak PL condition. In the second part of the paper, the authors then extend their convergence analysis to the setting where the policy gradient is not available exactly and only a stochastic version of it can be used.

**Strengths:**

The paper is well written and easy to follow. In particular, I find Section 4, where the convergence analysis is generalized to the stochastic setting strong and practical useful. Proofs are derived very clearly and detailed with I appreciate.

**Weaknesses:**

1) Numerical example missing: In my opinion, it would be interesting to include a numerical example to the paper. In particular it would be interesting to see how tight the complexity bounds of Theorem 4.4 are on a concrete example.

2) Finite state/action spaces often are restrictive: What can you say about the results being generalised to continuous state and action spaces? Maybe for a linear system to start with.

3) How relevant are un-discounted problems? Could you add a detailed motivation, why these problems are practically relevant?


**Questions:**

1) Could you clarify better which results are new and which are existing results. For example, I don’t understand if Theorem 2.2 is an existing result or if it is new.

2) In Equation (5), I don’t understand the dimension of $\theta$. I think there is something wrong here.

3) In Lemma 2.1, why is the superscript of the expectation operator $\pi_{(h)}$ and not $\pi$ ?


**Limitations:**

Without Section 4, the work would have been limited as the gradients typically are not known exactly. But thanks to the results in Section 4, I think the paper is rather complete and I do not see major limitations.

---

> ### Author Rebuttal · Authors · 2023-08-09
>
> Dear reviewer, thank you very much for taking the time to review our article and the overall positive feedback!
>
> **1. Numerical examples**
>
> In the main rebuttal we added graphs for the deterministic analysis in the very simple MDP problem of optimally stopping when throwing a dice 5 times. In this very (!) simple example we can see the $\frac{1}{n}$ convergence of Theorem 3.8. We won't be able to carry out a full analysis also for the stochastic setting in time. Unfortunately, we also not expect the complexity analysis in Theorem 4.4 to be tight. This is due to the fix step sizes and the resulting sufficient large sample sizes to guarantee $c_h>0$.
> Let us mention, that general stochastic gradient methods under the imposed assumption of (weak) PL-type, are not well-understood yet. Improving this result is ongoing work.
>
>
> **2. Finite state/action spaces**
>
> The analysis presented in this paper relies on the tabular softmax parametrization. This tabular structure is exploited to obtain the PL-inequality which ensures the gradient domination property. Therefore, we cannot expect a straight forward extension to continuous state/action spaces. Nevertheless, there are recent works on specific policy gradient algorithms for the stationary discounted MDP problem (see for example *"Stochastic Policy Gradient Methods: Improved Sample Complexity for Fisher-non-degenerate Policies", I. Fatkhullin et. al (2023)*), where a different policy parametrisation fulfills similar properties on continuous sets. Transferring these to the non-stationary case would be an interesting future research direction.
>
>
> **3. Undiscounted problems**
>
> Many real-world problems in practice are indeed undiscounted MDPs. A few specific examples are given in the following:
> 1. Training of language models is typically done using policy gradient. This is also an undiscounted MDP problem.
> 2. Project Scheduling: An agent schedules tasks to complete a project in finite time. The goal is to minimize the total cost of the project while ensuring all tasks are completed on time.
> 3. Medical Treatment Planning: An agent decides how to allocate resources (e.g., water, fertilizer, labor) to different crops over a growing season to maximize overall yield while minimizing resource usage.
> 4. Online Advertisement: An agent allocates a fixed budget to different campaigns over a specific period. The goal is to maximize the total number of clicks or conversions within the budget constraint.
>
> We will add a brief discussion in the introduction of the manuscript.
>
>
> **4. Clarification of new results**
>
> Due to the specific non-stationary policy setting, many well-known results from general MDPs appear in a different version than usually known for stationary policies. For example, in Theorem 2.2 we show the policy gradient theorem for the specific non-stationary setting. To the best of our knowledge this version is nowhere else stated, even though the proof idea is similar to the stationary policy version in finite-time horizons and therefore the result is no surprise. Still, all results stated in this paper are novel and whenever there exists a similar version in the stationary setting, we pointed that out in the texts before or after the results.
>
>
> **5. Dimension of $\theta$**
>
> As in almost all recent articles on the subject, we consider the tabular softmax parametrization such that for every state-action pair one parameter is used. Due to the non-stationary policy, we train a different parameter in every decision epoch of the MDP. Thus, the dimension of $\theta$ for time-point $h$ equals the number of state-action pairs in this epoch. As we allow for the states to be different between epochs (does not need to be the case) and the number of actions to depend on the current state, the number of state action pairs for epoch $h$ are given by $\sum_{s\in\mathcal{S}} |\mathcal{A}_s|$.
> In settings, where $\mathcal{A}_s = \mathcal{A}$ for all states and $\mathcal{S}_h =\mathcal{S}$ for all epochs, this simplifies to a dimension of $d_h = d = |\mathcal{S}|\cdot|\mathcal{A}|$ for all epochs.
>
>
> **6. Notation Lemma 2.1**
>
> The notation $\pi_{(h)}$ stands for a non-stationary policy from time-point $h$ to $H-1$ and is defined in line 105.
> Especially in Lemma 2.1, where we start in epoch $h$ with $S_h =s$, we used this notation to point out that there is no dependency on earlier policies. Still as we dropped the subscript $(h)$ in the policy of the value and advantage function you are right that the notation in this line is inconsistent. We work over these cases throughout the manuscript.

---

> > ### Comment · Reviewer_3eeX · 2023-08-12
> > **Response to Rebuttal**
> >
> > I would like to thank the authors for the detailed answers. I will keep my score.

---

### Official Review · Reviewer_6bDX · 2023-07-10

**Soundness:** 3 good
**Presentation:** 3 good
**Contribution:** 2 fair
**Rating:** 6
**Confidence:** 4

**Summary:**

The paper analyzes the convergence of non-stationary softmax policy gradient methods where the policy is parameterized by different parameters at each decision epoch. The convergence results on REINFORCE algorithm are provided under undiscounted finite-time and infinote-horizon cases.

**Strengths:**

- The paper obtains new global convergence results for softmax policy under undiscounted finite-time and infinite-horizon MDPs. The assumptions used are standard in policy gradient literature.

- The analyses in the paper also cover policy parameterized by deep neural network in deep reinforcement learning.

- The paper is well-written and easy to follow.

**Weaknesses:**

- My major concern is about the factor $c_h$ in Theorem 3.8. The authors use it as a constant but I argue it is not. Since $c_h$ is the infimum over all iteration of the minimum output of softmax policy, it is not known or can be computed before-hand. In fact, it depends on the choice of number of iteration $n$ and the current parameter $\theta_n$. More discussion is needed in this case.

- The results in Theorem 3.8 and Theorem 4.4 are valid for the exact policy gradient which is rather restrictive and impractical. I believe having results in the stochastic case using stochastic policy gradient will strengthen the contribution of the paper.

**Questions:**

1. The factor $c_h$ can be very small (if the policy is well-trained). Can we impose certain condition to overcome this?

2. I wonder if the results can be extended to other class of policy gradient method, such as ones with different policy gradient estimator.

**Limitations:**

I do not find clear discussion on the limitation of the work but I find the use of exact gradient is the main limitation for the key results in the paper.

---

> ### Author Rebuttal · Authors · 2023-08-09
>
> Dear reviewer, thank you very much for taking the time to review our article!
>
>
> **1. Limitation to exact gradients: not true**
>
> It is correct, that access to exact gradients is a restrictive assumption for practical applications. However, the article has two main contributions. Firstly, we extend the analysis for policy gradient methods (with exact gradients) from discounted MDPs (Mei2020) to the dynamic programming inspired finite-time policy-gradient method for undiscounted MDPs. Secondly, we show how to overcome the limitation of knowledge of exact gradients, which you have pointed out. To be more precise, we show in both the finite-time and also in the discounted infinite-time case how to analyse the stochastic version when the exact gradients are replaced by samples (REINFORCE algorithm). You can find the results on removing the limitation of exact gradients in Section 4, Theorem 5.2 and Section 6. We will present a more detailed discussion about our main contributions in the introduction of our manuscript.
>
>
> **2. Constant $c_h$**
>
> As you observed correctly the factor $c_h$ is the infimum over all iterations of the minimum over all states, but for the probability of taking the **best action** $a^\ast(s)$ in these states. Hence, training a perfect policy is not a problem concerning this factor, as the probability of choosing the best action would then be very high.
> We also emphasize that $c_h$ is indeed a positive constant with respect to the number of iterations $n$. This result is formulated in Lemma 3.7. We will make this more precise in the manuscript.
>
>
>
> **3. Extend to other classes of policy gradient methods**
>
> Using different policy gradient methods like natural gradient or regularized policy gradient requires a new derivation of the PL-inequality and is not within the scope of our article. There are some results regarding these methods for discounted MDPs and as mentioned in the conclusion considering these algorithms in the context of finite-time MDPs is an interesting future research direction.
> On the other hand considering different type of stochastic policy gradient estimators to approximate the exact gradient (different from the one we introduced in (8)) would indeed be possible as long as they are unbiased with bounded variance and the variance can converge to zero by increasing the batch-size. In particular, it would be interesting to consider variance reduced gradient estimators in the spirit of stochastic gradient descent methods for finite sums such as stochastic average gradient (SAG), stochastic average gradient amélioré (SAGA) or stochastic variance reduced gradient methods (SVRG).

---

### Author Rebuttal · Authors · 2023-08-09

Dear reviewers, thank you all very much for your reviews which we appreciate a lot!


**1. Stochastic vs. deterministic**

Please note that there are two contributions of the article. We will improve the abstract to make this more clear. Extending Mei2020 to the finite case and, most importantly, show that the **stochastic policy gradient** (sample based) approach is meaningful. This covers also the famous REINFORCE algorithm for discounted MDPs for softmax parametrization. The second part is a non-trivial extension of the fact that, under PL inequality in weak form, GD (resp. PG) can be extended to SGD (resp. SPG) as long as sufficient sample sizes are used. This extension is in itself non-trivial and to best of our knowledge not known in the literature of SGD, yet. Moreover, since in contrast to SGD in stochastic policy gradient samples are not iid, it is somewhat surprising that the arguments can be extended.

**2. Numerical examples**

As some of the referees asked for numerical validations we enclosed a numerical toy example of a very simple MDP problem of optimally stopping when throwing a dice 5 times. The example was chosen as this is one of the only non-trivial examples that we know of for which exact policy gradients can be computed. The simulations show that the theoretical results (in the exact gradient setup) are sharp up to constants.

Figure 1 in the attached pdf shows a log-log-plot to visualize the $\frac{1}{n}$ convergence rate for the deterministic time dependent policy gradient algorithm proven in Theorem 3.8. Here, the magenta dotted line is a plot of the upper bound $\frac{4(H-h)R^* |\mathcal{A}|}{c_h^2 n}$ as a function in $n$. And the red line visualizes the difference of $J_h^\ast - J_h(\theta_n)$ also as a function in $n$. The constant gab between the two lines shows that our rate is sharp up to constants.

In Figure 2 (b) we visualize a similar log-log-plot for Thm 5.1. A detailed description of the plot is given in the caption and we will analyze the figure in the following section.


**3. Comparison to a different algorithm**

One of you suggested that a comparison to the policy gradient algorithm where the parameters of different epochs are updated simultaneously (instead of backwards one by one) would be interesting. That algorithm is typically used in practice, without theoretical backup. Indeed, it turned out that this is a very interesting question, it seems like our estimates can be used to analyse the simultaneous update algorithm as well. In Figure 2 (a) and (b) we plotted different versions of our algorithm, with different target accuracy $\varepsilon$ and compare to the simultaneous update algorithm.

We refer you to the captions in order to understand the following take-aways.

What do we learn from these simulations? The funny curves in Fig 2 (a) reflect the epoch-by-epoch training scheme of our algorithm. The curves always move up fast if a new policy is started to be trained. The simultaneously trained algorithm does not have that feature, it improves fast at the beginning but takes more time to get close to the optimal value.

1. The blue lines from our backward induction algorithm perform slightly better than the magenta line from simultaneous training (and has theoretical error guarantees). For a given accuracy $\varepsilon$ the backwards trained algorithms is always a few gradient steps faster then the simultaneously trained algorithm up to the same accuracy. This can especially seen in Figure 2 (b), as the dotted blue line is slightly below the dashed magenta line.
2. Given a fixed number of overall gradient steps (summed over all epochs) the backward inductive approach is more accurate then the simultaneous approach if $\varepsilon$ is chosen accordingly.
3. The dot-dashed green line with the same number of updates in each epoch performs much worse than the epoch-dependent updates suggested by our theoretical analysis. Epochs should not be treated equal but according to the choice in Thm 5.1!

The policy gradient theorem (Thm 2.2) tells us that estimation errors are pushed backwards through the estimated reward to go (Q-function). Thus, errors at later epochs imply errors at earlier epochs.

What we see is not surprising. Our algorithm first optimizes the late policies (reducing errors that are pushed backwards) but does not optimize earlier epochs at the beginning. Thus, the algorithm must be poor at the beginning but has the chance to be accurate once all epochs are trained. The simultaneous algorithm first improves all policies better but then becomes weaker as there is less accuracy for late policies and the errors are automatically pushed towards all other epochs via the policy gradient theorem.

We believe that there is even a simple take-away for practitioners. If you can, then first train more accurately the later epochs, later focus on early epochs.


**4. Resulting interesting future research direction**

Training backwards vs. simultaneously are two extremes. We are quite certain that combinations of both will be the way to go in the stochastic setting. Of course, it will depend on the actual situation and how roll-outs must/can be sampled. For instance, in offline training with many data available this might result in a more efficient use of computational power. We are convinced that a theoretical analysis of the simultaneous and mixed approaches is possible, reducing to the arguments of this paper. After all, the reason for convergence is the backwards training, early training of other epochs only helps.

---

### Decision · Program_Chairs · 2023-09-21

**Decision:**

Reject

**Comment:**

This paper studies stochastic policy gradient (PG) methods under tabular softmax parameterization and finite-horizon settings. The main algorithms use policy gradient update in a backward iteration / dynamic programming fashion, i.e., updating the parameters from the bottom to top layer.

The authors firstly prove that the exact gradient version of this algorithm converge globally, and then with an increasing batch size they show that the stochastic version of the algorithm also achieves global convergence.

While it is definitely important to study global convergence of policy gradient methods with stochastic gradients, one reviewer pointed that the algorithms analyzed in the paper are not the widely used policy gradient methods, but a reduction from MDPs to contextual bandits using dynamic programming. During discussion the authors acknowledge that this is the case.

The AC has read the discussions as well as the paper and the proofs. I agree with Reviewer h4oF that this reduction from true PG to contextual bandit actually also reduces the difficulty in PG analysis. For example, the results in Section 3 for exact gradients are simpler to obtain than results for true PG in literature. Since the setting is finite-horizon, global convergence results can be obtained by repeatedly using existing asymptotic convergence results, because of every layer is like a contextual bandit as mentioned in the discussions. The stochastic results are obtained using increasing batch size which is similar to Ding et al., 2022.

The authors argued that the dynamic programming fashion of using PG has its own advantages, which I think could be possible. However, this actually requires further investigations and I believe conclusions cannot be drawn easily in this paper. This means the paper at least needs a major revision to either analyze the true PG (not the version in the paper) and keep the claims, or spend some efforts on seriously justifying the dynamic programming fashion of using PG is more favourable than the widely used version, which I think could be a bigger deal, since as reviewers mentioned the standard PG methods have already been widely used in many scenarios such as training language models.